# Fair Wrapping for Black-box Predictions

**Alexander Soen**
Australian National University
`alexander.soen@anu.edu.au`

**Ibrahim Alabdulmohsin**
Google Research
`ibomohsin@google.com`

**Sanmi Koyejo**
Google Research
Stanford University
`sanmik@google.com`

**Yishay Mansour**
Google Research
Tel Aviv University
`mansour@google.com`

**Nyalleng Moorosi**
Google Research
`nyalleng@google.com`

**Richard Nock**
Google Research
Australian National University
`richardnock@google.com`

**Ke Sun**
Australian National University
CSIRO's Data61
`sunk@ieee.org`

**Lexing Xie**
Australian National University
`lexing.xie@anu.edu.au`

## Abstract

We introduce a new family of techniques to post-process ("wrap") a black-box classifier in order to reduce its bias. Our technique builds on the recent analysis of improper loss functions whose optimization can correct any *twist* in prediction, unfairness being treated as a twist. In the post-processing, we learn a wrapper function which we define as an $\alpha$-tree, which modifies the prediction. We provide two generic boosting algorithms to learn $\alpha$-trees. We show that our modification has appealing properties in terms of composition of $\alpha$-trees, generalization, interpretability, and KL divergence between modified and original predictions. We exemplify the use of our technique in three fairness notions: conditional value-at-risk, equality of opportunity, and statistical parity; and provide experiments on several readily available datasets.

## 1 Introduction

The social impact of Machine Learning (ML) has seen a dramatic increase over the past decade – enough so that the bias of model outputs must be accounted for alongside accuracy [1, 14, 35]. Considering the various number of fairness targets [20] and the energy and CO2 footprint of ML [19, 28], the combinatorics of training accurate *and* fair models is non-trivial. This is especially so given the inherit incompatibilities of fairness constraints [17] and the underlying tension of satisfying fairness whilst maintaining accuracy. One trend in the field "decouples" the two constraints by post-processing *pretrained* (accurate) models to achieve fairer outputs [35]. Post-processing may be the only option if we have no access to the model's training data / algorithm / hyperparameters (etc.).

Within the post-processing approach, three trends have emerged: learning a new fair model close to the black-box, tweaking the output subject to fairness constraints, and exploiting sets of classifiers. If the task is class probability estimation [24], the estimated black-box is an accurate but potentially unfair posterior $\eta_u : \mathcal{X} \to [0, 1]$ which neither can be opened nor trained further. The goal is then to learn a fair posterior $\eta_f$ from it. In addition to the black-box constraint, a number of *desiderata* can be considered for post-processing. Ideally in correcting a black-box, we would want the approach to have **(flexibility)** in satisfying substantially different fairness criteria, **(proximity)** of the learnt

36th Conference on Neural Information Processing Systems (NeurIPS 2022).

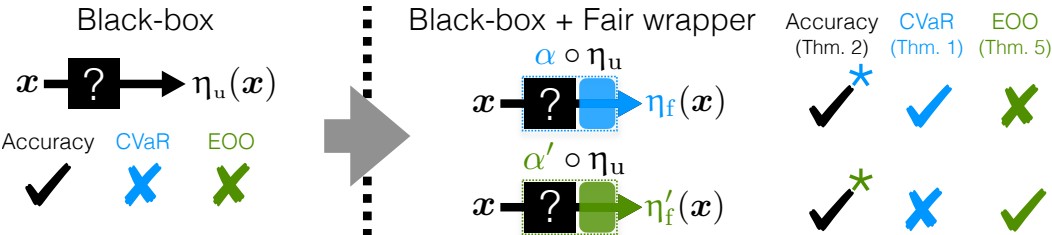

Figure 1: Summary of using different $\alpha$-correction wrappers to obtain different fairness criteria guarantees. See Sections 5 and 6 for full details on guarantees.

$\eta_f$ to the original $\eta_u$, and meaningful (**composability**) properties if, *e.g.*, $\eta_f$ was later treated as a new black-box to be post-processed. To facilitate a specific style of correction, we may also want the representation of the correction to facilitate (**explainability**) for auditing the post-processing procedure and bounds on the increased model (**complexity**) of the final classifier $\eta_f$. In training the correction, algorithmically we would also want guarantees for (**convergence**).

**Our contribution** satisfies the aforementioned desiderata in its correction, representation, and algorithmic guarantees. By leveraging recent theory in *im*proper loss functions, we utilize a *universal* correction of black-box posteriors defined by the $\alpha$-loss. This allows for a flexible correction which yields convenient divergence bounds between $\eta_f$ and $\eta_u$, a convenient form for the Rademacher complexity of the class of $\eta_f$, and a simple composability property. Representation-wise, the corrections we learn are easy-to-interpret tree-shaped functions that we define as $\alpha$-*trees*. Algorithmically speaking, we provide two formal boosting algorithms to learn $\alpha$-trees building upon seminal results [15]. We demonstrate our algorithm for conditional value-at-risk (CVAR) [32], equality of opportunity (EOO), and statistical parity; as depicted in Fig. 1. Experiments are provided against five baselines on readily available datasets. All proofs and more experiments are in an Appendix denoted as SI.

## 2 Related Work

Post-processing models to achieve fairness is one of three different categories in tackling the ML + fairness challenge [35, Section 6.2]. Although other notions exist, *e.g.* individual fairness [12], we limit our analysis to group fairness, which concerns itself with ensuring that statistics of sub-populations are similar. We further segment this cluster into three subsets: (I) approaches learning a new model with two constraints: being close to the pretrained model and being fair [16, 22, 31, 34]; (II) approaches biasing the output of the pretrained model at classification time, modifying observations for fairer outcomes [1, 14, 18, 21, 33, 34]; and (III) techniques consisting of exploiting sets of models to achieve fairness [11]. None of these approaches formulates substantial guarantees on all of the desiderata in the introduction. Some bring contributions with the (**flexibility**) of being applicable to more than two fairness notions [7, 31, 11, 34]. Two of which provide the convenience of analytic conditions on new fairness notions to fit in the approach [31, 11]. However, for all of them, the algorithmic price-tag is unclear [7, 11] or heavily depends on convex optimization routines [31]. [1, 34] provide strong guarantees regarding (**proximity**), w.r.t. *consistency and generalization*. To our knowledge, our approach of correcting prediction unfairness through improper losses [29] is new.

## 3 Setting and Motivating Example

Let $\mathcal{X}$ be a domain of observations, $\mathcal{Y} \doteq \{-1, 1\}$ be labels and $S$ is a sensitive attribute in $\mathcal{X}$. We assume that the modalities of $S$ induce a partition of $\mathcal{X}$. We further let $D$ denote the joint measure over $\mathcal{X} \times \mathcal{Y}$, $M$ denote the marginal measure over $\mathcal{X}$, and $\pi \doteq \mathbb{P}[Y = 1]$ being the prior. We denote conditioning of $M$ through a subscript, *e.g.*, $M_s$ for $s \in S$ denotes $M$ conditioned on a sensitive attribute subgroup $S = s$. We leave the $\sigma$-algebra to be implicit (which is assumed to be the same everywhere). As is often assumed in ML, sampling is i.i.d.; we make no notational distinction between empirical and true measures to simplify exposition – most of our results apply for both.

Consider the task of learning a function $\eta \in [0, 1]^{\mathcal{X}}$ to estimate the true posterior $\eta^\star(\boldsymbol{x}) = \mathbb{P}[Y = 1 | X = \boldsymbol{x}]$ in binary classification. For instance, we may want to predict the probability of hiring an applicant for a company. Given the pointwise loss $L(\eta(X), \eta^\star(X))$, which determines the loss of a

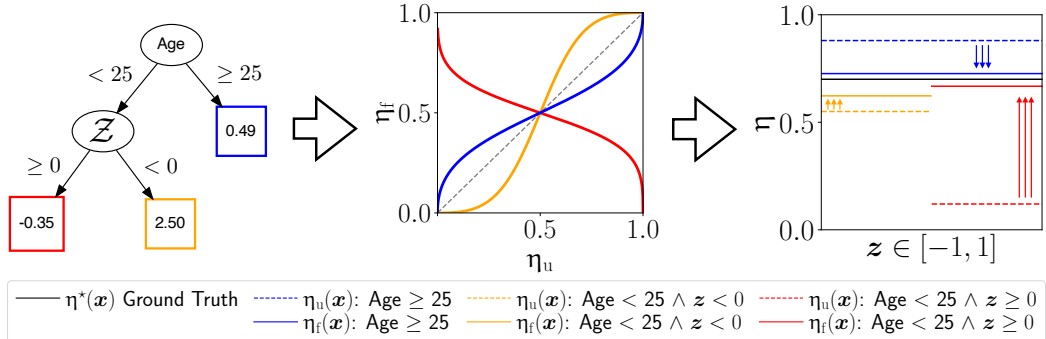

Figure 2: Improving CVaR for a toy hiring task with $\alpha$-trees. An $\alpha$-tree (left) transforms the posterior via (3) (middle); resulting in an input-dependent fairness correction of the posterior $\eta_u$ (right).

single example, the *(total) risk* is defined as

$$L(\eta; M, \eta^\star) \quad \doteq \quad \mathbb{E}_{X \sim M}\left[L(\eta(X), \eta^\star(X))\right] \tag{1}$$

(with slight abuse of notation). A low risk corresponds to good classification performance. In this paper, we consider the risk determined by the *log-loss*:

$$L(\eta; M, \eta^\star) \doteq \mathbb{E}_{X \sim M}\left[\eta^\star(X) \cdot -\log \eta(X) + (1 - \eta^\star(X)) \cdot -\log(1 - \eta(X))\right]. \tag{2}$$

We now consider a simple fairness problem, centered around the example of Fig. 2. Suppose that we are given a black-box $\eta_u$ which predicts hiring probabilities without considering fairness. Although the minimized total risk of (1) might be small, there can be discrepancies in the performance between different subgroups. Instead of considering the total risk, the predictive performance of specific subgroups can be examined through the *subgroup risk* $L(\eta_u; M_s, \eta^\star)$, for a subgroup $s \in S$. For instance, we might want to examine the discrepancy of subgroup risks among the age of applications. A natural fairness task would be to improve the worst performing subgroup, say $s_w \in S$.

To post-process unfairness, we want to learn a function $\alpha : \mathcal{X} \to \mathbb{R}$ which "wraps" $\eta_u$ and *lowers* the worst subgroup risk. We propose the following "wrapping", inspired by improper loss functions [29]

$$\eta_f(\boldsymbol{x}) \quad \doteq \quad \frac{\eta_u(\boldsymbol{x})^{\alpha(\boldsymbol{x})}}{\eta_u(\boldsymbol{x})^{\alpha(\boldsymbol{x})} + (1 - \eta_u(\boldsymbol{x}))^{\alpha(\boldsymbol{x})}} \quad \in [0, 1]. \tag{3}$$

Notice when $\alpha(\boldsymbol{x}) = 1$ the resulting posterior is the original $\eta_u(\boldsymbol{x})$. Importantly, (3) is flexible enough to transform any input black-box $\eta_u$ to any needed $\eta_f$. Looking at Fig. 2 (left and middle), intuitively by setting different $\alpha(\boldsymbol{x})$ values, (3) "sharpens" (yellow, $\alpha > 1$), "dampens" (blue, $0 < \alpha < 1$), or "polarity reverses" (red, $\alpha < 0$) the original posterior $\eta_u$. To improve fairness, we need a combination of these corrections to accommodate different subsets of the input domain (thus learning $\alpha(.)$ as a function). We specifically learn $\alpha(.)$ to be a tree structure, which allows an interpretable correction alongside other formal properties (Section 5).

**Definition 1.** *An $\alpha$-tree is a rooted, directed binary tree, with internal nodes labeled with observation variables. Outgoing arcs are labeled with tests over the nodes' variable. Leaves are real valued. $\Lambda(\Upsilon)$ is the leafset of $\alpha$-tree $\Upsilon$. An $\alpha$-tree induces a correction over posteriors as per (3) with $\alpha = \Upsilon$.*

Our fairness problem is now learning an $\alpha$-tree $\eta$ which provides a corresponding correction that improves the worst subgroup risk, *i.e.*, $L(\eta_u; M_{s_w}, \eta^\star) > L(\eta_f; M_{s_w}, \eta^\star)$. The entirety of Fig. 2 presents such a process. In this hiring task example, the ground truth hiring rate is constant *w.r.t.* the inputs, $\eta^\star(\boldsymbol{x}) = 0.7$. Despite this, the black-box $\eta_u(.)$ unfairly depends on the age of applicants and incorrectly depends on a noise feature $\mathcal{Z}$. By choosing $\alpha(\boldsymbol{x})$ as per Fig. 2 (left), the correction changes $\eta_u$ to be closer to $\eta^\star$, improving the risk of the worst subgroup (alongside the other subgroup in this example): the worse-case loss improves from 1.09 to 0.62.

Although the fairness criteria discussed might be considered simple, the procedure of iteratively minimizing the worse performing subgroup can be used to improve the *conditional value-at-risk* (lower is better) fairness criteria [32]:

$$\text{CVaR}_\beta(\eta_f) \quad \doteq \quad \mathbb{E}_S[L(\eta_f; M_S, \eta^\star) \mid L(\eta_f; M_S, \eta^\star) \geq L_\beta], \tag{4}$$

**Algorithm 1** TOPDOWN $(M_t, \eta_t, \Upsilon_0, B)$

**Input** mixture $M_t$, posterior $\eta_t$, $\alpha$-tree $\Upsilon_0$, $B \in \mathbb{R}_{+*}$;
Step 1: $\Upsilon \leftarrow \Upsilon_0$;
Step 2 : **while** stopping condition not met **do**
   Step 2.1 : pick leaf $\lambda^\star \in \Lambda(\Upsilon)$; // *i.e.* heaviest leaf
   Step 2.2 : $h^\star \leftarrow \arg\min_{h \in \mathcal{H}} \mathrm{H}(\Upsilon(\lambda^\star, h); M_t, \eta_t)$;
   Step 2.3 : $\Upsilon \leftarrow \Upsilon(\lambda^\star, h^\star)$; // split using $h^\star$ at $\lambda^\star$
Step 3 : label leaves $\forall \lambda \in \Lambda(\Upsilon)$:

$$\Upsilon(\lambda) \doteq \tilde{\iota}\left(\frac{1 + \mathsf{e}(M_\lambda, \eta_t)}{2}\right), \quad // \; \alpha\text{-value (5)}$$

**Output** $\Upsilon$;

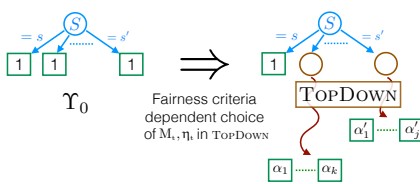

Figure 3: Picking $\Upsilon_0$ a stump on the fairness attribute allows to finely tune growths of sub-$\alpha$-trees to the fairness criterion at hand.

where $L_\beta$ is the risk value for the $\beta$ quantile among subgroups, which is user-defined. The difference is that $\mathrm{CVAR}_\beta$ not only considers the worse case subgroup, but all subgroups above the $L_\beta$ risk value (let these subgroups be $\mathcal{S}_\beta$). One can simply iteratively improve all $s \in \mathcal{S}_\beta$, as done in the example of Fig. 2. Indeed, in the example, $\mathrm{CVAR}_\beta$ with $\beta = 0.9$ improves from 1.09 to 0.62 (which is equivalent to worse-case loss in this case).

## 4 Growing Alpha-Trees

We now introduce the procedure to grow an $\alpha$-tree via a boosting algorithm TOPDOWN, Algorithm 1. TOPDOWN can be thought of as a generalization of the standard decision tree induction used for classification [15]. We first introduce relevant concepts from decision tree induction to explain TOPDOWN. We contextualize TOPDOWN through its application in improving the CVAR criteria.

We first introduce a technical assumption for the black-box $\eta_u$ to be post-processed:

**Assumption 1.** *The black-box prediction is bounded away from the extremes:* $\exists B > 0$ *such that*

$$\mathrm{Im}(\eta_u) \subseteq \mathbb{I} \doteq \left[(1 + \exp(B))^{-1}, (1 + \exp(-B))^{-1}\right] \quad (a.s.). \tag{6}$$

Compliance with Assumption 1 can be done by clipping the black-box's output with a user-fixed $B$ or making sure it is calibrated and then finding $B$.

**Entropy-based updates for fairness:** An important component in standard decision tree induction is the *edge* function, which measures the *label purity* (proportion of positive examples) of a decision tree node. We introduce a generalization which considers the *alignment purity* of a black-box.

**Definition 2.** *Let* $\iota(u) \doteq \log(u/(1-u))$ *the logit of* $u \in (0, 1)$ *and* $\tilde{\iota}(u) \doteq \iota(u)/B$ *a normalization which satisfies* $\tilde{\iota}(\mathbb{I}) = [-1, 1]$. *The **alignment edge** of* $M_t$ *and* $\eta_t$ *is defined as,*

$$\mathsf{e}(M_t, \eta_t) \doteq \mathbb{E}_{(X,Y) \sim D_t}\left[Y\tilde{\iota}(\eta_u(X))\right], \tag{7}$$

*where* $D_t$ *is the joint measure induced by* $M_t$ *and* $\eta_t$. *With Assumption 1,* $\mathsf{e}(M_t, \eta_t) \in [-1, 1]$.

By replacing the normalized logit $\tilde{\iota}$ with a constant 1, (7) reduces to a measure of label purity used in regular classification. In our case, (7) measures how well the black-box $\eta_u$ "aligns" with the true labels $Y$ through the logit. This also takes into account the "confidence" of the black-box's predictions: for a high alignment purity, predictions not only need be correct but also to be highly confident ($\eta_u$ close to the endpoints of $\mathbb{I}$). Similar to how the splits of a decision tree classifier are determined by the entropy of a tree's label purity, an $\alpha$-tree splits based on its alignment entropy.

**Definition 3.** *Given an $\alpha$-tree $\Upsilon$ with leafset $\Lambda$, when Assumption 1 is satisfied, the **entropy of** $\Upsilon$ is:*

$$\mathrm{H}(\Upsilon; M_t, \eta_t) \doteq \mathbb{E}_{\lambda \sim M_{\Lambda(\Upsilon)}}\left[\mathrm{H}_1(\lambda; M_t, \eta_t)\right], \tag{8}$$

*where* $\mathrm{H}(q) \doteq -q\log(q) - (1-q)\log(1-q)$, $\mathrm{H}_1(\lambda; M_t, \eta_t) \doteq \mathrm{H}((1 + \mathsf{e}(M_\lambda, \eta_t))/2)$, $M_\lambda$ *is* $M_t$ *conditioned to leaf* $\lambda \in \Lambda(\Upsilon)$, *and* $M_{\Lambda(\Upsilon)}$ *is a measure induced on* $\Lambda(\Upsilon)$ *by leaf weights on* $M_t$.

**Theorem 1.** *For any* $M_t, \eta_t$, *let* $\Upsilon$*'s leaves follow* (5). *Then* $L(\eta_f; M_t, \eta_t) \leq \mathrm{H}(\Upsilon; M_t, \eta_t)$.

Algorithm 1 can now be explained by repeatedly leveraging Theorem 1. Suppose that we have a hypothesis set of possible splits $\mathcal{H}$ to grow our $\alpha$-tree. Denote $\Upsilon(\lambda, h)$ as the $\alpha$-tree $\Upsilon$ split at leaf

$\lambda \in \Lambda(\Upsilon)$ using test $h$. The inner loop within Step 2 is the process of finding the best possible leaf splits which helps to minimize the $\alpha$-tree's entropy and to reduce the risk as per Theorem 1. The $\alpha$-values of (5) calculated in step 3 are those used to ensure Theorem 1 holds. By setting $M_t \leftarrow M_{s_w}$ and $\eta_t \leftarrow \eta^\star$, Algorithm 1 improves CVAR by iteratively improving the $\alpha$-tree's worst subgroup entropy $H(\Upsilon; M_{s_w}, \eta^\star)$, which as a surrogate improves the worst subgroup risk $L(\eta_u; M_{s_w}, \eta^\star)$. To accommodate for different $\beta$ quantiles values for CVAR, TOPDOWN can be run repeatedly (replacing the initial input tree $\Upsilon_0$) to progressively improve all $s \in \mathcal{S}_\beta$. Hence, to reduce CVAR, we basically

$$\text{use TOPDOWN with } M_t \leftarrow M_s \ (s \in \mathcal{S}_\beta) \text{ and } \eta_t \leftarrow \eta^\star. \qquad \text{(CVAR)}$$

As alluded to by the notation used to instantiate TOPDOWN, the inputs of the algorithm can be instantiated to optimize for fairness criteria beyond CVAR. This is discussed in Section 6. In the usual ML setting, $M_t, \eta_t$ can be *estimated* from a training sample (see Section 7).

**Initialization:** In the procedure of improving CVAR, the worst subgroups can be iteratively improved. However, we also need to make sure that improvement of a subgroup does not adversely affect another subgroup (which could potentially harm CVAR instead). As such, we introduce an additional structure to the $\alpha$-tree $\Upsilon$ by tweaking the initial tree structure $\Upsilon_0$ used in TOPDOWN. Since the fairness attribute $S$ partitions the dataset, a convenient choice of initializing the $\alpha$-tree is to split by the subgroup modalities, as depicted in Fig. 3. As such, we grow separate sub-$\alpha$-trees for each of the sensitive modalities. For CVAR, this allows the subgroup risk of individual subgroups to be tweaked without adversely affecting other subgroups.

## 5 Formal Properties

We move to the formal properties of our approach. We first detail the background of improper loss functions which motivates our correction given in Section 3. We then present the formal properties of this correction. The useful properties of having $\alpha(.)$ represented by a tree structure is then discussed. Finally, we present a convergence analysis of Algorithm 1 and an alternative boosting scheme.

**Can $\eta_f$ as per (3) correct (any) potential unfairness? Yes.** In short, this comes from recent theory in *improper loss functions* for class probability estimation (CPE) [29]. We are interested in the pointwise minimizer (eventually set-valued) of:

$$t_\ell(\eta) \ \doteq \ \arg \inf_{u \in [0,1]} L(u, \eta). \qquad (9)$$

Dubbed as the *Bayes tilted estimate* of a loss $\ell$ [29], $t_\ell(\eta)$ is the set of optimal "responses" given a ground truth (pointwise) posterior $\eta$. Common loss functions are *proper*: the ground truth value $\eta \in t_\ell(\eta)$ is an optimal response. However, in the case where $\eta$ cannot be trusted (for instance when it is *unfair*), we may not want to default to imitating $\eta$. In addition we also want to make sure that the Bayes tilted estimate can fit to any desired (in our context, *fair*) target. The so-called $\alpha$-loss $\ell^\alpha$, which generalizes the (proper) log-loss, is a good candidate parameterized by a variable $\alpha$. Its Bayes tilted estimate is the pointwise version of (3), for $\alpha \notin \{0, \infty\}$ and $\eta \neq 1/2$:

$$t_{\ell^\alpha}(\eta) \ = \ \{\eta^\alpha/(\eta^\alpha + (1-\eta)^\alpha)\}. \qquad (10)$$

Importantly for $\alpha$-losses, for any $\eta \notin \{0, 1/2, 1\}$ and any $\eta' \in (0, 1)$, there exists $\alpha \in \mathbb{R}$ in (10) such that $t_{\ell^\alpha}(\eta) = \{\eta'\}$. This property, called *twist-properness* [29], allows for any pointwise correction. By extending $\alpha$ to a function (of $x \in \mathcal{X}$, as per (3)), twist-properness ensures that given an initial unfair posterior an appropriately learned $\alpha(.)$ can correct any unfairness. This allows for **(flexible)** fairness post-processing – different $\alpha$ functions can be learned for different criteria (*i.e.*, Fig. 1).

**Why use the Log-Loss?** As per Section 3, we minimize the log-loss. We choose the log-loss for two reasons: ① it is strictly proper and so minimizing $L(\eta_f; M_t, \eta_t)$ (*i.e.*, via TOPDOWN) "pushes" $\eta_f$ towards target $\eta_t$; and ② it is the $\alpha$-loss for $\alpha = 1$, so we are guaranteed that for the minimizer $\eta_f \to \eta_u \iff \alpha \to 1$. With alternative (*i.e.* non-strictly proper) loss, we might have only "$\Rightarrow$".

**Do we have guarantees on some proximity of $\eta_f$ with respect to $\eta_u$? Yes, with light assumptions.** We examine the **(proximity)** of black-box and post-processed posteriors with the KL divergence [2]:

$$\text{KL}(\eta_u, \eta_f; M) \ \doteq \ \mathbb{E}_{(X,Y) \sim D_u} \left[ \log \left( dD_u((X,Y))/dD_f((X,Y)) \right) \right], \qquad (11)$$

where $D_u, D_f$ are the product measures defined from $M$ and their respective posteriors. To bound the proximity (11), we present setting **(S1)**.

**(S1)** Assumption 1 holds for some $0 < B \leq 3$ and function $\alpha$ satisfies $|\alpha(\boldsymbol{x}) - 1| \leq 1/B$ (a.s.).

This setting lead to the following *data independent* proximity bound.

**Theorem 2.** *For any* M, **(S1)** *implies* $\mathrm{KL}(\eta_\mathrm{u}, \eta_\mathrm{f}; \mathrm{M}) \leq \pi^2/(6 \cdot (2 + \exp(B) + \exp(-B)))$.

As an example, fix $B = 3$ for **(S1)**. In this case, we want $\alpha(.) \in [2/3, 4/3]$ (a.s.) which is a reasonable sized interval centered at 1. The clamped black-box posterior's interval is approximately $[0.04, 0.96]$, which is quite flexible and the distortion is upperbounded as $\mathrm{KL}(\eta_\mathrm{u}, \eta_\mathrm{f}; \mathrm{M}) \leq 7.5E - 2$.

**Is the composition of transformations meaningful? Yes.** The analytical form in (3) brings the following easy-to-check **(composability)** property.

**Lemma 1.** *The composition of any two wrapping transformations* $\eta_\mathrm{u} \overset{\alpha}{\mapsto} \eta_\mathrm{f} \overset{\alpha'}{\mapsto} \eta_\mathrm{f}'$ *following* (3) *is equivalent to the single transformation* $\eta_\mathrm{u} \overset{\alpha \cdot \alpha'}{\mapsto} \eta_\mathrm{f}'$.

This gives an *invertibility* condition – wrapping $\eta_\mathrm{f}$ with $\alpha' = 1/\alpha$ recovers the original black-box $\eta_\mathrm{u}$.

**Given some capacity parameter for $\eta_\mathrm{u}$, can we easily compute that of $\eta_\mathrm{f}$? Yes, *e.g.*, for decision trees.** Such a question is particularly relevant for generalization. As we are using the log-loss (2), a relevant capacity notion to assess the uniform convergence of risk minimization for the whole wrapped model is the Rademacher **(complexity)** [3]. We examine the following set of functions:

$$\mathcal{H}_\mathrm{f} \doteq \{\eta_\mathrm{f} : \eta_\mathrm{f}(\boldsymbol{x}) \text{ given by (3) with } \alpha, \eta_\mathrm{u}; \forall(\alpha, \eta_\mathrm{u})\}, \tag{12}$$

where we assume known the set of functions from which $\eta_\mathrm{u}$ was trained. We now assume we have a $m$-training sample $\mathcal{S} \doteq \{(\boldsymbol{x}_i, y_i) \sim \mathrm{D}\}_{i=1}^m$. The empirical Rademacher complexity of a set of functions $\mathcal{H}$ from $\mathcal{X}$ to $\mathbb{R}$, $\mathfrak{R}_\mathcal{S}(\mathcal{H}) \doteq \mathbb{E}_{\boldsymbol{\sigma}} \sup_{h \in \mathcal{H}} \mathbb{E}_i[\sigma_i h(\boldsymbol{x}_i)]$ (sampling uniform with $\sigma_i \in \{-1, 1\}$), is a capacity parameter that yields efficient control of uniform convergence when the loss used is Lipschitz [3, Theorem 7], which is the case of the log-loss. To see how the $\alpha$-tree affects the Rademacher complexity of classification using $\eta_\mathrm{f}$ instead of $\eta_\mathrm{u}$, suppose real-valued prediction based on $\eta_\mathrm{u}$ is achieved via logit mapping, $\iota \circ \eta_\mathrm{u}$ (12). Such mappings are common for decision trees [26].

**Lemma 2.** *Suppose* $\{\eta_\mathrm{u}\}$ *is the set of decision trees of depth* $\leq d$ *and denote* $\mathfrak{R}_\mathcal{S}(\mathrm{DT}(d))$ *the empirical Rademacher complexity of decision trees of depth* $\leq d$ *[3] and* $d'$ *the maximum depth allowed for $\alpha$-trees. Then we have for* $\mathcal{H}_\mathrm{f}$ *in* (12): $\mathfrak{R}_\mathcal{S}(\mathcal{H}_\mathrm{f}) \leq \mathfrak{R}_\mathcal{S}(\mathrm{DT}(d + d'))$.

The proof is straightforward once we remark that elements in $\mathcal{H}_\mathrm{f}$ can be represented as decision trees, where we plug at each leaf of $\eta_\mathrm{u}$ a copy of the $\alpha$-tree $\Upsilon$.

**Does Algorithm 1 have any convergence properties? Yes, it is a boosting algorithm.** Following a similar blueprint to classical decision tree induction, it comes with no surprise that TOPDOWN can achieve boosting compliant convergence. To show that TOPDOWN is a boosting algorithm, we need a *Weak Hypothesis Assumption (WHA)*, which postulates informally that each chosen split brings a small edge over random splits for a tailored distribution.

**Definition 4.** *Let* $\lambda \in \Lambda(\Upsilon)$ *and* $\mathrm{D}_{\mathrm{t}\lambda}$ *be the product measure on* $\mathcal{X} \times \mathcal{Y}$ *conditioned on* $\lambda$. *The* ***balanced*** *product measure* $\mathrm{D}_{\mathrm{t}\lambda}'$ *at leaf* $\lambda$ *is defined as* ($z \doteq (\boldsymbol{x}, y)$ *for short):*

$$\mathrm{D}_{\mathrm{t}\lambda}'(z) \doteq \frac{1 - \mathrm{e}(\mathrm{M}_\lambda, \eta_\mathrm{t}) \cdot y\tilde{\iota}(\eta_\mathrm{u}(\boldsymbol{x}))}{1 - \mathrm{e}(\mathrm{M}_\lambda, \eta_\mathrm{t})^2} \cdot \mathrm{D}_{\mathrm{t}\lambda}(z). \tag{13}$$

We check that $\int_\lambda \mathrm{d}\mathrm{D}_{\mathrm{t}\lambda}' = 1$ because of Def. 2. Our balanced distribution is named after [15]'s: ours indeed generalizes theirs. The key difference comes from the change in setting, where we consider the alignment purity of a leaf and not its label purity. The "*fairness-free case*" where $\tilde{\iota}(.)$ is replaced by constant 1 yields the original balanced distribution [15]. We now state our WHA.

**Assumption 2.** *Let* $h : \mathcal{X} \to \mathcal{Y}$ *be the function splitting leaf* $\lambda$. *For* $\gamma > 0$, *then* $h$ $\gamma$-***witnesses*** *the Weak Hypothesis Assumption (**WHA**) at* $\lambda$ *iff*

**(i)** $\left| \mathbb{E}_{(\mathsf{X},\mathsf{Y}) \sim \mathrm{D}_{\mathrm{t}\lambda}'} [\mathsf{Y}\tilde{\iota}(\eta_\mathrm{u}(\mathsf{X})) \cdot h(\mathsf{X})] \right| \geq \gamma$; **(ii)** $\mathrm{e}(\mathrm{M}_\lambda, \eta_\mathrm{t}) \cdot \mathbb{E}_{(\mathsf{X},\mathsf{Y}) \sim \mathrm{D}_{\mathrm{t}\lambda}} \left[ (1 - \tilde{\iota}^2(\eta_\mathrm{u}(\mathsf{X}))) \cdot h(\mathsf{X}) \right] \leq 0$.

Intuitively, (i) gives a condition on the split $h$'s correlation with unfair posterior $\eta_\mathrm{u}$ agreement with labels and (ii) does the same for the confidence of $\eta_\mathrm{u}$ predictions. Similarly to the balanced

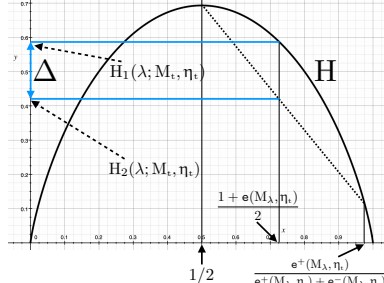 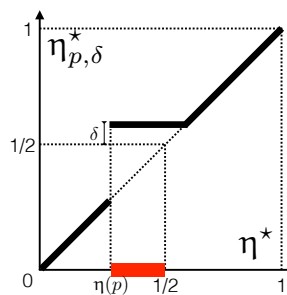

Figure 4: *Left*: Difference between the per-leaf bounds on risk $L(\eta_u; M_t, \eta_t)$ using (8) and Theorem 1 (conservative scoring) and (15) (audacious scoring). Details in the proof of Lemma 3. *Right*: A representation of the $(p, \delta)$-pushup of $\eta^\star$, where $\eta(p) \doteq \inf \eta^\star(\mathcal{X}_p) < 1/2$ (Def. A). All posteriors in $[\eta(p), 1/2 + \delta]$ are mapped to $1/2 + \delta$; others do not change. New posterior $\eta^\star_{p,\delta}$ eventually reduces the accuracy of classification for observations whose posterior lands in the thick red interval ($x$-axis).

distribution, in the fairness-free case where we only care about the label purity of splits, the WHA simplified to that of [15]'s – *i.e.*, only (i) matters. A further discussion on our balanced distribution and WHA is in the SI, Section III. We now state TOPDOWN's boosting compliant **(convergence)**.

**Theorem 3.** *Suppose (a) Assumption I holds, (b) we pick the heaviest leaf to split at each iteration in Step 2.1 of* TOPDOWN *and (c)* $\exists \gamma > 0$ *such that each split* $h^\star$ *(Step 2.2) in* $\Upsilon$ $\gamma$-witnesses the *WHA. Then there exists a constant* $c > 0$ *such that* $\forall \varepsilon > 0$, *if the number of leaves of* $\Upsilon$ *satisfies* $|\Lambda(\Upsilon)| \geq (1/\varepsilon)^{c \log(\frac{1}{\varepsilon})/\gamma^2}$, *then* $\eta_f$ *crafted from (3) using* TOPDOWN's $\Upsilon$ *achieves* $L(\eta_f; M_t, \eta_t) \leq \varepsilon$.

**Are there alternative ways of growing $\alpha$-trees? Yes.** Let us call *conservative* the scoring scheme in (5). There is an alternative scoring scheme, which can lead to substantially larger corrections in absolute values, hence the naming, and yields better entropic bounds for the $\alpha$-tree.

**Definition 5.** *For any mixture* $M_t$ *and posteriors* $\eta_u, \eta_t$, *let* $e^+(M_t, \eta_t)$ *and* $e^-(M_t, \eta_t)$ *be defined by*

$$e^\pm(M_t, \eta_t) \doteq \mathbb{E}_{(X,Y) \sim D_t} [\max\{0, \pm Y \tilde{\iota}(\eta_u(X))\}]. \tag{14}$$

*The **audacious** scoring schemes at the leaves of the $\alpha$-tree replaces (5) in Step 3 by:*

$$\Upsilon(\lambda) \doteq \tilde{\iota}\left(\frac{e^+(M_\lambda, \eta_t)}{e^+(M_\lambda, \eta_t) + e^-(M_\lambda, \eta_t)}\right), \quad \forall \lambda \in \Lambda(\Upsilon).$$

**Theorem 4.** *Suppose Assumption I holds and let* $H_2(q) \doteq H(q)/\log 2 \ (\in [0,1])$, $H$ *being defined in Definition 3. For any leaf* $\lambda \in \Lambda(\Upsilon)$, *denote for short:*

$$H_2(\lambda; M_t, \eta_t) \doteq \log(2) \cdot \left(1 + (e^+_\lambda + e^-_\lambda) \cdot \left(H_2\left(\frac{e^+_\lambda}{e^+_\lambda + e^-_\lambda}\right) - 1\right)\right),$$

*where let* $e^b_\lambda \doteq e^b(M_\lambda, \eta_t), \forall b \in \{+, -\}$. *Using audacious scoring, we get instead of Theorem 1:*

$$L(\eta_f; M_t, \eta_t) \leq \mathbb{E}_{\lambda \sim M_{\Lambda(\Upsilon)}} [H_2(\lambda; M_t, \eta_t)]. \tag{15}$$

While upperbounds in Theorem 1 and Theorem 4 may look incomparable, it takes a simple argument to show that (15) is never worse and can be much tighter.

**Lemma 3.** $\forall \alpha$-*tree* $\Upsilon$, $\mathbb{E}_{\lambda \sim M_{\Lambda(\Upsilon)}} [H_2(\lambda; M_t, \eta_t)] \leq H(\Upsilon; M_t, \eta_t)$.

It thus comes at no surprise that using the audacious scoring also results in a boosting result for TOPDOWN guaranteeing the same rates as in Theorem 3. It also takes a simple picture to show that the per-leaf slack in Lemma 3 can be substantial, a slack which can be represented using a simple picture, see Figure 4 (left), following from the use of Jensen's inequality in the Lemma's proof.

**As audacious scoring is better boosting-wise, is conservative scoring useful? Yes.** If we only cared about accuracy, we would barely have any reason to use the conservative correction. Even thinking about generalization, the Rademacher complexity of decision trees is a function of their depth so faster the convergence, the better [3, Section 4.1]. Adding fairness substantially changes the picture: some constraints, like equality of opportunity (Section 6) can antagonize accuracy to some extent. In such a case, using the conservative correction can keep posteriors $\eta_u$ and $\eta_t$ close enough (Theorem 2) so that fairness can be achieved without substantial sacrifice on accuracy.

# 6 Fairness and Societal Considerations

In this section, we present the fairness guarantees TOPDOWN can achieve. In particular, we provide a discussion about how Theorem 3 can guarantee minimization of the CVAR criteria. Furthermore, we provide alternative inputs to TOPDOWN which allows for EOO to be targeted as a fairness criteria. In the SI Section II, we further present a treatment of statistical parity. Lastly, we discuss how using $\alpha$-trees provides explainable corrections and how utilization of the sensitive attribute (as per Fig. 3) can be circumvented.

**Guarantees on CVaR:** As discussed in previous sections, one way to improve the CVAR fairness criteria (as per (4)) is to focus optimization on the worst treated subgroups. Given a specified quantile group $\beta$ and the set of worse subgroups $\mathcal{S}_\beta$, we can repeat (CVAR) until $\mathrm{CVAR}_\beta(\eta_\mathrm{f})$ gets below a threshold or (more specifically) its worst tread group gets a risk below a threshold (*i.e.*, a stopping criterion). Importantly, Theorem 3 provides a guarantee: to ensure $\mathrm{CVAR}_\beta$ is below $\varepsilon$, we simply need to boost for $|S|$ times the tree size bound $|\Lambda(\Upsilon)|$ given in Theorem 3.

**Guarantees on EOO:** EOO requires to smooth discrimination within an "advantaged" group, modeled by the label $\mathsf{Y} = 1$ [14]. We say that $\eta_\mathrm{f}$ achieves $\varepsilon$-equality of opportunity iff a mapping $h_\mathrm{f}$ of $\eta_\mathrm{f}$ to $\mathcal{Y}$ (*e.g.* using the sign of its logit $\iota$) satisfies

$$\max_{s \in S} \mathbb{P}_{\mathsf{X} \sim \mathrm{P}_s} [h_\mathrm{f}(\mathsf{X}) = 1] - \min_{s \in S} \mathbb{P}_{\mathsf{X} \sim \mathrm{P}_s} [h_\mathrm{f}(\mathsf{X}) = 1] \leq \varepsilon, \tag{16}$$

where $\mathrm{P}_s$ is the positive observations' measure conditioned to value $\mathsf{S} = s$ for the sensitive attribute. It is clear that EOO can be antagonistic to accuracy: the rate of advantage in the data $\mathrm{D}$ may not be equal among the subgroups. As such, unlike CVAR, we do not want to target the Bayes posterior $\eta_\mathrm{t} \not\leftarrow \eta^\star$ for EOO. Instead, we target a skewed posterior which aims to improve the least advantaged subgroup, *i.e.*, increasing $s^\circ \in \arg\min_{s \in S} \mathbb{P}_{\mathsf{X} \sim \mathrm{P}_s} [h_\mathrm{f}(\mathsf{X}) = 1]$. Our strategy consists of picking a target posterior which skews part of the original $\eta^\star$ to be more advantaged, thus reducing the LHS of (16) until (16) is satisfied[1]. For this, we create a $(p, \delta)$-*pushup* of $\eta^\star$, defined in SI Appendix I.

Fig. 4 (right) presents an example of a pushup map. Notice that the pushup only changes the predicted probability of example which do not have a "confident prediction" (the interval $[\eta(p), 1/2 + \delta]$). Intuitively, $p$ controls how many examples are corrected and $\delta$ controls how much the correction "pushes up" advantage, further discussion in SI Appendix I. We then run TOPDOWN using as mixture the *positive* measure conditioned to $\mathsf{S} = s^\circ$ and $p \doteq \mathbb{P}_{\mathsf{X} \sim \mathrm{P}_{s^\star}} [h_\mathrm{f}(\mathsf{X}) = 1] + \varepsilon/(K-1), \delta \doteq K\varepsilon/(K-1)$, with $K > 1$ user-fixed. Thus, we do

> Use TOPDOWN with $\mathrm{M}_\mathrm{t} \leftarrow \mathrm{P}_{s^\circ}$ and $\eta_\mathrm{t} \leftarrow \eta^\star_{p,\delta}$ . (EOO)

**Theorem 5.** *If* TOPDOWN *is run until* $L(\eta_\mathrm{f}; \mathrm{M}_\mathrm{t}, \eta_\mathrm{t}) \leq (\varepsilon^4/2) + \mathbb{E}_{\mathsf{X} \sim \mathrm{M}_\mathrm{t}} [\mathrm{H}(\eta_\mathrm{t}(\mathsf{X}))]$, *then after the run we observe* $\mathbb{P}_{\mathsf{X} \sim \mathrm{P}_{s^\star}} [h_\mathrm{f}(\mathsf{X}) = 1] - \mathbb{P}_{\mathsf{X} \sim \mathrm{P}_{s^\circ}} [h_\mathrm{f}(\mathsf{X}) = 1] \leq \varepsilon$.

In the full context of EOO, in the optimization we should not wait to get the bound on $L(\eta_\mathrm{f}; \mathrm{M}_\mathrm{t}, \eta_\mathrm{t})$. Rather, we should make sure (a) we update $\arg\min_{s \in S} \mathbb{P}_{\mathsf{X} \sim \mathrm{P}_s} [h_\mathrm{f}(\mathsf{X}) = 1]$ (and thus $s^\circ$) after each split in the $\alpha$-tree and (b) we keep $\arg\max_{s \in S} \mathbb{P}_{\mathsf{X} \sim \mathrm{P}_s} [h_\mathrm{f}(\mathsf{X}) = 1]$ as is, to prevent switching targets and eventually composing pushup transformations for the same $\mathsf{S} = s^\circ$, which would not necessarily comply with our theory. Notably, the guarantee presented in Theorem 5 depends on the mapping $h_\mathrm{f}$ and not the direct posterior $\eta_\mathrm{f}$, as typically considered [14]. When taking a threshold (sign of the logit), $h_\mathrm{f}$ can be interpreted as forcing the original posterior to be extreme values of 0 or 1.

Unlike the CVAR case, EOO (as per (17), SI) requires an explicit approximation of $\eta^\star$. In practice, we find that taking a simple approximation of $\eta^\star$ still can yield fairness gains. However, if one does not want to make such an approximation, one can adapt the statistical parity approach (detailed in SI, Section II). Similarly, if wants to consider the typical EOO definitions depending on posterior values, the target measure can be replaced (*i.e.*, swapping measure $\mathrm{M}_\mathrm{t}$ with the positive examples $\mathrm{P}$).

**Explainability:** $\alpha$-trees using the initialization proposed in Section 4 (and Fig. 3) allows for **(explainability)** properties similar to that of decision tree classifiers. Fixing a sensitive attribute

---

[1]If we instead *reduce* $\arg\max_{s \in S} \mathbb{P}_{\mathsf{X} \sim \mathrm{P}_s} [h_\mathrm{f}(\mathsf{X}) = 1]$ we get a symmetric strategy. The application informs which to use: if positive class implies money spending (*e.g.* loan prediction), then our strategy implies spending more money; while the latter aims to reduce money lent to achieve fairness.

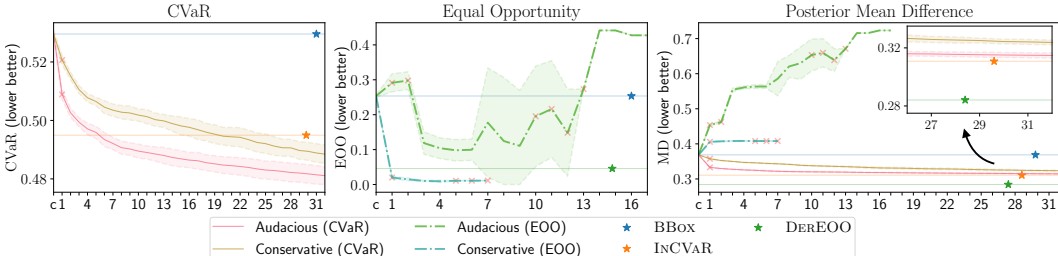

Figure 5: **ACS 2015 with Binary Sensitive Attribute and Random Forest Black-box**: Evaluation of TOPDOWN over boosting iterations (x-axis) for different fairness criteria. 'c' on the x-axis denotes the clipped black-box. '$\times$' denote when a subgroup's $\alpha$-tree is initiated (over any fold). The shade denotes $\pm$ a standard deviation from the mean, this disappears when folds have early stopping.

S = s, the corresponding sub-$\alpha$-tree $\Upsilon_s$ can be examined to scrutinize the correction done for the corresponding subgroup. If the splits of the $\alpha$-tree are simple, similarly to standard decision tree classifiers, corresponding partitions of the input domain can be examined. Furthermore, the type of corrections can also be examined, as discussed in Section 3, where corrections can be classified as "sharpening", "dampening", or "polarity flipping" depending on the leaves' $\alpha$-values.

**Usage of sensitive attribute:** Post-processing methods have been flagged in the context of fair classification for the fact that they require explicit access to the sensitive feature at classification time [35, § 6.2.3]. Our basic approach to the induction of $\alpha$-trees falls in this category (Fig. 3), but there is a simple way to *mask* the use of the sensitive attribute and the polarity of disparate treatment it induces: it consists in first inducing a decision tree to *predict* the sensitive feature based on the other features and use this decision tree as an alternative initialization to naively splitting on subgroups. We thus also *redefine* sensitive groups based on this decision tree – thus alleviating the need to use the sensitive attribute in the $\alpha$-tree. The use of *proxy sensitive attributes* in a similar manner has seen ample use in a various domain such as health care [6, 5] and finance [13]. We however note that its application in post-process and $\alpha$-trees may not be appropriate across all domains [8].

## 7 Experiments

To evaluate TOPDOWN[2], we consider three datasets presenting a range of different size / feature types, Bank and German Credit (preprocessed by `AIF360` [4]) and the American Community Survey (ACS) dataset preprocessed by `Folktables`[3] [9]. The SI (pg 32) presents all results at length (including considerations on proxy sensitive attributes, distribution shift, and interpretability), along with the different black-boxes considered (random forests and neural nets). We concentrate in the Section on the ACS dataset for income prediction in the state of CA and evaluate TOPDOWN's application to various fairness criteria (as per Section 6 and SI pg 15) with Random Forests (RF). For these experiments, we consider *age* as a binary sensitive attribute with a bin split at 25 (a trinary modality is deferred to the SI). For the black-box, we consider a clipped (Assumption 1 with $B = 1$) random forest (RF) from `scikit-learn` calibrated using Platt's method [23]. The RF consists of an ensemble of 50 decision trees with a maximum depth of 4 and a random selection of 10% of the training samples per decision tree. Data is split into 3 subsets for black-box training, post-processing training, and testing; consisting of 40:40:20 splits in 5 fold cross validation. For EOO, we utilize an out-of-the-box Gaussian Naive Bayes classifier from `scikit-learn` to approximate $\eta^\star$.

**Multiple fairness criteria** We evaluate TOPDOWN for CVAR, equality of opportunity EOO, and statistical parity SP. The complete treatment of SP is pushed to SI (Sections II, XII). SP aims to make subgroup's expected posteriors similar and is popular in a various post-processing methods [31, 1]. The definition can be found in SI (pg 15) along with the strategy used in TOPDOWN. Conservative and audacious updates rules are also tested. For each of these TOPDOWN configurations, we boost for 32 iterations. The initial $\alpha$-tree is initialized as in Fig. 3.

We compare against 5 baseline approaches. For CVAR we consider the in-processing approach (INCVAR) presented in [32]. For EOO, we consider a derived predictor (DEREOO) [14]. Our

---

[2]Implementation public at: `https://github.com/alexandersoen/alpha-tree-fair-wrappers`
[3]Public at: `https://github.com/zykls/folktables`

SP baselines include an optimized score transformation approach (OST) [31]; a derived predictor modified for SP (DERSP) [14]; and a randomized threshold optimizer approach (RTO) [1]. We denote the clipped black-box as BBOX. The experiments for CVAR and EOO are summarized in Fig. 5; the full plot with SP is presented at SI Fig. 10. For clarity we only plot the baselines and wrappers which are directly associated to each fairness criteria. We also plot the posterior mean difference between the data and debiased posteriors MD (0/1 loss) to examine the effects on accuracy.

**For CVAR**, both conservative and audacious approaches decreases CVAR, which results in better CVAR values than both the original BBOX and in-processing baseline INCVAR – which is good news since INCVAR directly optimizes CVAR. We note that there are cases in which the in-processing approach is better than ours (trinary sensitive attributes in SI), but this is expected given INCVAR's optimization goal. Interestingly, the audacious update is superior in both CVAR and MD than the conservative update. This is also consistent for trinary sensitive attributes. Thus, the audacious update is desirable when optimizing CVAR. Another observation is that only one sensitive attribute subgroup's $\alpha$-tree is initialized (only one '×'). This indicates that after 32 iterations the worse case subgroup does not change in the binary case.

**For EOO**, there is a huge difference between conservative and audacious updates as the former gets to the most fair outcomes of all baselines. Even if we used early stopping or pruning of the $\alpha$-tree (taking an earlier iterations) the audacious update would fail at producing outcomes as fair as its conservative counterpart. Furthermore, the audacious update comes with a significant degradation of accuracy MD. Furthermore, by looking at the iterations in which subgroup $\alpha$-trees are initialized, the audacious update causes large (primarily bad) jumps in performance. This rejoins our remark on the interest of having a conservative update in Section 5. When compared to DEREOO, we find that the conservative TOPDOWN approach produces lower EOO. However, DEREOO tend to have better accuracy scores in MD. These observations are consistent with the trinary sensitive attribute (SI).

**For SP**, we can observe fairness results that can be on par with contenders for the conservative update, but observe a substantial degradation of MD. This, we believe, follows from a simple plug-in instantiation of $M, \eta_t$ for the fairness notion in SI Section II, resulting in potentially harsh updates. In SI (pg 15), we discuss an alternative approach using ties with optimal transport.

## 8 Limitations and Conclusion

Given the context of fairness, it is important to highlight possible limitations of our approach and the potential social harm from such misuse. We highlight two of these for our TOPDOWN approach. Firstly, our approach has shown to have failure cases in *small data cases*. This can be seen in the experiments on German Credit dataset (SI, Section XIII-XV). This, we believe, has a formal basis in our approach. For instances, EOO requires accurate data posterior estimation which may be difficult in small data regimes. Secondly, TOPDOWN is of course not unilaterally better than all other post-processing fairness approaches – there is *No Free Lunch*. As such, we do not claim that our instantiation of TOPDOWN is optimal in CVAR, EOO, or SP. However, considering that such different fairness models instantiated in the *same* algorithm can lead to competitive results with the respective state of the art, the avenue for improved instantiations or accurate extensions to new fairness constraints appears promising. We leave these for future work.

## Acknowledgments and Disclosure of Funding

YM received funding from the European Research Council (ERC) under the European Union's Horizon 2020 research and innovation program (grant agreement No. 882396), the Israel Science Foundation(grant number 993/17), Tel Aviv University Center for AI and Data Science (TAD), and the Yandex Initiative for Machine Learning at Tel Aviv University. AS and LX thank members of the ANU Humanising Machine Intelligence program for discussions on fairness and ethical concerns in AI, and the NeCTAR Research Cloud for providing computational resources, an Australian research platform supported by the National Collaborative Research Infrastructure Strategy.

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
