# Supplementary Material

## Abstract

This is the Supplementary Material to Paper "Fair Wrapping for Black-box Predictions". To differentiate with the numberings in the main file, the numbering of Theorems is letter-based (A, B, ...).

## Table of contents

# I    Additional Equality of Opportunity Details

In this section, we present additional details for the equality of opportunity (EOO) strategy presented in Section 6. In particular, we present the full definition of the $(p, \delta)$-*pushup* of $\eta^\star$.

**Definition A.** *Fix $p \in [0, 1]$ and let $\mathcal{X}_p$ be a subset of $\mathcal{X}$ such that (i) $\inf \eta^\star(\mathcal{X}_p) \geq \sup \eta^\star(\mathcal{X} \backslash \mathcal{X}_p)$ and (ii) $\int_{\mathcal{X}_p} \mathrm{dM} = p$. For any $\delta \geq 0$, the $(p, \delta)$-pushup of $\eta^\star$, $\eta^\star_{p,\delta}$, is the posterior defined as $\eta^\star_{p,\delta} = \eta^\star$ if $\inf \eta^\star(\mathcal{X}_p) \geq 1/2$ and otherwise:*

$$\eta^\star_{p,\delta}(\boldsymbol{x}) \quad \dot{=} \quad \begin{cases} \frac{1}{2} + \delta & \textit{if } \eta^\star(\boldsymbol{x}) \in [\inf \eta^\star(\mathcal{X}_p), 1/2 + \delta] \\ \eta^\star(\boldsymbol{x}) & \textit{otherwise.} \end{cases} \tag{17}$$

Following Fig. 4 (right) and discussion in the main-text, we make a couple more observation. From Def. A, the selection of the set $\mathcal{X}_p$ (*i.e.* choice of $p$) presents a tradeoff between accuracy and the fairness objective. As $p$ increases, the size of $\mathcal{X}_p$ necessarily increases. For instance, taking $\mathcal{X}_p = \{\boldsymbol{x} : \eta^\star(\boldsymbol{x}) \in > l\}$, the larger $\mathcal{X}_p$ is the more negative prediction are flipped. Thus intuitively, $p$ measures the size of correct posterior values and $\delta$ defines the size of the correction.

# II    Handling Statistical parity

**Statistical parity** (SP) is a group fairness notion [10], implemented recently in a context similar to ours [1] as the constraint that per-group expected treatments must not be too far from each other. We say that $\eta_{\mathrm{f}}$ achieves $\varepsilon$-statistical parity (across all groups induced by sensitive attribute $S$) iff

$$\max_{s \in S} \mathbb{E}_{\mathsf{X} \sim \mathrm{M}_s} [\eta_{\mathrm{f}}(\mathsf{X})] - \min_{s \in S} \mathbb{E}_{\mathsf{X} \sim \mathrm{M}_s} [\eta_{\mathrm{f}}(\mathsf{X})] \quad \leq \quad \varepsilon. \tag{18}$$

Denote $s^\circ \dot{=} \arg\min_{s \in S} \mathbb{E}_{\mathsf{X} \sim \mathrm{M}_s} [\eta_{\mathrm{f}}(\mathsf{X})], s^* \dot{=} \arg\max_{s \in S} \mathbb{E}_{\mathsf{X} \sim \mathrm{M}_s} [\eta_{\mathrm{f}}(\mathsf{X})]$. Since the risk we minimize in (2) involves a proper loss, the most straightforward use of TOPDOWN is to train the sub-$\alpha$-tree for one of these two groups, giving as target posterior the *expected* posterior of the other group, *i.e.* we use $\eta_{\mathrm{t}}(\boldsymbol{x}) = \mathbb{E}_{\mathsf{X} \sim \mathrm{M}_{s^*}} [\eta_{\mathrm{u}}(\mathsf{X})] \dot{=} \overline{\eta}_{\mathrm{u}\,s^*}$ if we grow the $\alpha$-tree of $s^\circ$ and thus iterate

> TOPDOWN with $\mathrm{M_t} \leftarrow \mathrm{M}_{s^\circ}$ and $\eta_{\mathrm{t}} \leftarrow \overline{\eta}_{\mathrm{u}\,s^*}$,

and we repeat until $s^\circ$ does not achieve anymore the smallest expected posterior. We then update the group and repeat the procedure until a given slack $\varepsilon$ is achieved between the extremes in (18). More sophisticated / gentle approaches are possible, including using the links between statistical parity and optimal transport (OT, [10, Section 3.2]), suggesting to use as target posterior the expected posterior obtained from an OT plan between groups $s^\circ$ and $s^*$.

# III    Weak Hypothesis Assumption Discussion

In this Section, we discuss the balanced distribution introduced in Definition 4 and the Weak Hypothesis Assumption (WHA) introduced in Assumption 2. In particular, we note that these definitions are a generalization of those introduced in "classical" decision tree boosting [15]. The primary difference in our case and the "classical" case, is that we must consider the posterior values of a base black-box $\eta_{\mathrm{u}}$, whereas the usual top-down tree induction only cares about minimizing its loss function to produce an accurate $\eta_{\mathrm{t}}$. In-fact, we can even interpret our variants as a case where we replace the labels in a classification task $\mathsf{Y}$ with "labels" (in $[0, 1]$) of how well another classifier does in classification $\mathsf{Y}\tilde{\iota}(\eta_{\mathrm{u}}(\mathsf{X}))$. We will explore various cases to show that our definitions reduce to the classical case; and further provide settings to intuitive describe our more complicated assumption.

**The balanced distribution** in Definition 4 acts as a re-weighting mechanism for particular samples of the original distribution $\mathrm{D}_{\mathrm{t}\lambda}(z)$. Let us first consider the reduction to the balanced distribution introduced by [15]. This comes from the fairness-free case, where we ignore the confidence of the black-box $\eta_{\mathrm{u}}$ and set $\tilde{\iota}(\eta_{\mathrm{u}}) \leftarrow 1$. This provides two simplifications. Firstly, the edge value becomes $\mathrm{e}(\mathrm{M}_\lambda, \eta_{\mathrm{t}}) = \mathbb{E}_{(\mathsf{X},\mathsf{Y}) \sim \mathrm{D}_{\mathrm{t}\lambda}} [\mathsf{Y}] = 2q_\lambda - 1$, where $q_\lambda$ the local proportion of positive examples in $\lambda$. That is, we have $q_\lambda = \mathbb{P}(\mathsf{Y} = +1 | \mathsf{X}$ at leaf $\lambda)$. The second simplification is in the numerator of

Equation 13. In summary, we get:

$$D'_{t\lambda}(z) \leftarrow \frac{1 - e(M_\lambda, \eta_t) \cdot y}{1 - e(M_\lambda, \eta_t)^2} \cdot D_{t\lambda}(z) = \frac{1 - y \cdot (2q_\lambda - 1)}{4q_\lambda(1 - q_\lambda)} \cdot D_{t\lambda}(z) = D_{t\lambda}(z) \cdot \begin{cases} \frac{1}{2q_\lambda} & \text{if } y = +1 \\ \frac{1}{2(1 - q_\lambda)} & \text{if } y = -1 \end{cases},$$
(19)

which is indeed the balanced distribution of [15]. Intuitively, this balanced distribution simply ensures that regardless of $\boldsymbol{x}$, it is equally likely for $y$ be either $-1$ or $+1$ (hence balanced in its prediction value $Y$). This can be seen by calculating the edge with the new distribution $\mathbb{E}_{(X,Y) \sim D'_{t\lambda}}[Y] = 0$.

To consider our balanced distribution, let us start with (19). In particular, we will replace the labels $Y \in \{-1, +1\}$ by the predictive ability of black-box $\eta_u$. This predictive ability is summarized by $Y\tilde{\iota}(\eta_u(X)) \in [-1, 1]$, where larger is better. In particular, taking from the boosting jargon, the *confidences* ($|\tilde{\iota}|$, [27]) of the black-box $\eta_u$ are considered. For instance, a highly confident $|\tilde{\iota}(\eta_u(X))| \to 1$ and correct $\text{sign}(Y) = \text{sign}(\iota(\eta_u(X)))$ prediction will lead to a value close to $+1$ (positive label). If the prediction is confident but incorrect $\text{sign}(Y) \neq \text{sign}(\iota(\eta_u(X)))$, then the value will be close to $-1$ (negative label). If the prediction is not confident $|\tilde{\iota}(\eta_u(X))| \to 0$, then we get a neutral response 0.

With this in mind, the balanced distribution in Definition 4 is "balanced" in the prediction score $Y\tilde{\iota}(\eta_u(X))$. Of course, the distribution $D'_{t\lambda}(z)$ is with respect to $\mathcal{X} \times \mathcal{Y}$ and not $\mathcal{X} \times [-1, 1]$, and thus we are not re-balancing for each possible $Y\tilde{\iota}(\eta_u(X)) \in [-1, 1]$. The balance of the distribution takes into account the confidence of the predictions, this can be seen in examination of the balanced edge in this setting:

$$\mathbb{E}_{(X,Y) \sim D'_{t\lambda}}[Y \cdot \tilde{\iota}(\eta_u(X))] = \frac{1}{1 - e(M_\lambda, \eta_t)^2} \cdot e(M_\lambda, \eta_t) \cdot \mathbb{E}_{(X,Y) \sim D_{t\lambda}}[1 - \tilde{\iota}^2(\eta_u(X))].$$
(20)

(20) is in general non-zero. However, in the case that the $\eta_u$ is confident such that $\tilde{\iota}(\eta_u(X)) \in \{-1, +1\}$, then the new edge goes to 0. In other words, when we have a maximally confident black-box, the balanced edge will be 0; otherwise, the edge is scaled with respect to a function of the confidence $\mathbb{E}_{(X,Y) \sim D_{t\lambda}}[1 - \tilde{\iota}^2(\eta_u(X))]$.

**Our Weak Learning Assumption (WHA)** in Assumption 2 has a similar analysis as that of the balanced distribution. Condition (i) of the assumption is exactly the same $\gamma$-witness condition in [15]. Condition (ii), similar to the analysis of (20), vanishes when we have a maximally confident black-box $\eta_u$. In general, condition (ii) ensures that one side of the split induced by $h$ are skewed to one side.

Let us exemplify how our WHA works if we have a leaf $\lambda$ where local "treatments due to the black-box" are bad ($y\tilde{\iota}(\eta_u(\boldsymbol{x})) < 0$ often). In such a case, $e(M_\lambda, \eta_t) < 0$ so the balanced distribution (Definition 4) reweights higher examples whose treatment is better than average, *i.e.* the local minority. Suppose (i) holds as is without the $|.|$. In such a case, the split "aligns" the treatment quality with $h$, so $h = +1$ for a substantial part of this minority. (ii) imposes $\mathbb{E}_{(X,Y) \sim D_{t\lambda}}[h(X)] \geq \mathbb{E}_{(X,Y) \sim D_{t\lambda}}[\tilde{\iota}^2(\eta_u(X)) \cdot h(X)]$: $h = -1$ for a substantial part of large confidence treatment. The split thus tends to separate mostly large confidence but bad treatments (left) and mostly good treatments (right). Before the split, the value $\Upsilon(\lambda)$ would be negative (5) and thus reverse the polarity of the black-box, which would be good for badly treated examples but catastrophic for the local minority of adequately treated examples. After the split however, we still have the left ($h = -1$) leaf where this would eventually happen, but the minority at $\lambda$ would have disproportionately ended in the right ($h = +1$) leaf, where it would be likely that $\Upsilon(.)$ would this time be *positive* and thus preserve the polarity of the treatment of the black-box.

**Why are the assumptions states using the balanced distribution?** A point worth clarifying is the "why's" of consider a WHA in the way it is formulated and the focus on the balanced distribution. In typical boosting algorithms, a weak learning hypothesis is needed: the assumption that a classifier / split / weak learner exists for *any* distribution which is better than random. In top-down tree induction, it is in-fact sufficient to only have this assumption hold for one type of distribution — the balanced distribution [15]. As such the WHA presented, is actually a weaker assumption than the "distributionally global" weak learning assumption used in other boosting algorithms.

## IV    General KL Distortion

We introduce a general Theorem for upperbounding the distortion an $\alpha$-correction can make, as promised in Section 5. Notably, the following result does not rely on any setting.

**Theorem A.** *For any function $\alpha : \mathfrak{X} \to \mathbb{R}$, any black-box posterior $\eta_{\mathrm{u}}$, and any integer $K \geq 2$, using (3) yields the following bound on the KL divergence:*

$$\mathrm{KL}(\eta_{\mathrm{u}}, \eta_{\mathrm{f}}; \mathrm{M}) \leq \mathbb{E}_{\mathsf{X} \sim \mathrm{M}} \left[ \sum_{k=2}^{K} \frac{\eta_{\mathrm{u}}(\mathsf{X})(1 - \eta_{\mathrm{u}}(\mathsf{X})) f^k(\alpha(\mathsf{X}), \eta_{\mathrm{u}}(\mathsf{X}))}{k(k-1)} \right] + o\left( \mathbb{E}_{\mathsf{X} \sim \mathrm{M}} \left[ (\alpha(\mathsf{X}) - 1)^K \right] \right),$$

(21)

*where we have used function $f : \mathbb{R} \times [0, 1] \to \mathbb{R}$ defined as:*
$$f(z, u) \quad \doteq \quad |\log\left(u/(1-u)\right) \cdot (1-z)| = |\iota(u) \cdot (1-z)|.$$

(22)

The proof is presented in Section V. In general, one can see that if $\alpha(\cdot)$ does not differ from $1$ too much, the KL divergence will be small. This intuitively makes sense as $\alpha = 1$ causes $\eta_{\mathrm{f}} = \eta_{\mathrm{u}}$ — there is no untwisting. Theorem A can be further weakened to provide easier to understand bounds, as per Corollary 2.

We further present an alternative setting to that in the main-text, which provides another bound on distortion. In this setting **(S2)**, we ensure that when the predictions of $\eta_{\mathrm{u}}$ are confident, then the corresponding $\alpha$ correction are small (close to $\alpha = 1$).

**(S2)** $f(\alpha(\boldsymbol{x}), \eta_{\mathrm{u}}(\boldsymbol{x})) = |\iota(\eta_{\mathrm{u}}(\boldsymbol{x})) \cdot (1 - \alpha(\boldsymbol{x}))| \leq 1$ (a.s.) , $f$ being in (22).

This provides the alternative upperbound on the distortion.

**Corollary B.** *Under setting (S2), we have the upperbound*

$$\mathrm{KL}(\eta_{\mathrm{u}}, \eta_{\mathrm{f}}; \mathrm{M}) \quad \leq \quad \pi^2/24 \approx 0.41.$$

(23)

The proof of the Corollary is in SI, Section VI and includes a graphical view of the values of $\eta_{\mathrm{u}}(.)$ and $\alpha(.)$ complying with **(S2)**.

## V    Proof of Theorem A

We first show two technical Lemmata.

**Lemma A.** *For any $a \geq 0$, let*

$$h(z) \quad \doteq \quad \log\left(\frac{1}{1 + a^{1+z}}\right).$$

(24)

*We have*

$$h^{(k)}(z) \quad = \quad -\frac{\log^k(a) \cdot a^{1+z}}{(1 + a^{1+z})^k} \cdot P_{k-1}(a^{1+z}),$$

(25)

*where $P_k(x)$ is a degree-$k-1$ polynomial. Letting $c_{k,j}$ the constant factor of monomial $x^j$ in $P_k(x)$, for $j \leq k-1$, we have the following recursive definitions: $c_{1,0} = 1$ ($k = 1$) and*

$$c_{k+1,k} \quad = \quad (-1)^k,$$ (26)
$$c_{k+1,j} \quad = \quad (j+1) \cdot c_{k,j} - (k+1-j) \cdot c_{k,j-1}, \forall 0 < j < k,$$ (27)
$$c_{k+1,0} \quad = \quad 1.$$ (28)

*Hence, we have for example $P_1(x) = 1, P_2(x) = -x + 1, P_3(x) = x^2 - 4x + 1, P_4(x) = -x^3 + 11x^2 - 11x + 1, \dots$*

**Proof:** We let

$$f(z) \quad \doteq \quad \frac{a^{1+z}}{1 + a^{1+z}},$$

(29)

so that $h'(z) = -\log(a) \cdot g(z)$ and we show

$$f^{(k)}(z) \;=\; \frac{\log^k(a) \cdot a^{1+z}}{(1 + a^{1+z})^{k+1}} \cdot P_k(a^{1+z}). \tag{30}$$

We first check

$$f'(z) \;=\; \frac{\log(a) \cdot a^{1+z}}{(1 + a^{1+z})^2}, \tag{31}$$

which shows $P_1(x) = 1$. We then note that for any $k \in \mathbb{N}_*$,

$$\frac{\mathrm{d}}{\mathrm{d}z} \frac{a^{1+z}}{(1 + a^{1+z})^k} \;=\; \frac{\log(a) \cdot a^{1+z}}{(1 + a^{1+z})^{k+1}} \cdot (-(k-1)a^{1+z} + 1), \tag{32}$$

so the induction case yields $f^{(k+1)}(z) \doteq {f^{(k)}}'(z)$, that is:

$$f^{(k+1)}(z)$$

$$= \;\; \log^k(a) \cdot \frac{\mathrm{d}}{\mathrm{d}z}\left( \frac{a^{1+z}}{(1 + a^{1+z})^{k+1}} \cdot P_k(a^{1+z}) \right)$$

$$= \;\; \log^k(a) \cdot \left( \frac{\log(a) \cdot a^{1+z}}{(1 + a^{1+z})^{k+2}} \cdot (-ka^{1+z} + 1) \cdot P_k(a^{1+z}) + \frac{a^{1+z} \cdot \log(a)}{(1 + a^{1+z})^{k+1}} \cdot a^{1+z} \cdot \left.\frac{\mathrm{d}P_k(x)}{\mathrm{d}x}\right|_{x=a^{1+z}} \right)$$

$$= \;\; \frac{\log^{k+1}(a) \cdot a^{1+z}}{(1 + a^{1+z})^{k+2}} \cdot \underbrace{\left( (-ka^{1+z} + 1) \cdot P_k(a^{1+z}) + a^{1+z}(1 + a^{1+z}) \cdot \left.\frac{\mathrm{d}P_k(x)}{\mathrm{d}x}\right|_{x=a^{1+z}} \right)}_{\doteq P_{k+1}(a^{1+z})}, \tag{33}$$

from which we check that $P_{k+1}$ is indeed a polynomial and its coefficients are obtained via identification from $P_k$, which establishes (30) and yields to the statement of the Lemma. ☐

**Lemma B.** *Coefficient $c_{k,j}$ admits the following bound, for any $0 \le j \le k$:*

$$|c_{k,j}| \;\le\; (k-1)! \binom{k-1}{j}. \tag{34}$$

**Proof:** First, we have the following recursive definition for the absolute value of the leveraging coefficients in $c_{.,.}$ (we call them $a_{.,.}$ for short): $|c_{.,.}| = a_{.,.}$ with

$$a_{k+1,k} \;=\; 1, \tag{35}$$
$$a_{k+1,j} \;=\; (j+1) \cdot a_{k,j} + (k+1-j) \cdot a_{k,j-1}, \forall 0 < j < k, \tag{36}$$
$$a_{k+1,0} \;=\; 1. \tag{37}$$

We now show by induction that $a_{k+1,j} \le k!\binom{k}{j} \doteq b_{k+1,j}$. For $j = 0$, $b_{k+1,0} = k! \ge a_{k+1,0}$ ($k \ge 2$) and for $j = k$, $b_{k+1,k} = k! \ge a_{k+1,0}$ as well. We now check, assuming the property holds at all ranks $k$, that for ranks $k + 1$, we have

$$a_{k+1,j} \;=\; (j+1) \cdot a_{k,j} + (k+1-j) \cdot a_{k,j-1}$$

$$\le \;\; (j+1)(k-1)!\binom{k-1}{j} + (k+1-j)(k-1)!\binom{k-1}{j-1}, \tag{38}$$

and we want to check that the RHS is $\le k!\binom{k}{j}$ for any $0 < j < k$. Simplifying yields the equivalent inequality

$$(j+1)(k-j) + (k+1-j)j \;\le\; k^2. \tag{39}$$

finding the worst case bound for $j$ yields $j = k/2$ (we disregard the fact that $j$ is an integer) and plugging in the bound yields the constraint on $k$: $k \ge 2$, which indeed holds. ☐

We also check that $h$ in Lemma A is infinitely differentiable. As a consequence, we get from Lemma A the Taylor expansion around $g = 1$ (for any $a \ge 0$) at any order $K \ge 2$,

$$\log\left( \frac{1}{1 + a^g} \right)$$

$$= \;\; \log\left( \frac{1}{1 + a} \right) - \frac{a \log a}{1 + a} \cdot (g - 1) - \underbrace{\sum_{k=2}^{K} \frac{a \log^k(a) P_{k-1}(a)}{k!(1+a)^k} \cdot (g - 1)^k}_{\doteq R_{K,a}(g)} + o((g-1)^K) \tag{40}$$

The choice to start the summation at $k = 2$ is done for technical simplifications to come. We thus have

$$
\begin{aligned}
\log \eta_f(\boldsymbol{x}) &= \log \left( \frac{1}{1 + \left( \frac{1 - \eta_u(\boldsymbol{x})}{\eta_u(\boldsymbol{x})} \right)^{\alpha(\boldsymbol{x})}} \right) \\
&= \log \eta_u(\boldsymbol{x}) - (1 - \eta_u(\boldsymbol{x})) \log \left( \frac{1 - \eta_u(\boldsymbol{x})}{\eta_u(\boldsymbol{x})} \right) \cdot (\alpha(\boldsymbol{x}) - 1) - R_{\frac{1 - \eta_u(\boldsymbol{x})}{\eta_u(\boldsymbol{x})}, K}(\alpha(\boldsymbol{x})) \\
&\quad + o((\alpha(\boldsymbol{x}) - 1)^K), \\
\log(1 - \eta_f(\boldsymbol{x})) &= \log(1 - \eta_u(\boldsymbol{x})) - \eta_u(\boldsymbol{x}) \log \left( \frac{\eta_u(\boldsymbol{x})}{1 - \eta_u(\boldsymbol{x})} \right) \cdot (\alpha(\boldsymbol{x}) - 1) - R_{\frac{\eta_u(\boldsymbol{x})}{1 - \eta_u(\boldsymbol{x})}, K}(\alpha(\boldsymbol{x})) \\
&\quad + o((\alpha(\boldsymbol{x}) - 1)^K).
\end{aligned}
$$

Define for short $\Delta_u(\boldsymbol{x}) \doteq \eta_u(\boldsymbol{x}) \cdot - \log \eta_f(\boldsymbol{x}) + (1 - \eta_u(\boldsymbol{x})) \cdot - \log(1 - \eta_f(\boldsymbol{x})) - (\eta_u(\boldsymbol{x}) \cdot - \log \eta_u(\boldsymbol{x}) + (1 - \eta_u(\boldsymbol{x})) \cdot - \log(1 - \eta_u(\boldsymbol{x})))$, so that $\mathrm{KL}(\eta_u, \eta_f; M) = \mathbb{E}_{X \sim M}[\Delta_u(X)]$. The Taylor expansion (40) unveils an interesting simplification:

$$
\begin{aligned}
\Delta_u(\boldsymbol{x}) &= -\eta_u(\boldsymbol{x}) \log \eta_u(\boldsymbol{x}) + \eta_u(\boldsymbol{x})(1 - \eta_u(\boldsymbol{x})) \log \left( \frac{1 - \eta_u(\boldsymbol{x})}{\eta_u(\boldsymbol{x})} \right) \cdot (\alpha(\boldsymbol{x}) - 1) \\
&\quad + \eta_u(\boldsymbol{x}) \cdot R_{\frac{1 - \eta_u(\boldsymbol{x})}{\eta_u(\boldsymbol{x})}, K}(\alpha(\boldsymbol{x})) \\
&\quad - (1 - \eta_u(\boldsymbol{x})) \log(1 - \eta_u(\boldsymbol{x})) + (1 - \eta_u(\boldsymbol{x})) \eta_u(\boldsymbol{x}) \log \left( \frac{\eta_u(\boldsymbol{x})}{1 - \eta_u(\boldsymbol{x})} \right) \cdot (\alpha(\boldsymbol{x}) - 1) \\
&\quad + (1 - \eta_u(\boldsymbol{x})) \cdot R_{\frac{\eta_u(\boldsymbol{x})}{1 - \eta_u(\boldsymbol{x})}, K}(\alpha(\boldsymbol{x})) \\
&\quad - (\eta_u(\boldsymbol{x}) \cdot - \log \eta_u(\boldsymbol{x}) + (1 - \eta_u(\boldsymbol{x})) \cdot - \log(1 - \eta_u(\boldsymbol{x}))) + o((\alpha(\boldsymbol{x}) - 1)^K) \\
&= \eta_u(\boldsymbol{x}) \cdot R_{\frac{1 - \eta_u(\boldsymbol{x})}{\eta_u(\boldsymbol{x})}}(\alpha(\boldsymbol{x})) + (1 - \eta_u(\boldsymbol{x})) \cdot R_{\frac{\eta_u(\boldsymbol{x})}{1 - \eta_u(\boldsymbol{x})}, K}(\alpha(\boldsymbol{x})) + o((\alpha(\boldsymbol{x}) - 1)^K), \forall \boldsymbol{x} \in \mathcal{X},
\end{aligned}
$$

so the divergence to the black-box prediction simplifies as well, this time using Lemma B:

$$
\begin{aligned}
\mathrm{KL}(\eta_u, \eta_f; M) &= \mathbb{E}_{X \sim M} \left[ \eta_u(X) \cdot R_{\frac{1 - \eta_u(X)}{\eta_u(X)}, K}(\alpha(X)) + (1 - \eta_u(X)) \cdot R_{\frac{\eta_u(X)}{1 - \eta_u(X)}, K}(\alpha(X)) \right] \\
&\quad + o \left( \mathbb{E}_{X \sim M} \left[ (\alpha(X) - 1)^K \right] \right).
\end{aligned}
\tag{41}
$$

Not touching the little-oh term, we simplify further and bound the term in the expectation: for any $\boldsymbol{x} \in \mathcal{X}$,

$$\eta_{\mathrm{u}}(\boldsymbol{x}) \cdot R_{\frac{1-\eta_{\mathrm{u}}(\boldsymbol{x})}{\eta_{\mathrm{u}}(\boldsymbol{x})}, K}(\alpha(\boldsymbol{x})) + (1 - \eta_{\mathrm{u}}(\boldsymbol{x})) \cdot R_{\frac{\eta_{\mathrm{u}}(\boldsymbol{x})}{1-\eta_{\mathrm{u}}(\boldsymbol{x})}, K}(\alpha(\boldsymbol{x}))$$

$$= \eta_{\mathrm{u}}(\boldsymbol{x}) \cdot \sum_{k=2}^{K} \frac{\frac{1-\eta_{\mathrm{u}}(\boldsymbol{x})}{\eta_{\mathrm{u}}(\boldsymbol{x})} \cdot \log^{k}\left(\frac{1-\eta_{\mathrm{u}}(\boldsymbol{x})}{\eta_{\mathrm{u}}(\boldsymbol{x})}\right) P_{k-1}\left(\frac{1-\eta_{\mathrm{u}}(\boldsymbol{x})}{\eta_{\mathrm{u}}(\boldsymbol{x})}\right)}{k! \left(1 + \frac{1-\eta_{\mathrm{u}}(\boldsymbol{x})}{\eta_{\mathrm{u}}(\boldsymbol{x})}\right)^{k}} \cdot (\alpha(\boldsymbol{x}) - 1)^{k}$$

$$+ (1 - \eta_{\mathrm{u}}(\boldsymbol{x})) \cdot \sum_{k=2}^{K} \frac{\frac{\eta_{\mathrm{u}}(\boldsymbol{x})}{1-\eta_{\mathrm{u}}(\boldsymbol{x})} \cdot \log^{k}\left(\frac{\eta_{\mathrm{u}}(\boldsymbol{x})}{1-\eta_{\mathrm{u}}(\boldsymbol{x})}\right) P_{k-1}\left(\frac{\eta_{\mathrm{u}}(\boldsymbol{x})}{1-\eta_{\mathrm{u}}(\boldsymbol{x})}\right)}{k! \left(1 + \frac{\eta_{\mathrm{u}}(\boldsymbol{x})}{1-\eta_{\mathrm{u}}(\boldsymbol{x})}\right)^{k}} \cdot (\alpha(\boldsymbol{x}) - 1)^{k}$$

$$= \eta_{\mathrm{u}}(\boldsymbol{x}) \cdot \sum_{k=2}^{K} \frac{\frac{1-\eta_{\mathrm{u}}(\boldsymbol{x})}{\eta_{\mathrm{u}}(\boldsymbol{x})} \cdot \log^{k}\left(\frac{1-\eta_{\mathrm{u}}(\boldsymbol{x})}{\eta_{\mathrm{u}}(\boldsymbol{x})}\right) \cdot \sum_{j=0}^{k-2} c_{k-1,j}\left(\frac{1-\eta_{\mathrm{u}}(\boldsymbol{x})}{\eta_{\mathrm{u}}(\boldsymbol{x})}\right)^{j}}{k! \left(1 + \frac{1-\eta_{\mathrm{u}}(\boldsymbol{x})}{\eta_{\mathrm{u}}(\boldsymbol{x})}\right)^{k}} \cdot (\alpha(\boldsymbol{x}) - 1)^{k}$$

$$+ (1 - \eta_{\mathrm{u}}(\boldsymbol{x})) \cdot \sum_{k=2}^{K} \frac{\frac{\eta_{\mathrm{u}}(\boldsymbol{x})}{1-\eta_{\mathrm{u}}(\boldsymbol{x})} \cdot \log^{k}\left(\frac{\eta_{\mathrm{u}}(\boldsymbol{x})}{1-\eta_{\mathrm{u}}(\boldsymbol{x})}\right) \cdot \sum_{j=0}^{k-2} c_{k-1,j}\left(\frac{\eta_{\mathrm{u}}(\boldsymbol{x})}{1-\eta_{\mathrm{u}}(\boldsymbol{x})}\right)^{j}}{k! \left(1 + \frac{\eta_{\mathrm{u}}(\boldsymbol{x})}{1-\eta_{\mathrm{u}}(\boldsymbol{x})}\right)^{k}} \cdot (\alpha(\boldsymbol{x}) - 1)^{k}$$

$$= \sum_{k=2}^{K} \frac{\log^{k}\left(\frac{1-\eta_{\mathrm{u}}(\boldsymbol{x})}{\eta_{\mathrm{u}}(\boldsymbol{x})}\right) \cdot \sum_{j=0}^{k-2} c_{k-1,j} \cdot \eta_{\mathrm{u}}^{k-j}(\boldsymbol{x})(1 - \eta_{\mathrm{u}}(\boldsymbol{x}))^{j+1}}{k!} \cdot (\alpha(\boldsymbol{x}) - 1)^{k}$$

$$+ \sum_{k=2}^{K} \frac{\log^{k}\left(\frac{\eta_{\mathrm{u}}(\boldsymbol{x})}{1-\eta_{\mathrm{u}}(\boldsymbol{x})}\right) \cdot \sum_{j=0}^{k-2} c_{k-1,j} \cdot (1 - \eta_{\mathrm{u}}(\boldsymbol{x}))^{k-j}\eta_{\mathrm{u}}^{j+1}(\boldsymbol{x})}{k!} \cdot (\alpha(\boldsymbol{x}) - 1)^{k} \qquad (42)$$

We now note, using Lemma B that for any $\boldsymbol{x} \in \mathcal{X}$,

$$\sum_{j=0}^{k-2} |c_{k-1,j}| \cdot \eta_{\mathrm{u}}^{k-j}(\boldsymbol{x})(1 - \eta_{\mathrm{u}}(\boldsymbol{x}))^{j+1}$$

$$= \eta_{\mathrm{u}}^{2}(\boldsymbol{x})(1 - \eta_{\mathrm{u}}(\boldsymbol{x})) \cdot \sum_{j=0}^{k-2} |c_{k-1,j}| \cdot \eta_{\mathrm{u}}^{k-2-j}(\boldsymbol{x})(1 - \eta_{\mathrm{u}}(\boldsymbol{x}))^{j}$$

$$\leq \eta_{\mathrm{u}}^{2}(\boldsymbol{x})(1 - \eta_{\mathrm{u}}(\boldsymbol{x})) \cdot \sum_{j=0}^{k-2} (k-2)! \binom{k-2}{j} \eta_{\mathrm{u}}^{k-2-j}(\boldsymbol{x})(1 - \eta_{\mathrm{u}}(\boldsymbol{x}))^{j}$$

$$= (k-2)! \cdot \eta_{\mathrm{u}}^{2}(\boldsymbol{x})(1 - \eta_{\mathrm{u}}(\boldsymbol{x})) \cdot \underbrace{\sum_{j=0}^{k-2} \binom{k-2}{j} \eta_{\mathrm{u}}^{k-2-j}(\boldsymbol{x})(1 - \eta_{\mathrm{u}}(\boldsymbol{x}))^{j}}_{=(1-\eta_{\mathrm{u}}(\boldsymbol{x})+\eta_{\mathrm{u}}(\boldsymbol{x}))^{k-2}=1}$$

$$= (k-2)! \cdot \eta_{\mathrm{u}}^{2}(\boldsymbol{x})(1 - \eta_{\mathrm{u}}(\boldsymbol{x})),$$

and similarly

$$\sum_{j=0}^{k-2} |c_{k-1,j}| \cdot (1 - \eta_{\mathrm{u}}(\boldsymbol{x}))^{k-j}\eta_{\mathrm{u}}^{j+1}(\boldsymbol{x}) \leq (k-2)! \cdot \eta_{\mathrm{u}}(\boldsymbol{x})(1 - \eta_{\mathrm{u}}(\boldsymbol{x}))^{2},$$

so plugging the two last bounds on (42) yields the bound on $\mathrm{KL}(\eta_u, \eta_f; \mathrm{M})$ from (41):

$$
\begin{aligned}
\mathrm{KL}(\eta_u, \eta_f; \mathrm{M}) \quad \leq \quad & \mathbb{E}_{\mathsf{X} \sim \mathrm{M}} \left[ \sum_{k=2}^{K} \frac{(\eta_u^2(\mathsf{X})(1 - \eta_u(\mathsf{X})) + \eta_u(\mathsf{X})(1 - \eta_u(\mathsf{X}))^2) \left| \log \left( \frac{1 - \eta_u(\mathsf{X})}{\eta_u(\mathsf{X})} \right) \right|^k}{k(k-1)} \cdot |\alpha(\mathsf{X}) - 1|^k \right] \\
& + o \left( \mathbb{E}_{\mathsf{X} \sim \mathrm{M}} \left[ (\alpha(\mathsf{X}) - 1)^K \right] \right) \\
= \quad & \mathbb{E}_{\mathsf{X} \sim \mathrm{M}} \left[ \sum_{k=2}^{K} \frac{\eta_u(\mathsf{X})(1 - \eta_u(\mathsf{X})) \left| \log \left( \frac{1 - \eta_u(\mathsf{X})}{\eta_u(\mathsf{X})} \right) \right|^k}{k(k-1)} \cdot |\alpha(\mathsf{X}) - 1|^k \right] \\
& + o \left( \mathbb{E}_{\mathsf{X} \sim \mathrm{M}} \left[ (\alpha(\mathsf{X}) - 1)^K \right] \right),
\end{aligned}
\tag{43}
$$

which yields the statement of Theorem A.

# VI Proof of Theorem 2 and B

We start by **(S1)**. We study function

$$
f_k(u) \quad \doteq \quad u(1 - u) \left| \log \left( \frac{1 - u}{u} \right) \right|^k, \forall u \in \left[ \frac{1}{1 + \exp(B)}, \frac{1}{1 + \exp(-B)} \right].
\tag{44}
$$

$f_k$ being symmetric around $u = 1/2$ and zeroing in $1/2$, we consider wlog $u < 1/2$ to find its maximum, so we can drop the absolute value. We have

$$
f_k'(u) \quad = \quad \log^{k-1} \left( \frac{1 - u}{u} \right) \cdot \left( (1 - 2u) \cdot \log \left( \frac{1 - u}{u} \right) - k \right).
\tag{45}
$$

Function $u \mapsto (1 - 2u) \cdot \log \left( \frac{1-u}{u} \right)$ is strictly decreasing on $(0, 1/2)$ and has limit $+\infty$ on $0^+$, so the unique maximum of $f$ on $[0, 1/2)$ (we close by continuity the interval in 0 since $\lim_{0^+} f = 0$) is attained at the only solution $u_k$ of

$$
(1 - 2u_k) \cdot \log \left( \frac{1 - u_k}{u_k} \right) \quad = \quad k,
\tag{46}
$$

and such a solution always exist for any $k \ll \infty$. It also follows $u_{k+1} < u_k$, so if we denote as $k^*$ the smallest $k$ such that

$$
u_{k^*} \quad \leq \quad \frac{1}{1 + \exp(B)},
\tag{47}
$$

then we will have the upperbound:

$$
\begin{aligned}
f_k(u) \quad \leq \quad & \frac{1}{1 + \exp(B)} \cdot \frac{1}{1 + \exp(-B)} \cdot B^k \\
= \quad & \frac{B^k}{2 + \exp(B) + \exp(-B)}, \forall k \geq k^*.
\end{aligned}
\tag{48}
$$

We can also compute $k^*$ exactly as it boils down to taking the integer part of the solution of (46) where $u_k$ is picked as in (47):

$$
k^* \quad = \quad \left\lfloor \frac{\exp(B) - 1}{\exp(B) + 1} \cdot B \right\rfloor,
\tag{49}
$$

to get $k^* = 2$, it is sufficient that $B \leq 3$, which thus gives:

$$
\mathrm{KL}(\eta_u, \eta_f; \mathrm{M}) \quad \leq \quad \sum_{k=2}^{K} \frac{\mathbb{E}_{\mathsf{X} \sim \mathrm{M}} \left[ (B \cdot |\alpha(\mathsf{X}) - 1|)^k \right]}{(2 + \exp(B) + \exp(-B)) k(k-1)} + G,
\tag{50}
$$

and if $|\alpha(\boldsymbol{x}) - 1| \leq 1/B = 1/3, \forall \boldsymbol{x} \in \mathcal{X}$, then we can include all terms for all $k \geq 2$ in the upperbound, which makes the little-oh remainder vanish and we get:

$$\text{KL}(\eta_{\text{u}}, \eta_{\text{f}}; \text{M}) \quad \leq \quad \lim_{K \to +\infty} \frac{1}{2 + \exp(B) + \exp(-B)} \cdot \sum_{k=2}^{K} \frac{1}{k(k-1)} \tag{51}$$

$$\leq \quad \frac{1}{2 + \exp(B) + \exp(-B)} \cdot \sum_{k \geq 1} \frac{1}{k^2} \tag{52}$$

$$= \quad \frac{\pi^2}{6(2 + \exp(B) + \exp(-B))}, \tag{53}$$

which is (21) and proves the Corollary for setting **(S1)**. The proof for setting **(S2)** is direct as in this case we get:

$$\text{KL}(\eta_{\text{u}}, \eta_{\text{f}}; \text{M}) \quad \leq \quad \lim_{K \to +\infty} \mathbb{E}_{\mathsf{X} \sim \text{M}} \left[ \sum_{k=2}^{K} \frac{\eta_{\text{u}}(\mathsf{X})(1 - \eta_{\text{u}}(\mathsf{X})) f^k(\alpha(\mathsf{X}), \eta_{\text{u}}(\mathsf{X}))}{k(k-1)} \right]$$

$$= \mathbb{E}_{\mathsf{X} \sim \text{M}} \left[ \sum_{k=2}^{K} \frac{\eta_{\text{u}}(\mathsf{X})(1 - \eta_{\text{u}}(\mathsf{X})) f^k(\alpha(\mathsf{X}), \eta_{\text{u}}(\mathsf{X}))}{k(k-1)} \right]$$

$$\leq \quad \mathbb{E}_{\mathsf{X} \sim \text{M}} \left[ \sum_{k=2}^{K} \frac{\eta_{\text{u}}(\mathsf{X})(1 - \eta_{\text{u}}(\mathsf{X}))}{k(k-1)} \right] \tag{54}$$

$$\leq \quad \frac{1}{4} \cdot \sum_{k=2}^{K} \frac{1}{k(k-1)} \tag{55}$$

$$\leq \quad \frac{1}{4} \cdot \sum_{k \geq 1} \frac{1}{k^2} \tag{56}$$

$$= \quad \frac{\pi^2}{24}, \tag{57}$$

as claimed.

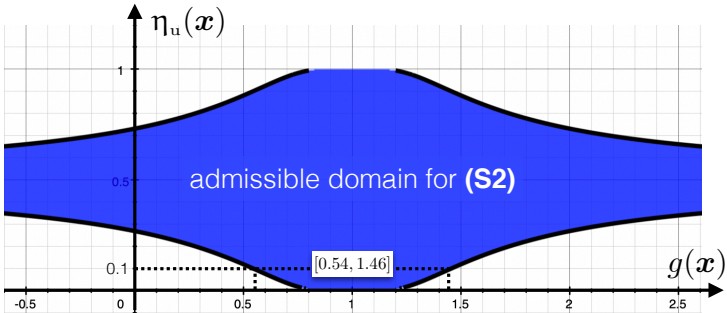

Figure 6: Admissible couples of values $(g, \eta_{\text{u}})$ (in blue) complying with setting **(S2)**. For example, any couple $(g, 0.1)$ with $g \in [0.54, 1.46]$ is admissible.

Figure 6 provides an idea of the set of *admissible* couples (correction, black-box posterior) that comply with **(S2)**, from which we see that the range of admissible corrections is quite flexible, even when $\eta_{\text{u}}$ comes quite close to $\{0, 1\}$.

## VII   Proof of Theorem 1 and 3

We proceed in two steps, first showing that the loss we care about for fairness (2) (main file) is upperbounded by the entropy of the $\alpha$-tree $\Upsilon$, then developing the boosting result from the minimization of the entropy itself. We thus start with the following Theorem.

**Theorem C** (Theorem 1 in Main Paper). *Suppose Assumption 1 holds and the outputs of $\Upsilon$ are:*

$$\Upsilon(\boldsymbol{x}) \;\doteq\; \tilde{\iota}\left(\frac{1 + \mathsf{e}(\mathrm{M}_{\lambda(\boldsymbol{x})}, \eta_{\mathsf{t}})}{2}\right), \forall \boldsymbol{x} \in \mathcal{X}, \tag{58}$$

*where $\lambda(\boldsymbol{x})$ is the leaf reached by $\boldsymbol{x}$ in $\Upsilon$. Then the following bound holds for the risk (2):*

$$L(\eta_{\mathsf{f}}; \mathrm{M}_{\mathsf{t}}, \eta_{\mathsf{t}}) \;\leq\; H(\Upsilon; \mathrm{M}_{\mathsf{t}}, \eta_{\mathsf{t}}). \tag{59}$$

**Proof:** We need a simple Lemma, see *e.g.* [29].

**Lemma C.** $\forall \kappa \in \mathbb{R}, \forall B \geq 0, \forall |z| \leq B,$

$$\log(1 + \exp(\kappa z)) \;\leq\; \log(1 + \exp(\kappa B)) - \kappa \cdot \frac{B - z}{2}. \tag{60}$$

We then note, using $z \doteq \log\left(\frac{1-\eta_{\mathsf{u}}}{\eta_{\mathsf{u}}}\right)$ (stripping variables for readability) and Assumption 1,

$$
\begin{aligned}
-\log \eta_{\mathsf{f}} \;&=\; -\log\left(\frac{\eta_{\mathsf{u}}^{\Upsilon}}{\eta_{\mathsf{u}}^{\Upsilon} + (1 - \eta_{\mathsf{u}})^{\Upsilon}}\right) \\
&=\; -\log\left(\frac{1}{1 + \left(\frac{1-\eta_{\mathsf{u}}}{\eta_{\mathsf{u}}}\right)^{\Upsilon}}\right) \\
&=\; \log\left(1 + \left(\frac{1 - \eta_{\mathsf{u}}}{\eta_{\mathsf{u}}}\right)^{\Upsilon}\right) \\
&=\; \log\left(1 + \exp\left(\Upsilon \log\left(\frac{1 - \eta_{\mathsf{u}}}{\eta_{\mathsf{u}}}\right)\right)\right) \\
&\leq\; \log(1 + \exp(\Upsilon B)) - \Upsilon \cdot \frac{B - \log\left(\frac{1-\eta_{\mathsf{u}}}{\eta_{\mathsf{u}}}\right)}{2} \tag{61} \\
&=\; \log(1 + \exp(\Upsilon B)) - \Upsilon \cdot \frac{B + \iota(\eta_{\mathsf{u}})}{2}, \tag{62}
\end{aligned}
$$

where in (61) we have used (60) with $\kappa \doteq \Upsilon$, using Assumption 1 guaranteeing $|\iota(\eta_{\mathsf{u}})| \leq B$. We also get, using this time $\kappa \doteq -\Upsilon$,

$$
\begin{aligned}
-\log(1 - \eta_{\mathsf{f}}) \;&=\; \log\left(1 + \exp\left(-\Upsilon \log\left(\frac{1 - \eta_{\mathsf{u}}}{\eta_{\mathsf{u}}}\right)\right)\right) \\
&\leq\; \log(1 + \exp(-\Upsilon B)) + \Upsilon \cdot \frac{B + \iota(\eta_{\mathsf{u}})}{2} \\
&=\; \log(1 + \exp(\Upsilon B)) - \Upsilon B + \Upsilon \cdot \frac{B + \iota(\eta_{\mathsf{u}})}{2} \\
&=\; \log(1 + \exp(\Upsilon B)) - \Upsilon \cdot \frac{B - \iota(\eta_{\mathsf{u}})}{2}. \tag{63}
\end{aligned}
$$

Assembling (62) and (63) for an upperbound to $L(\eta_{\mathrm{f}}; M_{\mathrm{t}}, \eta_{\mathrm{t}})$, we get, using the fact that an $\alpha$-tree partitions $\mathcal{X}$ into regions with constant predictions,

$$L(\eta_{\mathrm{f}}; M, \eta_{\mathrm{t}})$$
$$\doteq \mathbb{E}_{X \sim M}\left[\eta_{\mathrm{t}}(X) \cdot -\log \eta_{\mathrm{f}}(X) + (1 - \eta_{\mathrm{t}}(X)) \cdot -\log(1 - \eta_{\mathrm{f}}(X))\right]$$
$$\leq \mathbb{E}_{X \sim M_{\mathrm{t}}}\left[\begin{array}{c} \eta_{\mathrm{t}}(X) \cdot \left(\log(1 + \exp(\Upsilon(X)B)) - \Upsilon(X) \cdot \frac{B + \iota(\eta_u(X))}{2}\right) \\ +(1 - \eta_{\mathrm{t}}(X)) \cdot \left(\log(1 + \exp(\Upsilon(X)B)) - \Upsilon(X) \cdot \frac{B - \iota(\eta_u(X))}{2}\right) \end{array}\right]$$
$$= \mathbb{E}_{X \sim M_{\mathrm{t}}}\left[\log(1 + \exp(\Upsilon(X)B)) - \Upsilon(X) \cdot \left(\begin{array}{c} \eta_{\mathrm{t}}(X) \cdot \frac{B + \iota(\eta_u(X))}{2} \\ +(1 - \eta_{\mathrm{t}}(X)) \cdot \frac{B - \iota(\eta_u(X))}{2} \end{array}\right)\right]$$
$$= \mathbb{E}_{\lambda \sim M_{\Lambda(\Upsilon)}}\left[\log(1 + \exp(\Upsilon(\lambda)B)) - \Upsilon(\lambda) \cdot \mathbb{E}_{X \sim M_\lambda}\left[\left(\begin{array}{c} \eta_{\mathrm{t}}(X) \cdot \frac{B + \iota(\eta_u(X))}{2} \\ +(1 - \eta_{\mathrm{t}}(X)) \cdot \frac{B - \iota(\eta_u(X))}{2} \end{array}\right)\right]\right]$$
$$= \mathbb{E}_{\lambda \sim M_{\Lambda(\Upsilon)}}\left[\log(1 + \exp(\Upsilon(\lambda)B)) - \Upsilon(\lambda) \cdot \mathbb{E}_{(X,Y) \sim D_{t\lambda}}\left[\frac{B + Y \cdot \iota(\eta_u(X))}{2}\right]\right]$$
$$= \mathbb{E}_{\lambda \sim M_{\Lambda(\Upsilon)}}\left[\log(1 + \exp(\Upsilon(\lambda)B)) - \Upsilon(\lambda) \cdot \frac{B + \mathbb{E}_{(X,Y) \sim D_{t\lambda}}[Y \cdot \iota(\eta_u(X))]}{2}\right]$$
$$= \mathbb{E}_{\lambda \sim M_{\Lambda(\Upsilon)}}\left[\log(1 + \exp(\Upsilon(\lambda)B)) - \Upsilon(\lambda)B \cdot \frac{1 + \mathsf{e}(M_\lambda, \eta_{\mathrm{t}})}{2}\right], \tag{64}$$

where we have used index notation for leaves introduced in the Theorem's statement, used the definition of $\mathsf{e}(M_\lambda, \eta_{\mathrm{t}})$ and let $\Upsilon(\lambda)$ denote $\lambda$'s leaf value in $\Upsilon$. Looking at (64), we see that we can design the leaf values to minimize each contribution to the expectation (noting the convexity of the relevant functions in $\Upsilon(\lambda)$), which for any $\lambda \in \Lambda(\Upsilon)$ we define with a slight abuse of notations as:

$$L(\Upsilon(\lambda)) \doteq \log(1 + \exp(\Upsilon(\lambda)B)) - \Upsilon(\lambda)B \cdot \frac{1 + \mathsf{e}(M_\lambda, \eta_{\mathrm{t}})}{2}. \tag{65}$$

We note

$$L'(\Upsilon(\lambda)) = B \cdot \left(\frac{\exp(\Upsilon(\lambda)B)}{1 + \exp(\Upsilon(\lambda)B)} - \frac{1 + \mathsf{e}(M_\lambda, \eta_{\mathrm{t}})}{2}\right),$$

which zeroes for

$$\Upsilon(\lambda) = \frac{1}{B} \cdot \log\left(\frac{1 + \mathsf{e}(M_\lambda, \eta_{\mathrm{t}})}{1 - \mathsf{e}(M_\lambda, \eta_{\mathrm{t}})}\right) = \tilde{\iota}\left(\frac{1 + \mathsf{e}(M_\lambda, \eta_{\mathrm{t}})}{2}\right),$$

yielding the bound (we use $\mathsf{e}(\lambda)$ as a shorthand for $\mathsf{e}(M_\lambda, \eta_{\mathrm{t}})$):

$$L(\eta_{\mathrm{f}}; M_{\mathrm{t}}, \eta_{\mathrm{t}})$$
$$\leq \mathbb{E}_{\lambda \sim M_{\Lambda(\Upsilon)}}\left[\log\left(1 + \frac{1 + \mathsf{e}(\lambda)}{1 - \mathsf{e}(\lambda)}\right) - \log\left(\frac{1 + \mathsf{e}(\lambda)}{1 - \mathsf{e}(\lambda)}\right) \cdot \frac{1 + \mathsf{e}(\lambda)}{2}\right]$$
$$= \mathbb{E}_{\lambda \sim M_{\Lambda(\Upsilon)}}\left[-\log\left(\frac{1 - \mathsf{e}(\lambda)}{2}\right) - \log\left(\frac{1 + \mathsf{e}(\lambda)}{1 - \mathsf{e}(\lambda)}\right) \cdot \frac{1 + \mathsf{e}(\lambda)}{2}\right]$$
$$= \mathbb{E}_{\lambda \sim M_{\Lambda(\Upsilon)}}\left[-\log\left(\frac{1 - \mathsf{e}(\lambda)}{2}\right) + \frac{1 + \mathsf{e}(\lambda)}{2} \cdot \log\left(\frac{1 - \mathsf{e}(\lambda)}{2}\right) - \frac{1 + \mathsf{e}(\lambda)}{2} \cdot \log\left(\frac{1 + \mathsf{e}(\lambda)}{2}\right)\right]$$
$$= \mathbb{E}_{\lambda \sim M_{\Lambda(\Upsilon)}}\left[-\frac{1 - \mathsf{e}(\lambda)}{2} \cdot \log\left(\frac{1 - \mathsf{e}(\lambda)}{2}\right) - \frac{1 + \mathsf{e}(\lambda)}{2} \cdot \log\left(\frac{1 + \mathsf{e}(\lambda)}{2}\right)\right]$$
$$= H(\Upsilon; M_{\mathrm{t}}, \eta_{\mathrm{t}}), \tag{66}$$

which is the statement of Theorem C. $\qquad\square$

Armed with Theorem C, what we now show is the boosting compliant convergence on the entropy of the $\alpha$-tree. For the informed reader, the proof of our result relies on a generalisation of [15, Lemma 2], then branching on the proofs of [15, Lemma 6, Theorem 9] to complete our result. For this objective, we first introduce notations, summarized in Figure 7, for the split of a leaf $\lambda_q$ in a subtree with two new leaves $\lambda_p, \lambda_r$. Here, we make use of simplified notation

$$\mathsf{e}_p \doteq \mathsf{e}(M_{\lambda_p}, \eta_{\mathrm{t}}), \tag{67}$$

and similarly for $\mathsf{e}_q$ and $\mathsf{e}_r$. Quantities $p, q, r \in [0, 1]$[4] are computed from the corresponding $\mathsf{e}_.$. $\tau$ is the probability, measured from $\mathrm{D}_{\mathsf{t}_{\lambda_q}}$, that an example has $h(.) = +1$, where $h$ is the split function at $\lambda_q$. We state and prove our generalisation to [15, Lemma 2].

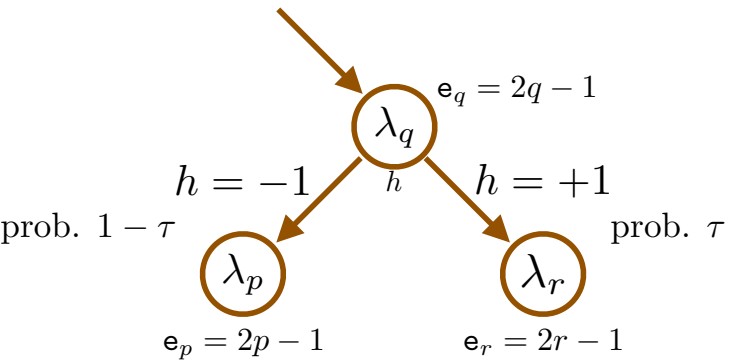

Figure 7: Main notations used in the proof of Theorem 3, closely following some notations of [15, Fig. 4].

**Lemma D.** *Assuming notations in Figure 7 for the split $h$ investigated at a leaf $\lambda_q$, and letting $\delta \doteq r - p$, if for some $\gamma > 0$ the split $h$ $\gamma$-witnesses the WHA at $\lambda$, then $\tau(1 - \tau)\delta \geq \gamma \cdot q(1 - q)$.*

**Proof:** Using the definition of the rebalanced distribution, we have:

$$\mathbb{E}_{(\mathsf{X},\mathsf{Y}) \sim \mathrm{D}'_{\mathsf{t}_{\lambda_q}}} \left[\mathsf{Y}\tilde{\iota}(\eta_{\mathsf{u}}(\mathsf{X}))h(\mathsf{X})\right]$$

$$= \mathbb{E}_{(\mathsf{X},\mathsf{Y}) \sim \mathrm{D}_{\mathsf{t}_{\lambda_q}}} \left[\frac{1 - \mathsf{e}_q \cdot \mathsf{Y} \cdot \tilde{\iota}(\eta_{\mathsf{u}}(\mathsf{X}))}{1 - \mathsf{e}_q^2} \cdot \mathsf{Y}h(\mathsf{X})\tilde{\iota}(\eta_{\mathsf{u}}(\mathsf{X}))\right]$$

$$= \frac{\mathbb{E}_{(\mathsf{X},\mathsf{Y}) \sim \mathrm{D}_{\mathsf{t}_{\lambda_q}}} \left[\mathsf{Y}h(\mathsf{X})\tilde{\iota}(\eta_{\mathsf{u}}(\mathsf{X}))\right] - \mathsf{e}_q \cdot \mathbb{E}_{(\mathsf{X},\mathsf{Y}) \sim \mathrm{D}_{\mathsf{t}_{\lambda_q}}} \left[\tilde{\iota}^2(\eta_{\mathsf{u}}(\mathsf{X}))h(\mathsf{X})\right]}{1 - \mathsf{e}_q^2}, \tag{68}$$

since $y^2 = 1, \forall y \in \mathcal{Y}$. We also have, by definition of the partition induced by $h$ and the definition of $\tau$,

$$\tau\mathsf{e}_r - (1 - \tau)\mathsf{e}_p = \tau \cdot \mathbb{E}_{(\mathsf{X},\mathsf{Y}) \sim \mathrm{D}_{\mathsf{t}_{\lambda_r}}} \left[\mathsf{Y} \cdot \tilde{\iota}(\eta_{\mathsf{u}}(\mathsf{X}))\right] - (1 - \tau) \cdot \mathbb{E}_{(\mathsf{X},\mathsf{Y}) \sim \mathrm{D}_{\mathsf{t}_{\lambda_p}}} \left[\mathsf{Y} \cdot \tilde{\iota}(\eta_{\mathsf{u}}(\mathsf{X}))\right]$$

$$= \mathbb{E}_{(\mathsf{X},\mathsf{Y}) \sim \mathrm{D}_{\mathsf{t}_{\lambda_q}}} \left[\mathsf{Y}h(\mathsf{X})\tilde{\iota}(\eta_{\mathsf{u}}(\mathsf{X}))\right]. \tag{69}$$

We can thus write:

$$\mathbb{E}_{(\mathsf{X},\mathsf{Y}) \sim \mathrm{D}'_{\mathsf{t}_{\lambda_q}}} \left[\mathsf{Y}\tilde{\iota}(\eta_{\mathsf{u}}(\mathsf{X}))h(\mathsf{X})\right]$$

$$= \frac{\tau\mathsf{e}_r - (1 - \tau)\mathsf{e}_p - \mathsf{e}_q \cdot \mathbb{E}_{(\mathsf{X},\mathsf{Y}) \sim \mathrm{D}_{\mathsf{t}_{\lambda_q}}} \left[\tilde{\iota}^2(\eta_{\mathsf{u}}(\mathsf{X}))h(\mathsf{X})\right]}{1 - \mathsf{e}_q^2} \tag{70}$$

$$= \frac{2\tau\mathsf{e}_r - \mathsf{e}_q \cdot \left(1 + \mathbb{E}_{(\mathsf{X},\mathsf{Y}) \sim \mathrm{D}_{\mathsf{t}_{\lambda_q}}} \left[\tilde{\iota}^2(\eta_{\mathsf{u}}(\mathsf{X}))h(\mathsf{X})\right]\right)}{1 - \mathsf{e}_q^2} \tag{71}$$

$$= \frac{2\tau\mathsf{e}_r - 2\tau\mathsf{e}_q}{1 - \mathsf{e}_q^2} - \mathsf{e}_q \cdot \frac{\left(1 - 2\tau + \mathbb{E}_{(\mathsf{X},\mathsf{Y}) \sim \mathrm{D}_{\mathsf{t}_{\lambda_q}}} \left[\tilde{\iota}^2(\eta_{\mathsf{u}}(\mathsf{X}))h(\mathsf{X})\right]\right)}{1 - \mathsf{e}_q^2} \tag{72}$$

$$= \frac{2\tau\mathsf{e}_r - 2\tau\mathsf{e}_q}{1 - \mathsf{e}_q^2} + \mathsf{e}_q \cdot \frac{\left(2\tau - 1 - \mathbb{E}_{(\mathsf{X},\mathsf{Y}) \sim \mathrm{D}_{\mathsf{t}_{\lambda_q}}} \left[\tilde{\iota}^2(\eta_{\mathsf{u}}(\mathsf{X}))h(\mathsf{X})\right]\right)}{1 - \mathsf{e}_q^2} \tag{73}$$

$$= \frac{2\tau\mathsf{e}_r - 2\tau\mathsf{e}_q}{1 - \mathsf{e}_q^2} + \frac{\mathsf{e}_q \cdot \mathbb{E}_{(\mathsf{X},\mathsf{Y}) \sim \mathrm{D}_{\mathsf{t}_{\lambda_q}}} \left[(1 - \tilde{\iota}^2(\eta_{\mathsf{u}}(\mathsf{X}))) \cdot h(\mathsf{X})\right]}{1 - \mathsf{e}_q^2}. \tag{74}$$

[4]Under Assumption 1.

Here, (70) follows from (68) and (69), (71) uses the fact that $\mathsf{e}_q = (1-\tau)\mathsf{e}_p + \tau\mathsf{e}_r$, (72) and (73) are convenient reformulations after adding $2\tau\mathsf{e}_q - 2\tau\mathsf{e}_q$ and (74) follows from $\mathbb{E}_{(\mathsf{X},\mathsf{Y})\sim D_{t_{\lambda_q}}}[h(\mathsf{X})] = 2\tau - 1$ by definition of $\tau$ and $h \in \{-1, 1\}$. Let

$$\Delta(h) \doteq \mathsf{e}_q \cdot \mathbb{E}_{(\mathsf{X},\mathsf{Y})\sim D_{t_{\lambda_q}}}\left[(1 - \tilde{\iota}^2(\eta_u(\mathsf{X}))) \cdot h(\mathsf{X})\right]. \tag{75}$$

We have $p = (1 + \mathsf{e}_p)/2$ (and similarly for $q = (1 + \mathsf{e}_q)/2$ and $r = (1 + \mathsf{e}_r)/2$), so we reformulate (73) as:

$$
\begin{aligned}
\mathbb{E}_{(\mathsf{X},\mathsf{Y})\sim D'_{t_{\lambda_q}}}\left[\mathsf{Y}\tilde{\iota}(\eta_u(\mathsf{X}))h(\mathsf{X})\right] &= \frac{2\tau(2r - 2q)}{4q(1-q)} + \frac{\Delta(h)}{4q(1-q)} \\
&= \frac{\tau(r - q)}{q(1-q)} + \frac{\Delta(h)}{4q(1-q)} \\
&= \frac{\tau(1-\tau)\delta}{q(1-q)} + \frac{\Delta(h)}{4q(1-q)},
\end{aligned} \tag{76}
$$

where the last identity comes from the fact that $r = q + (1-\tau)\delta$. We now have two cases depending on what removing the absolute value in the WHA leads to:

**Case 1** (i) is $\mathbb{E}_{(\mathsf{X},\mathsf{Y})\sim D'_{t_{\lambda_q}}}\left[\mathsf{Y}\tilde{\iota}(\eta_u(\mathsf{X}))h(\mathsf{X})\right] \geq \gamma$. We get from (76):

$$\tau(1-\tau)\delta \geq \gamma \cdot q(1-q) - \frac{\Delta(h)}{4}, \tag{77}$$

and since (ii) brings $\Delta(h) \leq 0$, we obtain $\tau(1-\tau)\delta \geq \gamma \cdot q(1-q)$, as claimed.

**Case 2** (i) is $\mathbb{E}_{(\mathsf{X},\mathsf{Y})\sim D'_{t_{\lambda_q}}}\left[\mathsf{Y}\tilde{\iota}(\eta_u(\mathsf{X}))h(\mathsf{X})\right] \leq -\gamma$. Since $\mathcal{H}$ is closed by negation we replace $h$ by $h' \doteq -h$, which satisfies $\mathbb{E}_{(\mathsf{X},\mathsf{Y})\sim D'_{t_{\lambda_q}}}\left[\mathsf{Y}\tilde{\iota}(\eta_u(\mathsf{X}))h'(\mathsf{X})\right] = -\mathbb{E}_{(\mathsf{X},\mathsf{Y})\sim D'_{t_{\lambda_q}}}\left[\mathsf{Y}\tilde{\iota}(\eta_u(\mathsf{X}))h(\mathsf{X})\right]$. The change switches the sign of $\delta$ by its definition and also $\Delta(h') = -\Delta(h)$ so (77) becomes $-\tau(1-\tau)\delta \leq -\gamma \cdot q(1-q) + \Delta(h')/4$, *i.e.*

$$\tau(1-\tau)\delta \geq \gamma \cdot q(1-q) - \frac{\Delta(h')}{4}, \tag{78}$$

which brings us back to Case 1 with the switch $h \leftrightarrow h'$ as $h'$ satisfies $\mathbb{E}_{(\mathsf{X},\mathsf{Y})\sim D'_{t_{\lambda_q}}}\left[\mathsf{Y}\tilde{\iota}(\eta_u(\mathsf{X}))h'(\mathsf{X})\right] \geq \gamma$. This ends the proof of Lemma D. $\square$

Branching Lemma D to the proof of Theorem 3 via the results of [15] is simple as all major parameters $p, q, r, \delta, \tau$ are either the same or satisfy the same key relationships (linked to the linearity of the expectation). This is why, if we compute the decrease $\mathrm{H}(\Upsilon; \mathsf{M}_t, \eta_t) - \mathrm{H}(\Upsilon(\lambda, h); \mathsf{M}_t, \eta_t)$, $\Upsilon(\lambda, h)$ being the $\alpha$-tree $\Upsilon$ with the split in Figure 7 performed with $h$ at $\lambda$, then we immediately get

$$\mathrm{H}(\Upsilon; \mathsf{M}_t, \eta_t) - \mathrm{H}(\Upsilon(\lambda, h); \mathsf{M}_t, \eta_t) \geq \gamma^2 q(1-q), \tag{79}$$

which comes from [15, Lemma 6], and (79) can be directly used in the proof of [15, Theorem 9] – which unravels the local decrease of $\mathrm{H}(.; \mathsf{M}_t, \eta_t)$ to get to the global decrease of the criterion for the whole of $\Upsilon$'s induction –, and to get $\mathrm{H}(\Upsilon; \mathsf{M}_t, \eta_t) \leq \varepsilon$, it is sufficient that

$$|\Lambda(g)| \geq \left(\frac{1}{\varepsilon}\right)^{\frac{c\log\left(\frac{1}{\varepsilon}\right)}{\gamma^2}}, \tag{80}$$

as claimed, for $c > 0$ a constant. This ends the proof of Theorem 3.

**Remark 1.** *Lemma C reveals an interesting property: instead of requesting $\Pi_{S',\lambda}(h) \leq 0$ in split-fair-compliance, suppose we strengthen the assumption, requesting for some $\beta > 0$ that*

$$\Pi_{S',\lambda}(h) \leq -\beta \cdot (1 - \mathsf{e}_q^2), \tag{81}$$

*then the "advantage" $\gamma$ becomes an advantage $\gamma + \beta$ in (80). Since we have $\Pi_{S',\lambda}(h) \doteq \mathsf{e}_q \cdot \mathbb{E}_{(\mathsf{X},\mathsf{Y})\sim D_{S',\lambda_q}}\left[(1 - \tilde{\iota}^2(\eta_u(\mathsf{X}))) \cdot h(\mathsf{X})\right]$, constraint (81) quickly vanishes as $|\mathsf{e}_q| \to 1$, i.e. as the black-box gest very good –or– very bad (in this last case, we remark that $1 - \eta_u$ becomes very good, so this is not a surprise). For example, if $\mathsf{e}_q \geq 1 - \varepsilon'$ for small $\varepsilon'$, then we just need*

$$\mathbb{E}_{(\mathsf{X},\mathsf{Y})\sim D_{S',\lambda_q}}\left[(1 - \tilde{\iota}^2(\eta_u(\mathsf{X}))) \cdot h(\mathsf{X})\right] \leq -\varepsilon'\beta \cdot \frac{2 - \varepsilon'}{1 - \varepsilon'}. \tag{82}$$

# VIII  Proof of Theorem 4

The proof is obtained via a generalisation of Lemma C.

**Lemma E.** *Fix any $B > 0$. For any $\alpha \in \mathbb{R}$, any $\theta, z \in [-B, B]$, if we let*

$$\vartheta(z) \;\doteq\; (z - \theta) \cdot \begin{cases} \frac{1}{B+\theta} & \text{if} \quad z < \theta, \\ 0 & \text{if} \quad z = \theta, \\ \frac{1}{B-\theta} & \text{if} \quad z > \theta. \end{cases} \tag{83}$$

*then we have*

$$\log(1 + \exp(\alpha z)) \;\leq\; \log\left(\frac{1 + \exp(B\alpha)}{1 + \exp(\theta\alpha)}\right) \cdot |\vartheta(z)| - B\alpha \max\{0, -\vartheta(z)\} + \log(1 + \exp(\theta\alpha)).$$

**Remark**: Lemma C is obtained for the choices $\theta = \pm B$.

**Proof:**  We fix any $\theta' \in [-1, 1]$ and let

$$\begin{align} \mathrm{l} &\;\doteq\; (-1, \log(1 + \exp(-\alpha))), \tag{84} \\ \mathrm{c} &\;\doteq\; (\theta', \log(1 + \exp(\alpha\theta'))), \tag{85} \\ \mathrm{r} &\;\doteq\; (1, \log(1 + \exp(\alpha))). \tag{86} \end{align}$$

The equation of the line passing through $\mathrm{l}, \mathrm{c}$ is

$$f_{\mathrm{l}}(z)$$

$$= \frac{\log\left(\frac{1+\exp(\theta'\alpha)}{1+\exp(-\alpha)}\right)}{1 + \theta'} \cdot z + \frac{\log\left(\frac{1+\exp(\theta'\alpha)}{1+\exp(-\alpha)}\right)}{1 + \theta'} + \log(1 + \exp(-\alpha)) \tag{87}$$

$$= -\frac{\log\left(\frac{1+\exp(\alpha)}{1+\exp(\theta'\alpha)}\right)}{1 + \theta'} \cdot z + \frac{\alpha z}{1 + \theta'} - \frac{\log\left(\frac{1+\exp(\alpha)}{1+\exp(\theta'\alpha)}\right)}{1 + \theta'} + \frac{\alpha}{1 + \theta'} + \log(1 + \exp(-\alpha)) \tag{88}$$

$$= \frac{\log\left(\frac{1+\exp(\alpha)}{1+\exp(\theta'\alpha)}\right)}{1 + \theta'} \cdot (\theta' - z) + \frac{\alpha(z - \theta')}{1 + \theta'} - \log\left(\frac{1 + \exp(\alpha)}{1 + \exp(\theta'\alpha)}\right) + \log(1 + \exp(\alpha)) \tag{89}$$

$$= \frac{\log\left(\frac{1+\exp(\alpha)}{1+\exp(\theta'\alpha)}\right)}{1 + \theta'} \cdot (\theta' - z) + \frac{\alpha(z - \theta')}{1 + \theta'} + \log(1 + \exp(\theta'\alpha)) \tag{90}$$

and the equation of the line passing through $\mathrm{c}, \mathrm{r}$ is

$$f_{\mathrm{r}}(z) \;=\; \frac{\log\left(\frac{1+\exp(\alpha)}{1+\exp(\theta'\alpha)}\right)}{1 - \theta'} \cdot z - \frac{\log\left(\frac{1+\exp(\alpha)}{1+\exp(\theta'\alpha)}\right)}{1 - \theta'} + \log(1 + \exp(\alpha)) \tag{91}$$

$$= \frac{\log\left(\frac{1+\exp(\alpha)}{1+\exp(\theta'\alpha)}\right)}{1 - \theta'} \cdot (z - \theta') - \log\left(\frac{1 + \exp(\alpha)}{1 + \exp(\theta'\alpha)}\right) + \log(1 + \exp(\alpha)) \tag{92}$$

$$= \frac{\log\left(\frac{1+\exp(\alpha)}{1+\exp(\theta'\alpha)}\right)}{1 - \theta'} \cdot (z - \theta') + \log(1 + \exp(\theta'\alpha)). \tag{93}$$

For any $z \in [-1, 1]$, define $\vartheta'(z) \in [-1, 1]$ to be:

$$\vartheta'(z) \;\doteq\; (z - \theta') \cdot \begin{cases} \frac{1}{1+\theta'} & \text{if} \quad z < \theta', \\ 0 & \text{if} \quad z = \theta', \\ \frac{1}{1-\theta'} & \text{if} \quad z > \theta'. \end{cases} \tag{94}$$

Function $z \mapsto \log(1 + \exp(\alpha z))$ being convex, we thus get the secant upperbound:

$$\log(1 + \exp(\alpha z))$$

$$\leq \log\left(\frac{1 + \exp(\alpha)}{1 + \exp(\theta'\alpha)}\right) \cdot |\vartheta'(z)| + \alpha \min\{0, \vartheta'(z)\} + \log(1 + \exp(\theta'\alpha)), \tag{95}$$

and this holds for $z \in [-1, 1]$. If instead $z \in [-B, B]$, then letting $\theta \doteq B\theta' \in [-B, B]$, we note:

$$\log(1 + \exp(\alpha z))$$
$$= \log(1 + \exp(\alpha B \cdot (z/B)))$$
$$\leq \log\left(\frac{1 + \exp(\alpha B)}{1 + \exp(\theta'\alpha B)}\right) \cdot |\vartheta'(z/B)| + \alpha B \min\{0, \vartheta'(z/B)\} + \log(1 + \exp(\theta'\alpha B)), \quad (96)$$

where this time,

$$\vartheta'\left(\frac{z}{B}\right) \doteq \left(\frac{z}{B} - \theta'\right) \cdot \begin{cases} \frac{1}{1+\theta'} & \text{if} \quad z < B\theta', \\ 0 & \text{if} \quad z = B\theta', \\ \frac{1}{1-\theta'} & \text{if} \quad z > B\theta'. \end{cases}$$

$$= (z - \theta) \cdot \begin{cases} \frac{1}{B+\theta} & \text{if} \quad z < \theta, \\ 0 & \text{if} \quad z = \theta, \quad \doteq \vartheta(z). \\ \frac{1}{B-\theta} & \text{if} \quad z > \theta. \end{cases} \quad (97)$$

We thus get

$$\log(1 + \exp(\alpha z))$$
$$\leq \log\left(\frac{1 + \exp(B\alpha)}{1 + \exp(\theta\alpha)}\right) \cdot |\vartheta(z)| + B\alpha \min\{0, \vartheta(z)\} + \log(1 + \exp(\theta\alpha)), \quad (98)$$

and since $\min\{0, z\} = -\max\{0, -z\}$, we get the statement of the Lemma. $\quad\square$

We use Lemma E with $\theta = 0$, which yields $\vartheta(z) = z/B$; using notations from the proof of Theorem C, we thus get (using the same notations as in the proof of Theorem 3),

$$-\log \eta_{\mathsf{f}}$$
$$= \log\left(1 + \exp\left(\Upsilon \cdot -\iota(\eta_{\mathsf{u}})\right)\right)$$
$$\leq \frac{1}{B} \cdot \log\left(\frac{1 + \exp(B\Upsilon)}{2}\right) \cdot |\iota(\eta_{\mathsf{u}}(\mathsf{X}))| + \Upsilon \min\{0, -\iota(\eta_{\mathsf{u}}(\mathsf{X}))\} + \log(2)$$
$$= \frac{1}{B} \cdot \log\left(\frac{1 + \exp(B\Upsilon)}{2}\right) \cdot |\iota(\eta_{\mathsf{u}}(\mathsf{X}))| - \Upsilon \max\{0, \iota(\eta_{\mathsf{u}}(\mathsf{X}))\} + \log(2) \quad (99)$$

$$-\log(1 - \eta_{\mathsf{f}})$$
$$= \log\left(1 + \exp\left(\Upsilon \cdot \iota(\eta_{\mathsf{u}}(\mathsf{X}))\right)\right)$$
$$\leq \frac{1}{B} \cdot \log\left(\frac{1 + \exp(B\Upsilon)}{2}\right) \cdot |\iota(\eta_{\mathsf{u}}(\mathsf{X}))| + \Upsilon \min\{0, \iota(\eta_{\mathsf{u}}(\mathsf{X}))\} + \log(2)$$
$$= \frac{1}{B} \cdot \log\left(\frac{1 + \exp(B\Upsilon)}{2}\right) \cdot |\iota(\eta_{\mathsf{u}}(\mathsf{X}))| - \Upsilon \max\{0, -\iota(\eta_{\mathsf{u}}(\mathsf{X}))\} + \log(2). \quad (100)$$

We get that the inequality in (64) now reads (for *any* values $\{\Upsilon(\lambda), \lambda \in \Lambda(\Upsilon)\}$) $L(\eta_{\mathsf{f}}; \mathsf{M}_{\mathsf{t}}, \eta_{\mathsf{t}}) = \mathbb{E}_{\lambda \sim \mathsf{M}_{\Lambda(\Upsilon)}}[J(\lambda)]$ with $J(\lambda)$ satisfying:

$$J(\lambda) \leq \mathbb{E}_{\mathsf{X} \sim \mathsf{M}_\lambda}\left[\begin{array}{l} \eta_{\mathsf{t}}(\mathsf{X}) \cdot \left(\frac{1}{B} \cdot \log\left(\frac{1+\exp(B\Upsilon(\lambda))}{2}\right) \cdot |\iota(\eta_{\mathsf{u}}(\mathsf{X}))| - \Upsilon(\lambda) \max\{0, \iota(\eta_{\mathsf{u}}(\mathsf{X}))\} + \log(2)\right) \\ +(1 - \eta_{\mathsf{t}}(\mathsf{X})) \cdot \left(\frac{1}{B} \cdot \log\left(\frac{1+\exp(B\Upsilon(\lambda))}{2}\right) \cdot |\iota(\eta_{\mathsf{u}}(\mathsf{X}))| - \Upsilon(\lambda) \max\{0, -\iota(\eta_{\mathsf{u}}(\mathsf{X}))\} + \log(2)\right) \end{array}\right]$$
$$= \log(2) - B\Upsilon(\lambda) \cdot \mathsf{e}^+(\mathsf{M}_\lambda, \eta_{\mathsf{t}}) + \log\left(\frac{1 + \exp(B\Upsilon(\lambda))}{2}\right) \cdot (\mathsf{e}^+(\mathsf{M}_\lambda, \eta_{\mathsf{t}}) + \mathsf{e}^-(\mathsf{M}_\lambda, \eta_{\mathsf{t}})), \quad (101)$$

and the bound takes its minimum on $\Upsilon(\lambda)$ for

$$\Upsilon(\lambda) = \frac{1}{B} \cdot \log\left(\frac{\mathsf{e}^+(\mathsf{M}_\lambda, \eta_{\mathsf{t}})}{\mathsf{e}^-(\mathsf{M}_\lambda, \eta_{\mathsf{t}})}\right) = \tilde{\iota}\left(\frac{\mathsf{e}^+(\mathsf{M}_\lambda, \eta_{\mathsf{t}})}{\mathsf{e}^+(\mathsf{M}_\lambda, \eta_{\mathsf{t}}) + \mathsf{e}^-(\mathsf{M}_\lambda, \eta_{\mathsf{t}})}\right), \quad (102)$$

yielding (using notations from Theorem 4),

$$J(\lambda) \leq \log(2) \cdot \left(1 - \mathsf{e}_\lambda^- - \mathsf{e}_\lambda^+\right) - \mathsf{e}_\lambda^+ \cdot \log\left(\frac{\mathsf{e}_\lambda^+}{\mathsf{e}_\lambda^-}\right) + \log\left(\frac{\mathsf{e}_\lambda^- + \mathsf{e}_\lambda^+}{\mathsf{e}_\lambda^-}\right) \cdot (\mathsf{e}_\lambda^- + \mathsf{e}_\lambda^+)$$
$$= \log(2) \cdot \left(1 + (\mathsf{e}_\lambda^- + \mathsf{e}_\lambda^+) \cdot \left(H_2\left(\frac{\mathsf{e}_\lambda^+}{\mathsf{e}_\lambda^+ + \mathsf{e}_\lambda^-}\right) - 1\right)\right), \quad (103)$$

and brings the statement of Theorem 4 after plugging the bound in the expectation.

## IX Proof of Lemma 3

We note that $H_2(1/2) = 1$, so we can reformulate:

$$\frac{H_2(\lambda; M, \eta_t)}{\log 2} = (1 - (e_\lambda^+ + e_\lambda^-)) \cdot H_2\left(\frac{1}{2}\right) + (e_\lambda^+ + e_\lambda^-) \cdot H_2\left(\frac{e_\lambda^+}{e_\lambda^+ + e_\lambda^-}\right), \quad (104)$$

and we also have $e_\lambda^+ \leq 0, e_\lambda^- \geq 0, e_\lambda^+ + e_\lambda^- \leq 1$, plus

$$(1 - (e_\lambda^+ + e_\lambda^-)) \cdot \left(\frac{1}{2}\right) + (e_\lambda^+ + e_\lambda^-) \cdot \left(\frac{e_\lambda^+}{e_\lambda^+ + e_\lambda^-}\right) = \frac{1 + e_\lambda^+ - e_\lambda^-}{2} = \frac{1 + e(M_\lambda, \eta_t)}{2}, \quad (105)$$

as indeed $e(M_\lambda, \eta_t) = e_\lambda^+ - e_\lambda^-$ from its definition. Thus, by Jensen's inequality, since $H$ is concave,

$$\log(2) \cdot \left(1 + (e_\lambda^+ + e_\lambda^-) \cdot \left(H_2\left(\frac{e_\lambda^+}{e_\lambda^+ + e_\lambda^-}\right) - 1\right)\right)$$

$$= \log(2) \cdot \left((1 - (e_\lambda^+ + e_\lambda^-)) \cdot H_2\left(\frac{1}{2}\right) + (e_{\rho,\lambda}^- + e_{\rho,\lambda}^+) \cdot H_2\left(\frac{e_\lambda^+}{e_\lambda^+ + e_\lambda^-}\right)\right)$$

$$\leq \log(2) \cdot H_2\left((1 - (e_\lambda^+ + e_\lambda^-)) \cdot \frac{1}{2} + (e_\lambda^+ + e_\lambda^-) \cdot \frac{e_\lambda^+}{e_\lambda^+ + e_\lambda^-}\right)$$

$$= \log(2) \cdot H_2\left(\frac{1 + e(M_\lambda, \eta_t)}{2}\right)$$

$$= H\left(\frac{1 + e(M_\lambda, \eta_t)}{2}\right),$$

which, after plugging in expectations and simplifying, yields the statement of Lemma 3.

## X Proof of Theorem 5

We remind that we craft product measures using a mixture and a posterior that shall be implicit from context: we thus note that the KL divergence

$$\text{KL}(\eta_t, \eta_f; M_t) \doteq \mathbb{E}_{(X,Y) \sim D_t}\left[\log\left(\frac{dD_t((X, Y))}{dD_f((X, Y))}\right)\right] \quad (106)$$

$$= \mathbb{E}_{X \sim M_t}\left[\eta_t(X) \cdot -\log\left(\frac{\eta_f(X)}{\eta_t(X)}\right) + (1 - \eta_t(X)) \cdot -\log\left(\frac{1 - \eta_f(X)}{1 - \eta_t(X)}\right)\right] \quad (107)$$

$$= L(\eta_f; M_t, \eta_t) - \mathbb{E}_{X \sim M_t}[H(\eta_t(X))], \quad (108)$$

where $D_t$ (resp. $D_f$) is obtained from couple $(M_t, \eta_t)$ (resp. $(M_t, \eta_f)$). Denote

$$s^\circ \doteq \arg\min_s \mathbb{P}_{X \sim P_s}[h_f(X) = 1], \quad (109)$$

where $h_f$ is the $+1/-1$ prediction obtained from the posterior $\eta_f$ using *e.g.* the sign of its logit. We define the total variation divergence:

$$\text{TV}(\eta_t, \eta_f; M_t) \doteq \int_{\mathcal{X} \times \mathcal{Y}} |dD_t((X, Y)) - dD_f((X, Y))|, \quad (110)$$

which, because of the definition of the product measures, is also equal to:

$$\text{TV}(\eta_t, \eta_f; M_t) = \int_{\mathcal{X}} |\eta_t(X)dM_t(X) - \eta_f(X)dM_t(X)| \quad (111)$$

$$+ \int_{\mathcal{X}} |(1 - \eta_t(X))dM_t(X) - (1 - \eta_f(X))dM_t(X)| \quad (112)$$

$$= 2 \int_{\mathcal{X}} |\eta_t(X) - \eta_f(X)|dM_t(X). \quad (113)$$

We have Pinsker's inequality, $\text{TV}(\eta_t, \eta_f; M_t) \leq \sqrt{2\text{KL}(\eta_t, \eta_f; M_t)}$ (see *e.g.* [30]), so if we run TOPDOWN until

$$L(\eta_f; M_t, \eta_t) \quad \leq \quad \frac{\tau^2}{2} + \mathbb{E}_{X \sim M_t}\left[H(\eta_t(X))\right], \tag{114}$$

then because of (108) and (113),

$$\int_{\mathcal{X}} |\eta_t(X) - \eta_f(X)| \text{d}M_t(X) \quad \leq \quad \tau. \tag{115}$$

Denote subgroups $s^\star \doteq \arg\max_s \mathbb{P}_{X \sim P_s}[h_f(X) = 1]$ and $s^\circ \doteq \arg\min_s \mathbb{P}_{X \sim P_s}[h_f(X) = 1]$. We pick

$$M_t \quad \leftarrow \quad P_{s^\circ} \tag{116}$$

for TOPDOWN and the $(p, \delta)$-push up posterior $\eta_t$, with

$$p \quad \doteq \quad \mathbb{P}_{X \sim P_{s^\star}}[h_f(X) = 1] + \frac{\delta}{2}, \tag{117}$$

assuming the RHS is $\leq 1$.

Denote $\mathcal{X}_{p,s^\circ}$ the subset of the support of $P_{s^\circ}$ such that $\eta_t(X) \geq (1/2) + \delta$. Notice that by definition,

$$\int_{\mathcal{X}_{p,s^\circ}} \text{d}P_{s^\circ}(X) \quad = \quad p. \tag{118}$$

We have two possible outcomes for $\eta_f$ of relevance on $\mathcal{X}_{p,s^\circ}$: (i) $\eta_f(X) \leq 1/2$ and (ii) $\eta_f(X) > 1/2$. Notice that in this latter case, we are guaranteed that $h_f(X) = 1$, which counts towards bringing closer $\mathbb{P}_{X \sim P_{s^\circ}}[h_f(X) = 1]$ to $\mathbb{P}_{X \sim P_{s^\star}}[h_f(X) = 1]$, so we have to make sure that (i) occurs with sufficiently small probability, and this is achieved via guarantee (115).

If the total weight on $\mathcal{X}_{p,s^\circ}$ of the event (i) $\eta_f(X) \leq 1/2$ is more than $\delta$, then

$$\begin{aligned}
\int_{\mathcal{X}} |\eta_t(X) - \eta_f(X)| \text{d}P_{s^\circ}(X) &\geq \int_{\mathcal{X}_{p,s^\circ}} |\eta_t(X) - \eta_f(X)| \text{d}P_{s^\circ}(X) \\
&\geq \left|\frac{1}{2} + \delta - \frac{1}{2}\right| \cdot \int_{\mathcal{X}_{p,s^\circ}} [\![\eta_f(X) \leq 1/2]\!] \text{d}P_{s^\circ}(X) \\
&> \left|\frac{1}{2} + \delta - \frac{1}{2}\right| \cdot \delta \\
&= \delta^2. 
\end{aligned} \tag{119}$$

If we have the relationship $\delta = \sqrt{\tau}$, then we get a contradiction with (115). In conclusion, if (114) holds, then

$$\int_{\mathcal{X}_{p,s^\circ}} [\![\eta_f(X) \leq 1/2]\!] \text{d}P_{s^\circ}(X) \quad \leq \quad \delta. \tag{120}$$

In summary, for any $\tau > 0$, if we run TOPDOWN with the choices $M_t \leftarrow P_{s^\circ}$ (which corresponds to the "worst treated" subgroup with respect to EOO) and craft the $(p, \delta)$-push up posterior $\eta_t$ with $p$ as in (117), then

$$\begin{aligned}
\mathbb{P}_{X \sim P_{s^\circ}}[h_f(X) = 1] &\geq \int_{\mathcal{X}_{p,s^\circ}} [\![\eta_f(X) > 1/2]\!] \text{d}P_{s^\circ}(X) \tag{121} \\
&= \int_{\mathcal{X}_{p,s^\circ}} (1 - [\![\eta_f(X) \leq 1/2]\!]) \text{d}P_{s^\circ}(X) \tag{122} \\
&= \int_{\mathcal{X}_{p,s^\circ}} \text{d}P_{s^\circ}(X) - \int_{\mathcal{X}_{p,s^\circ}} [\![\eta_f(X) \leq 1/2]\!] \text{d}P_{s^\circ}(X) \tag{123} \\
&\geq p - \delta \tag{124} \\
&= \mathbb{P}_{X \sim P_{s^\star}}[h_f(X) = 1] - \frac{\delta}{2}, \tag{125}
\end{aligned}$$

where (124) makes use of (118) and (120). Fixing $\delta \doteq 2\varepsilon$, $\varepsilon$ being used in (16) (main file), we obtain

$$\mathbb{P}_{\mathsf{X} \sim \mathrm{P}_{s^\star}} [h_{\mathrm{f}}(\mathsf{X}) = 1] - \mathbb{P}_{\mathsf{X} \sim \mathrm{P}_{s^\circ}} [h_{\mathrm{f}}(\mathsf{X}) = 1] \;\; \leq \;\; \varepsilon, \tag{126}$$

and via relationship $\delta = \sqrt{\tau}$, we check that (114) becomes the following function of $\varepsilon$:

$$L(\eta_{\mathrm{f}}; \mathrm{M}_{\mathrm{t}}, \eta_{\mathrm{t}}) \;\; \leq \;\; 8\varepsilon^4 + \mathbb{E}_{\mathsf{X} \sim \mathrm{M}_{\mathrm{t}}} [H(\eta_{\mathrm{t}}(\mathsf{X}))], \tag{127}$$

and we get the statement of the Theorem for the choice (117), which corresponds to $K = 2$ and reads

$$p \;\; \doteq \;\; \mathbb{P}_{\mathsf{X} \sim \mathrm{P}_{s^\star}} [h_{\mathrm{f}}(\mathsf{X}) = 1] + \varepsilon. \tag{128}$$

If the RHS in (128) is not $\leq 1$, we can opt for an alternative with one more free variable, $K \geq 1$,

$$p \;\; \doteq \;\; \mathbb{P}_{\mathsf{X} \sim \mathrm{P}_{s^\star}} [h_{\mathrm{f}}(\mathsf{X}) = 1] + \frac{\delta}{K}, \tag{129}$$

where $K$ is large enough for the constraint to hold. In this case, to keep (126) we must have $\delta(K-1)/K = \varepsilon$, which elicitates

$$\delta \;\; = \;\; \frac{K\varepsilon}{K-1} \tag{130}$$

instead of $\delta \doteq 2\varepsilon$, bringing

$$p \;\; \doteq \;\; \mathbb{P}_{\mathsf{X} \sim \mathrm{P}_{s^\star}} [h_{\mathrm{f}}(\mathsf{X}) = 1] + \frac{\varepsilon}{K-1}, \tag{131}$$

and a desired approximation guarantee for TOPDOWN of:

$$L(\eta_{\mathrm{f}}; \mathrm{M}_{\mathrm{t}}, \eta_{\mathrm{t}}) \;\; \leq \;\; \frac{K^4}{2(K-1)^4} \cdot \varepsilon^4 + \mathbb{E}_{\mathsf{X} \sim \mathrm{M}_{\mathrm{t}}} [H(\eta_{\mathrm{t}}(\mathsf{X}))]. \tag{132}$$

Since $K > 1$, $K^4/(K-1)^4 \geq 1$, so we are guaranteed that (132) holds if we ask for

$$L(\eta_{\mathrm{f}}; \mathrm{M}_{\mathrm{t}}, \eta_{\mathrm{t}}) \;\; \leq \;\; \frac{\varepsilon^4}{2} + \mathbb{E}_{\mathsf{X} \sim \mathrm{M}_{\mathrm{t}}} [H(\eta_{\mathrm{t}}(\mathsf{X}))], \tag{133}$$

# XI  SI Experiment Settings

In this SI section, we briefly discuss the additional datasets[5] and experimental settings included in the subsequent sections. In particular, we highlight the datasets used, the black-boxes post-processed, and specifics of the TOPDOWN algorithm. German Credit and Bank are standard public benchmark datasets in the literature. ACS is a newer dataset with curation details listed in [9, Section 3].

**Datasets**

- **German Credit.** In the SI, we additionally consider the German Credit dataset, preprocessed by `AIF360` [4]. The dataset consists of only 1000 examples, which is the smallest of the 3 datasets considered. On the other hand, the dataset provided by `AIF360` contains 57 features, primarily from one-hot encoding.

- **Bank.** Another dataset we consider in the SI is the Bank dataset, preprocessed by `AIF360` [4]. The dataset consists 30488 examples, above the German Credit dataset but below the ACS datasets. The dataset also has 57 features which is largely from one-hot encoding.

- **ACS.** The American Community Survey dataset is the dataset we present in the main text. More specifically, we consider the income prediction task (as depicted in the `Folktables` `Python` package [9]) over 1-year survey periods in the state of CA. Our of the 3 datasets, ACS provides the largest dataset, with 187475 examples for the 2015 sample of the dataset. Despite this, `Folktables` only provides 10 features for its prediction task. Through one-hot encoding, this is extended to 29 features.

`AIF360` uses a Apache License 2.0 and `Folktables` uses a MIT License.

Additional $Z$-score normalization was used where appropriate. Sensitive attributes are binned into binary and trinary modalities, as specified in the main text (and one-hot encoded for the trinary case).

Each experiment / dataset is used with 5-fold cross-validation and further split such that there are subset partitions for: (1) training the black-box; (2) training a post-processing method; and (3) testing and evaluation. In particular, we utilize standard cross-validation to split the data into a 80:20 training testing split. The training split is then split randomly equally for separate training of the black-box and post-processing method. The final data splits result in 40:40:20 partitions.

**Black-boxes**

- **Random Forest.** As per the main text, we primarily consider a calibrated random forest classifier provided by the `scikit-learn` `Python` package. The un-calibrated random forest classifier consists of 50 decision trees in an ensemble. Each decision tree has a maximum depth of 4 and is trained on a 10% subset of the black-box training data. In calibration, 5 cross validation folds are used for Platt scaling.

- **Neural Network.** Additionally to random forests, we consider a calibrated neural network in the SI, also provided by `scikit-learn`. The un-calibrated neural network is trained using mostly default parameters provided by `scikit-learn`. The exception to this is the specification of 300 training iterations and the specification of 10% of the training set to be used for early stopping.

The black-boxes are additionally clipped to adhere to Assumption 1 with $B = 1$ for all sections except for Appendix XVII.

For the criteria we evaluate TOPDOWN and baselines to in the SI, we consider those introduced in the main-text alongside AUC as an additional metric for accuracy.

**Compute**

Compute was done with no GPUs. Virtual machines were used with RAM 16GB VCPUs 8 VCPU Disk 30GB from [a local HPC facility] (to be named after publication).

---

[5]Public at: https://github.com/Trusted-AI/AIF360

**Code**

The code used in this submission is attached in the supplementary material, which will be released upon publication.

**TOPDOWN Specifics**

The $\alpha$-trees learnt by TOPDOWN are initialized as per Fig. 3. That is, we initialize sub-$\alpha$-trees with $\alpha = 1$ for each of the modalities of the sensitive attribute. In addition, each split of the $\alpha$-tree consists of projects to a specific feature / attribute. The split is either a modality of the discrete feature or a single linear threshold of a continuous feature. In addition, to avoid over-fitting we restrict splits to only those which result in children node that have at least 10% of the parent node's examples; and at a minimum have at least $30$ examples for each child node.

For EOO we utilize a Gaussian Naive Bayes classifier from `scikit-learn` with default parameters to fit $\eta^\star$ from data. We note that no fine-tuning was done for this classifier.

In the SI, we consider all variants TOPDOWN examined in the main-text. Additionally we consider the symmetric variant of the SP TOPDOWN approach. We reiterate the two ways of enforcing SP: as per Section 6, there are two symmetric strategies for SP. In particular, we aim to match the either the maximum or minimum subgroup to the opposite extreme (conditionally) expected posterior. As such, we can either match to the largest posterior, which we denote as ($\uparrow$), or we can match to the smallest posterior ($\downarrow$). We already present SP $\uparrow$ in the main-text and additionally present evaluation for SP $\downarrow$ here.

## XII    Experiments on Statistical Parity

We refer to SI, Section II for the formal aspects of handling statistical parity. **For SP**, a similar pattern to that of EOO follows, except the differences are more extreme. In particular, the audacious approach fails to optimize for SP and instead harms it significantly, but does slightly improve MD. The audacious update is problematic here as the target $\eta_t$ in the SP strategy is a constant (and does not take into consideration of the subgroup being updated). As such the audacious approach should not be used for SP as it will optimize to match the constant $\eta_t$ more harshly. On the other hand, the conservative update variant provides an improvement to SP whilst antagonizing MD accuracy. Notably, the "best" iteration for SP and MD occurs at its first iteration (which shows interest in early stopping / pruning the $\alpha$-tree). This is expected, as there is large shift when changing from an $\alpha = 1$ to the initial rooted value (*e.g.*, (5)). The pattern of conservative updates being superior to the audacious counterparts is consistent in other datasets and sensitive attribute modalities. Comparing the conservative SP TOPDOWN to the baselines, discounting OST for MD, we find that DERSP and RTO result in lower SP and MD. This is unsurprising: our TOPDOWN treatment SP can result in harsh updates; in SI (pg 15), we discuss an alternative approach using ties with optimal transport.

## XIII    Additional Main Text Experiments

In this section, we report the experiments identical to that presented in the main-text, including missing criteria, settings, and the additional German Credit and Bank datasets. Each plot we present provides the binary and trinary sensitive attribute settings over all criteria discussed in the previous setting.

In particular:

- Fig. 8 presents the evaluation using a RF black-box with $B = 1$ clipping on the German Credit dataset.

- Fig. 9 presents the evaluation using a RF black-box with $B = 1$ clipping on the Bank dataset.

- Fig. 10 presents the evaluation using a RF black-box with $B = 1$ clipping on the ACS dataset.

**Fairness Models**

In comparison to ACS, Fig. 9 for the Bank dataset performs similarly to the main text figure. There are only slight deviations in the ordering of which TOPDOWN settings perform best. For example, the CVAR optimization of audacious and conservative updates are a lot closer in the Bank dataset than that of the ACS 2015 dataset.

In comparison, the result's of TOPDOWN on the German Credit largely deviate from that of the other experiments. This can be clearly seen in the number of boosting iteration TOPDOWN completes being significantly lower before the entropy stops being decreased (and thus terminating the algorithm). Another major deviation is that CVAR fails to get lowered for both binary and trinary sensitive attribute modalities in the German Credit dataset. Despite this, EOO and SP both have slight improvements for the best corresponding TOPDOWN setting (conservative EOO and conservative SP ↑), which is consistent with other datasets. This is despite the original classifier's EOO and SP being significantly lower than the ACS dataset. However, there is a major cost in the case of EOO, where the accuracy (both for MD and AUC) is harmed significantly.

A reason for the significantly worse performance, predominantly in CVAR optimization, of TOP-DOWN for the German Credit is likely the significantly smaller number of example available in the dataset. Given that there are only 1000 examples and 57 features variables, the 40:40:20 split of the dataset results in the subsets to not be representative of the entire dataset's support. Additionally, CVAR is strongly tied to the cross-entropy loss function and empirical risk minimization [32, 25]. As such, given the nonrepresentative subsets of the dataset used for training TOPDOWN, minimizing the CVAR for low sample inputs is difficult. Thus for such a failure cases, one should confirm that TOPDOWN is not decreasing fairness to prevent social harms.

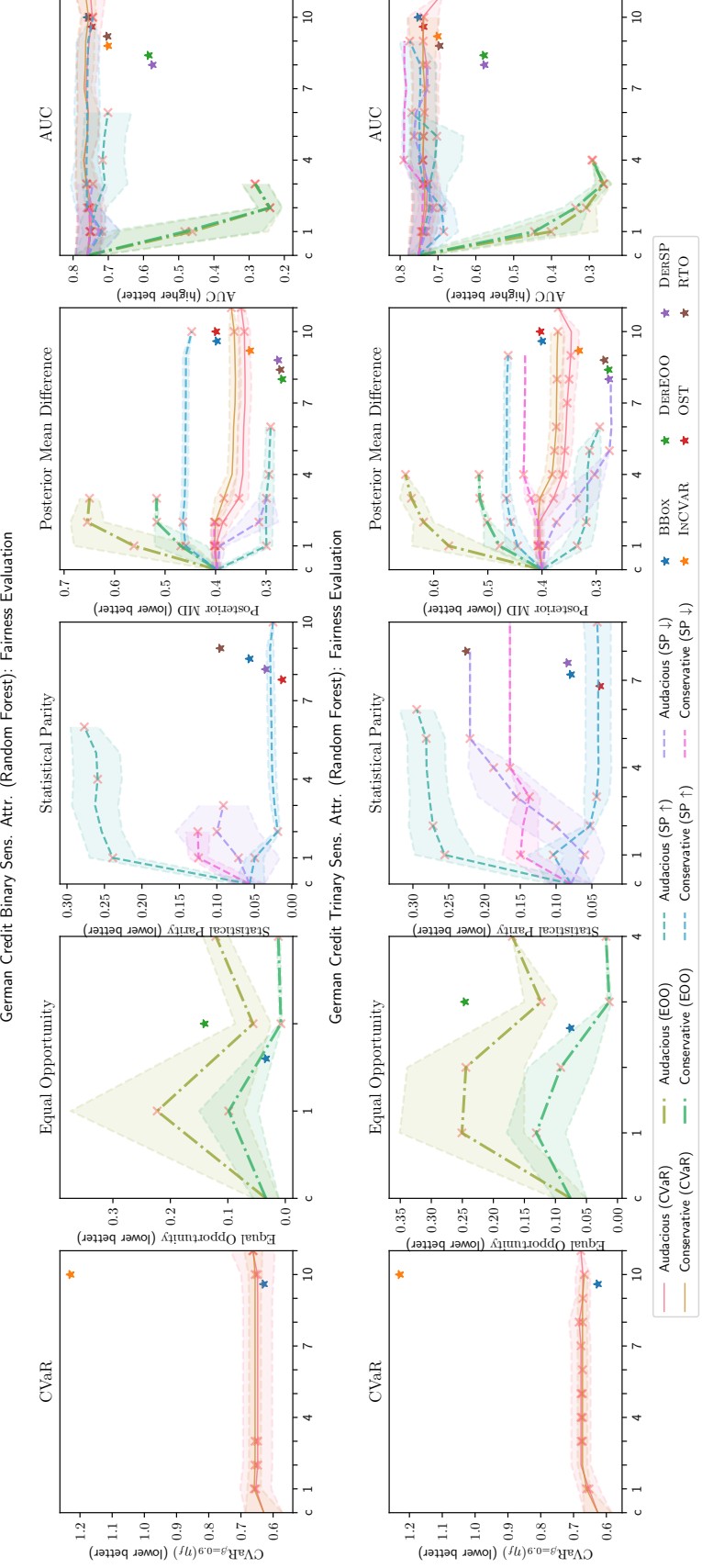

Figure 8: TOPDOWN optimized for different fairness models evaluated on German Credit with binary (up) and trinary (down) sensitive attributes. Crosses denote when a subgroup's α-tree is initiated (over any fold). The shade depicts ± a standard deviation from the mean. However, this disappears in the case where other folds stop early.

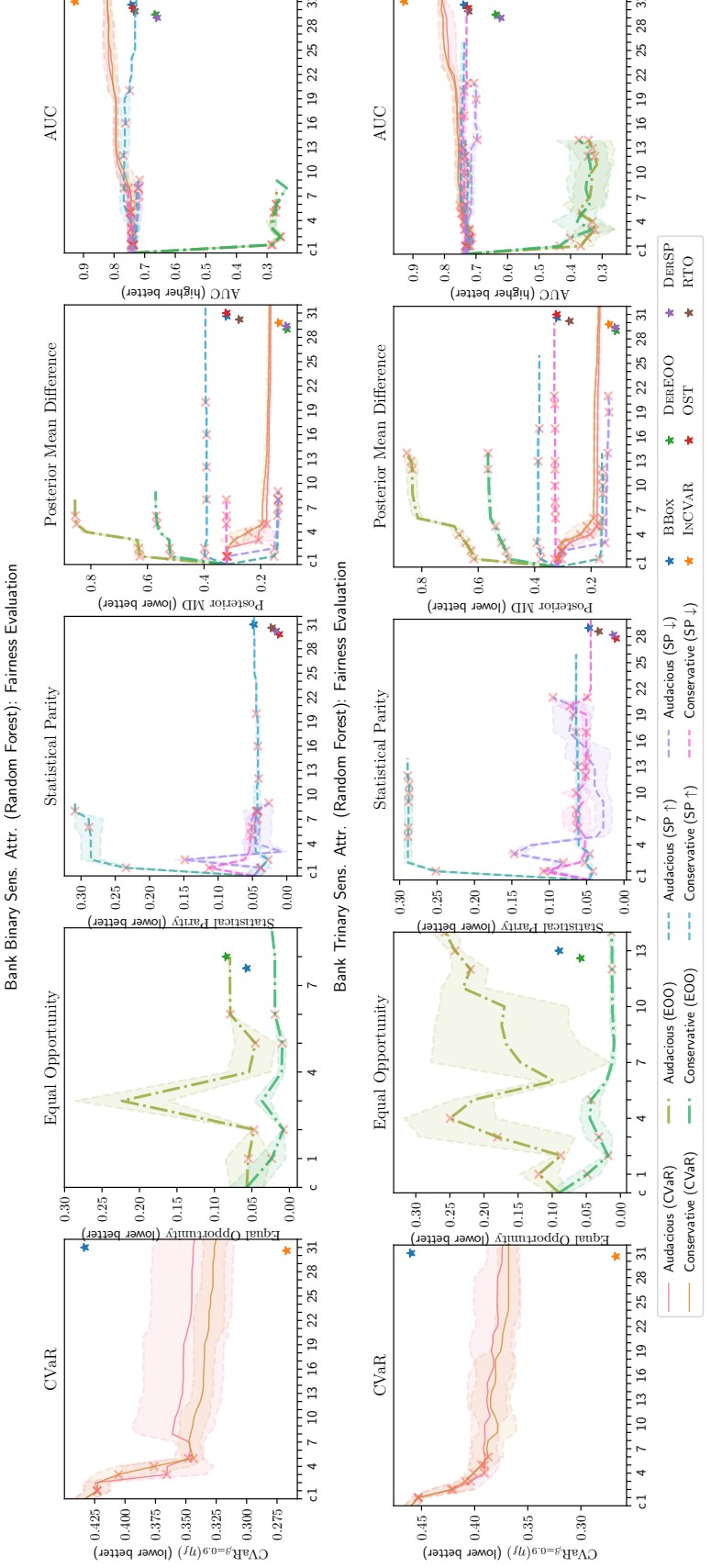

Figure 9: TopDown optimized for different fairness models evaluated on Bank with binary (up) and trinary (down) sensitive attributes. Crosses denote when a subgroup's $\alpha$-tree is initiated (over any fold). The shade depicts $\pm$ a standard deviation from the mean. However, this disappears in the case where other folds stop early.

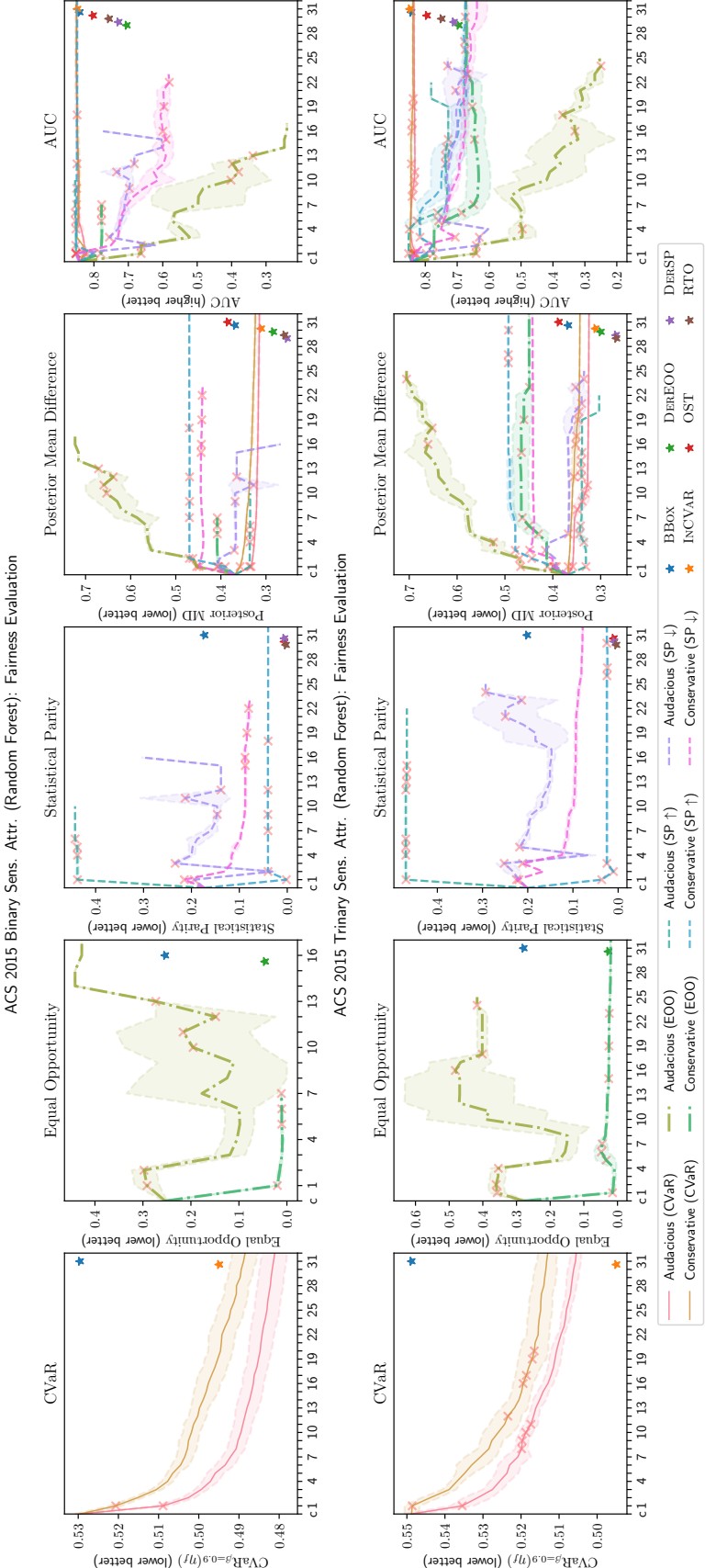

Figure 10: TopDown optimized over boosting iterations for different fairness models evaluated on ACS 2015 with binary (up) and trinary (down) sensitive attributes. "c" on the x-axis denotes the clipped black-box. Crosses denote when a subgroup's $\alpha$-tree is initiated (over any fold). The shade depicts $\pm$ a standard deviation from the mean. However, this disappears in the case where other folds stop early.

## XIV    Neural Network Experiments

In this SI section, we repeat all experiments evaluating different fairness models and proxy sensitive attributes using the neural network (NN) black-box. Figs. 11 to 13 presents neural network equivalent plots to that of Figs. 8 to 10.

In particular:

- Fig. 11 presents the evaluation using a NN black-box with $B = 1$ clipping on the German Credit dataset.
- Fig. 12 presents the evaluation using a NN black-box with $B = 1$ clipping on the Bank dataset.
- Fig. 13 presents the evaluation using a NN black-box with $B = 1$ clipping on the ACS dataset.

When comparing the NN experiments to the experiments corresponding to that of the random forest (RF) black-box experiments, only minor deviation can be seen with most trends staying the same. One consistent deviation is that the CVAR criterion and accuracy measures (MD and AUC) are frequently smaller at the initial and final point of boosting. This comes from the strong representation power of the NN black-box being translated from the initial black-box to the final wrapper classifier. In this regard, switching to a NN did not help the optimization of CVAR for the German Credit dataset, see Fig. 11.

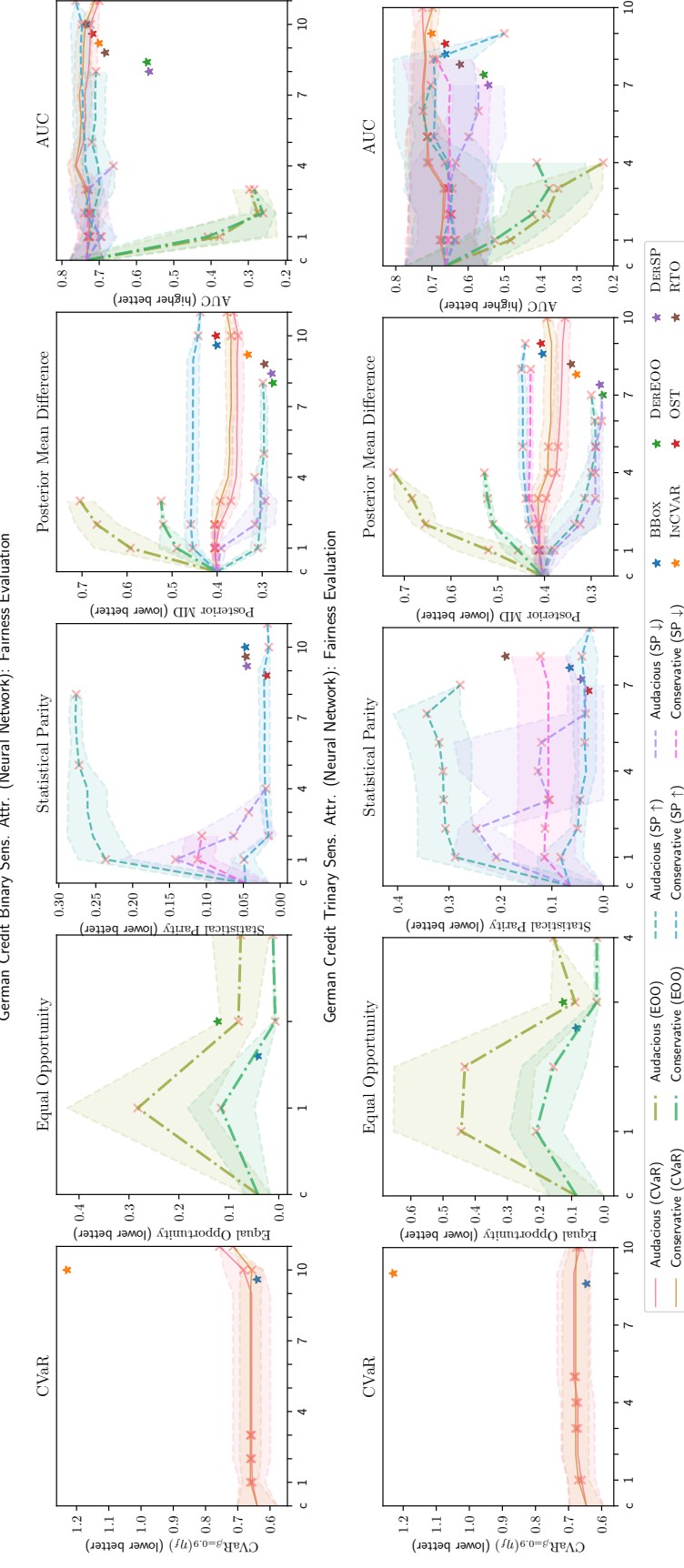

Figure 11: TopDown optimized for different fairness models evaluated on German Credit with binary (up) and trinary (down) sensitive attributes. Crosses denote when a subgroup's $\alpha$-tree is initiated (over any fold). The shade depicts $\pm$ a standard deviation from the mean. However, this disappears in the case where other folds stop early.

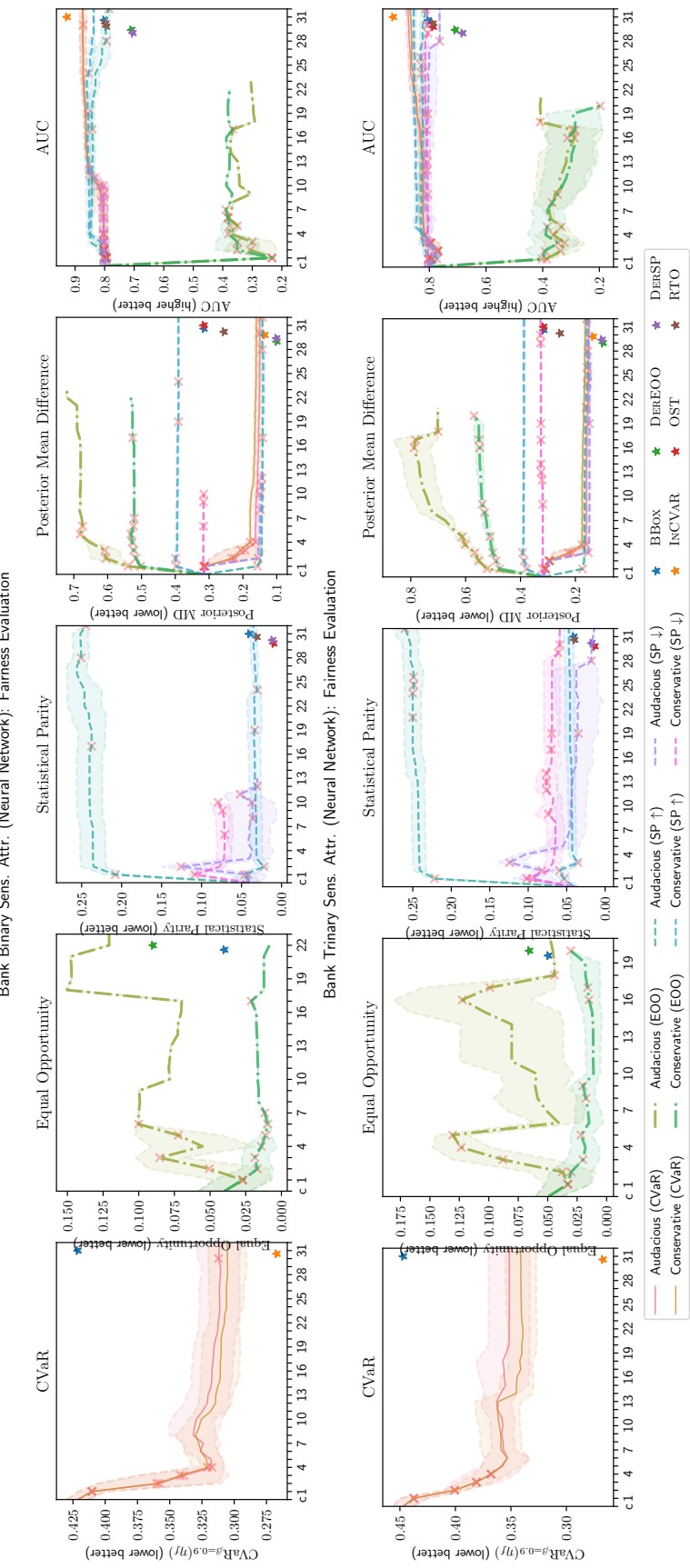

Figure 12: TOPDOWN optimized for different fairness models evaluated on Bank with binary (up) and trinary (down) sensitive attributes. Crosses denote when a subgroup's $\alpha$-tree is initiated (over any fold). The shade depicts ± a standard deviation from the mean. However, this disappears in the case where other folds stop early.

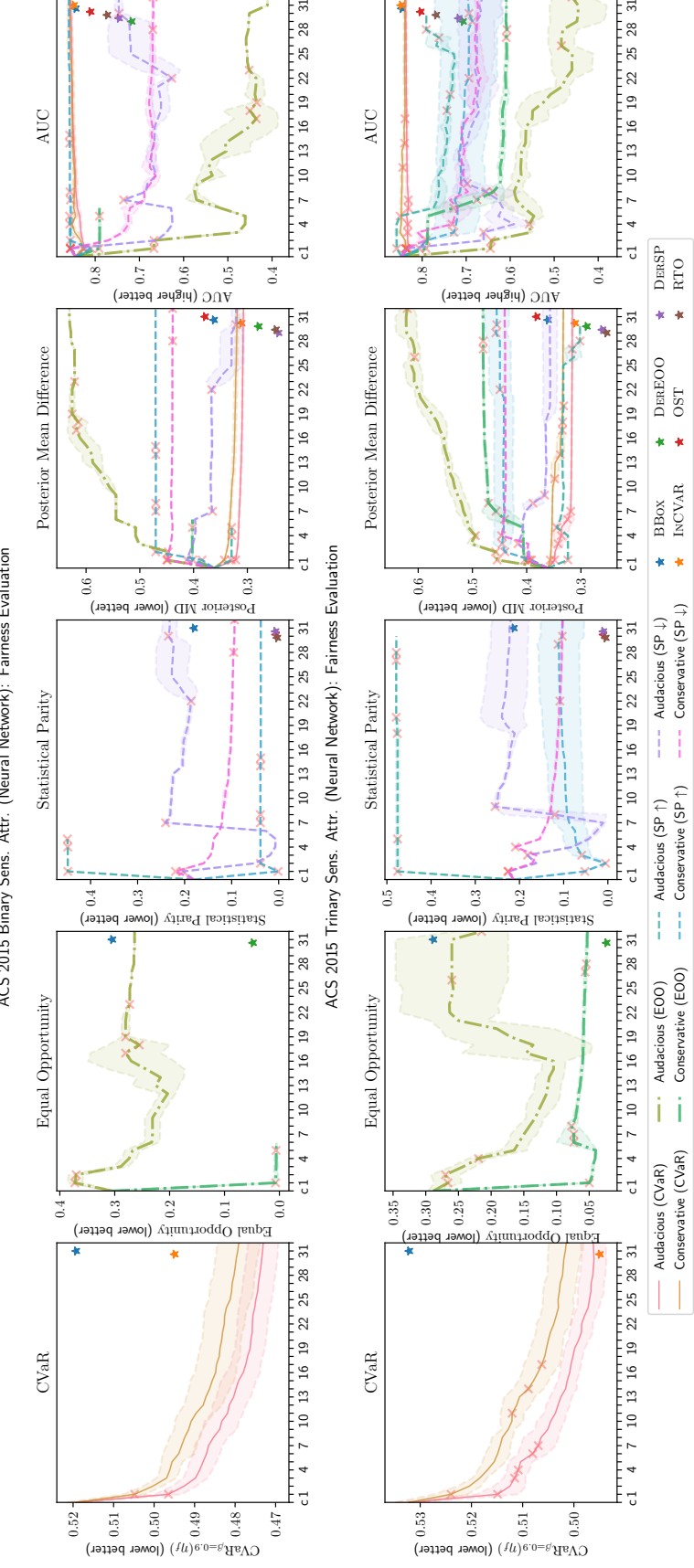

Figure 13: TopDown optimized for different fairness models evaluated on Bank with binary (up) and trinary (down) sensitive attributes. Crosses denote when a subgroup's $\alpha$-tree is initiated (over any fold). The shade depicts $\pm$ a standard deviation from the mean. However, this disappears in the case where other folds stop early.

## XV  Proxy Sensitive Attributes

We examine the use of sensitive attribute proxies to remove sensitive attribute requirements at test time. In particular, we use a decision tree with a maximum depth of $8$ to predict sensitive attributes (from other features) as a proxy to the true sensitive attribute. We present results for both RF and NN black-boxes.

In particular:

- Fig. 14 presents the proxy sensitive attribute evaluation using a RF and NN black-box with $B = 1$ clipping on the German Credit dataset.
- Fig. 15 presents the proxy sensitive attribute evaluation using a RF and NN black-box with $B = 1$ clipping on the Bank dataset.
- Fig. 16 presents the proxy sensitive attribute evaluation using a RF and NN black-box with $B = 1$ clipping on the ACS dataset.

Fig. 16 (top) presents the RF TOPDOWN proxy sensitive attribute experiments results of the ACS 2015 dataset not present in the main text. We focus on the binary case (left). Unsurprisingly, the proxy increases the variance of CVAR and AUC whilst also being worse than their non-proxy counterparts; but still manages to improve CVAR and AUC at the end (with an initial dip quickly erased for the latter criterion). Remark the non-trivial nature of the proxy approach, as growing the $\alpha$-tree is based on groups learned at the decision tree leaves *but* the CVAR computation still relies on the *original* sensitive grouping.

Figs. 14 and 15 (top) presents the RF TOPDOWN proxy sensitive attribute results of the German Credit and Bank datasets. The ACS and Bank experiments presented here are similar to that presented in the main text. For German Credit, similar degradation in CVAR in the non-proxy case can be seen for TOPDOWN results using proxy attributes.

When comparing to the MLP variants (Figs. 14 to 16 bottom), results are quite similar with slight increases in CVAR from the change in black-box. One notable difference can be seen in Fig. 16. In particular, the proxy and regular curves do not "cross". This indicates that (given that the sensitive attribute proxy used is the same as RF) the black-box being post-processed is an important consideration in the use of proxies. In particular, as RF has a higher / worse initial CVAR, which is highly tied to the loss / cross entropy of the black-box, the robustness of the black-box needs to be considered.

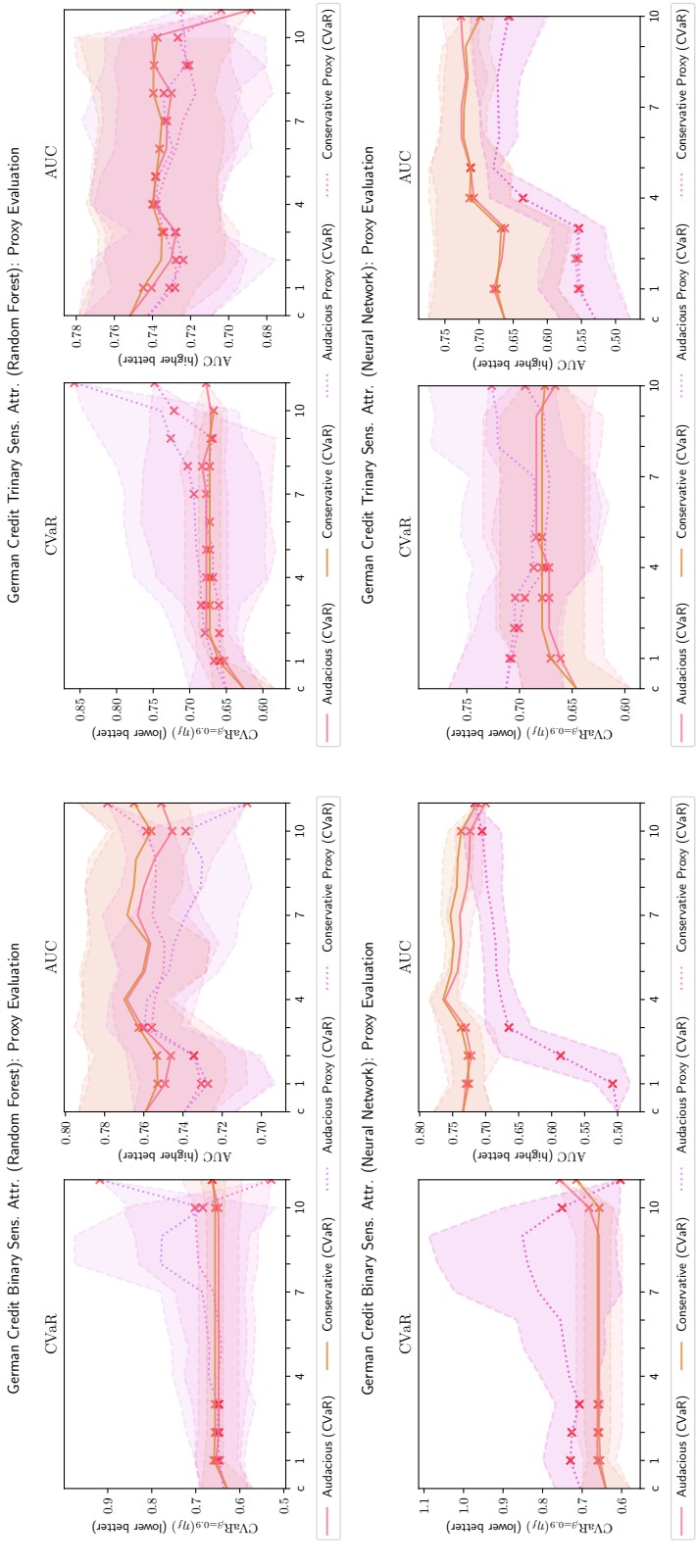

Figure 14: RF (top) and MLP (bottom) evaluation of replacing sensitive attributes with a proxy decision tree on the German Credit datasets.

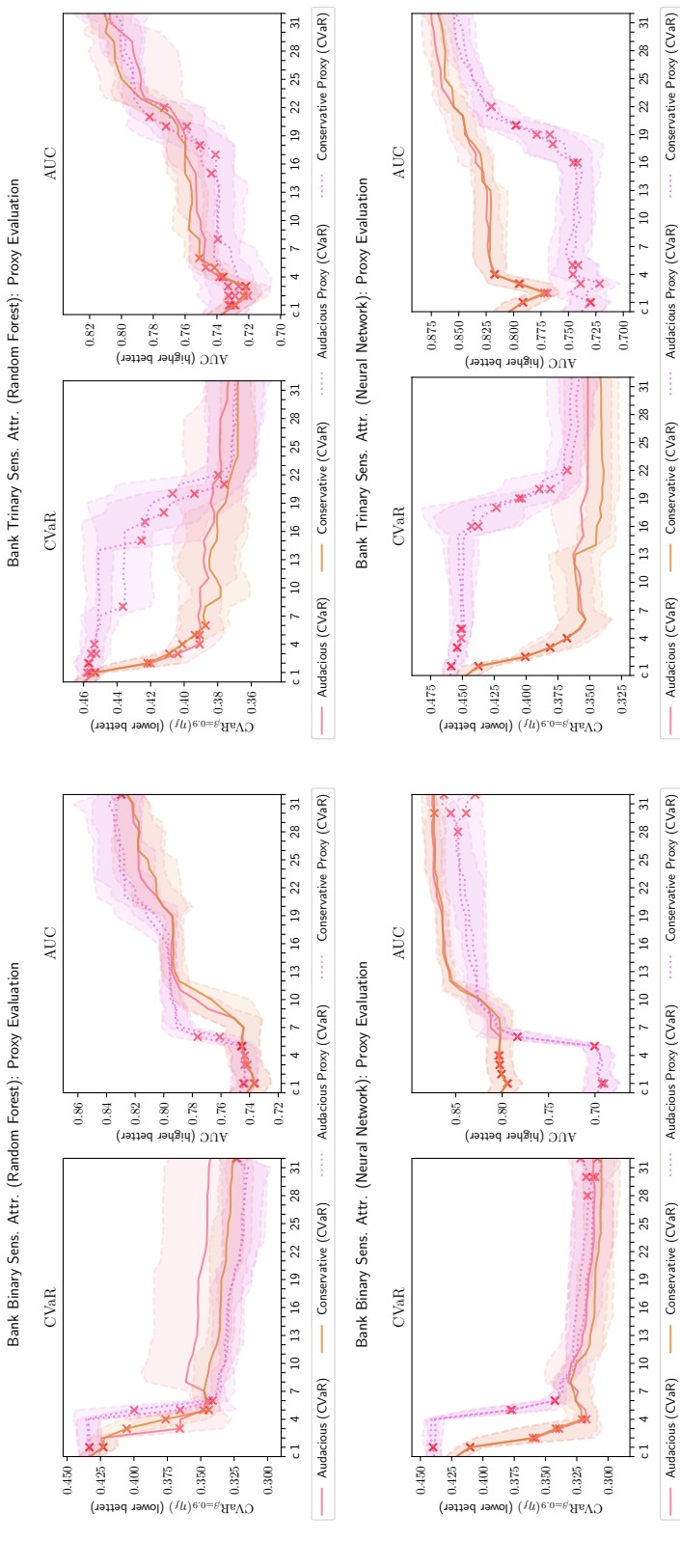

Figure 15: RF (top) and MLP (bottom) evaluation of replacing sensitive attributes with a proxy decision tree on the Bank datasets.

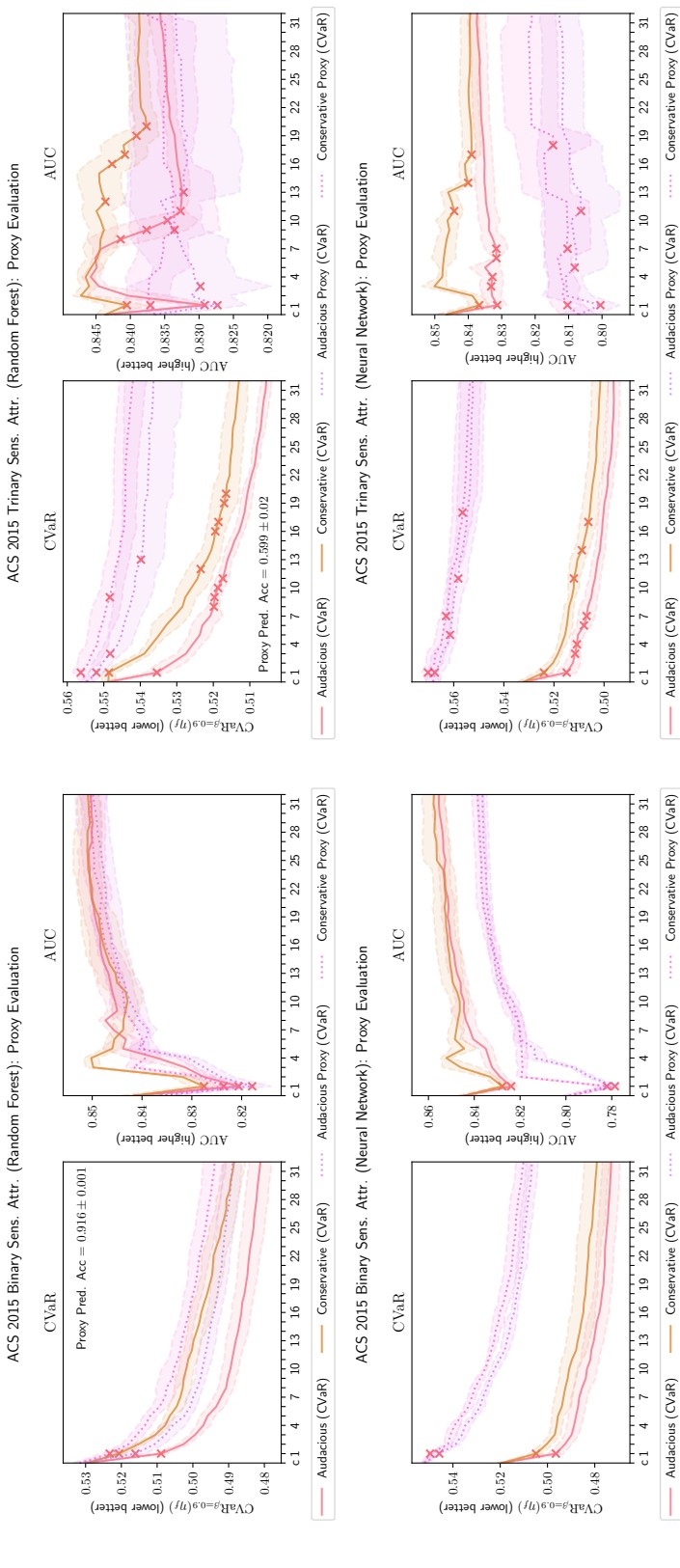

Figure 16: RF (top) and MLP (bottom) evaluation of replacing sensitive attributes with a proxy decision tree on the ACS 2015 datasets.

## XVI   Distribution shift

To examine how TOPDOWN is effected by distribution shift, we train various wrappers over multiple years of the ACS dataset. In particular, we train and evaluate CVAR wrappers over the ACS dataset from years 2015 to 2018. Figs. 17 and 18 report the CVAR values over the multiple years for the random forest (RF) black-box. Figs. 19 and 20 likewise reports corresponding results for neural network (NN) black-boxes.

In particular:

- Fig. 17 presents the conservative update distribution shift evaluation using a RF black-box with $B = 1$ clipping on the ACS dataset over years 2015 to 2018.

- Fig. 18 presents the aggressive update distribution shift evaluation using a RF black-box with $B = 1$ clipping on the ACS dataset over years 2015 to 2018.

- Fig. 19 presents the conservative update distribution shift evaluation using a MLP black-box with $B = 1$ clipping on the ACS dataset over years 2015 to 2018.

- Fig. 20 presents the aggressive update distribution shift evaluation using a MLP black-box with $B = 1$ clipping on the ACS dataset over years 2015 to 2018.

As the ACS dataset consists of census data, one could expect that prior years of the data will be (somewhat) represented in subsequent years of the data. This is further emphasised in the plots, where curves become more closely group together as the training year used to train TOPDOWN increases, *i.e.*, 2018 containing enough example which are indicative of prior years' distributions. Unsurprisingly, we can see that most circumstances the largest decrease in CVAR (mostly) comes from instances where the data matches the evaluation. *i.e.,* the 2015 curve in (top) Fig. 17. Nevertheless, we can see that despite the training data, all evaluation curves decrease from their initial values in all plots; where a slight 'break' in 'monotonicity' occurs in some instances of miss-matching data — most prominently in (top) Fig. 17 for the 2015 plot around 21 boosting iterations. We also remark, perhaps surprisingly, that there is no crossing between curves (*e.g.* as could be expected for the test-2015 and test-2016 curves on training from 2016's data in Figure 17), but if test-2015 remains best, we also remark that it does become slightly worse for train-2016 while test-2016 expectedly improves with train-2016 compared to train-2015. Ultimately, all test-* curves converge to a 'midway baseline' on train-2018.

In general, there is little change when comparing the two different black-boxes. The only consist pattern in comparison is that the NN approaches start and end with a smaller CVAR value than their RF counter parts. When comparing binary versus trinary results, there is a distinct larger spread between evaluation curves (between each year within a plot) for the trinary counterparts. This is expected as in the trinary sensitive attribute modality, CVAR is sensitive to additional partitions of the dataset. The spread is further strengthened as the final $\alpha$-tree in TOPDOWN often does not provide an $\alpha$-correction for all subgroups, *i.e.*, at least one subgroup is not changed by the $\alpha$-tree with $\alpha = 1$. When comparing conservative versus aggressive approaches, it can also be seen that there is a larger spread between evaluation curves for the aggressive variant.

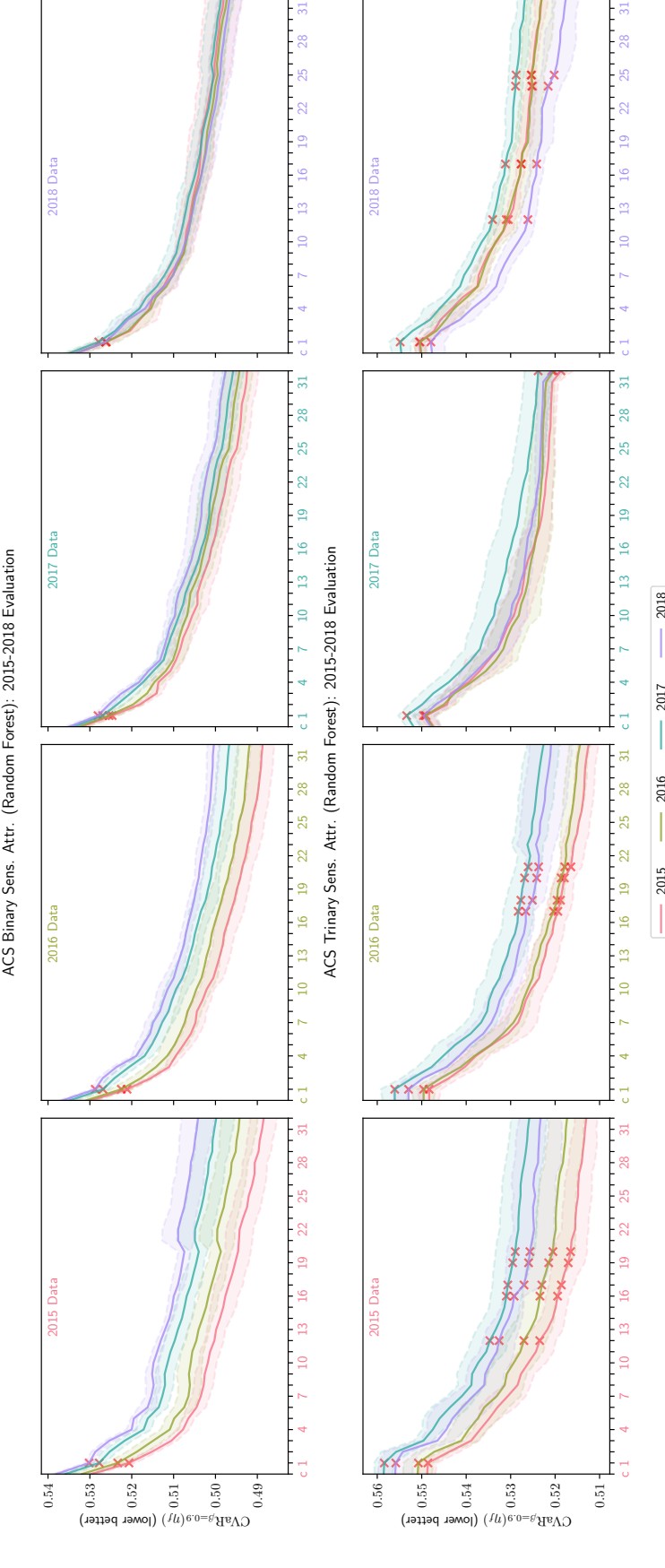

Figure 17: Random forest black-box conservative CVAR wrapper trained for ACS 2015 to 2018 datasets Each plot is trained on a different dataset year. Each curve colour, indicates the data being used to evaluate the wrapper.

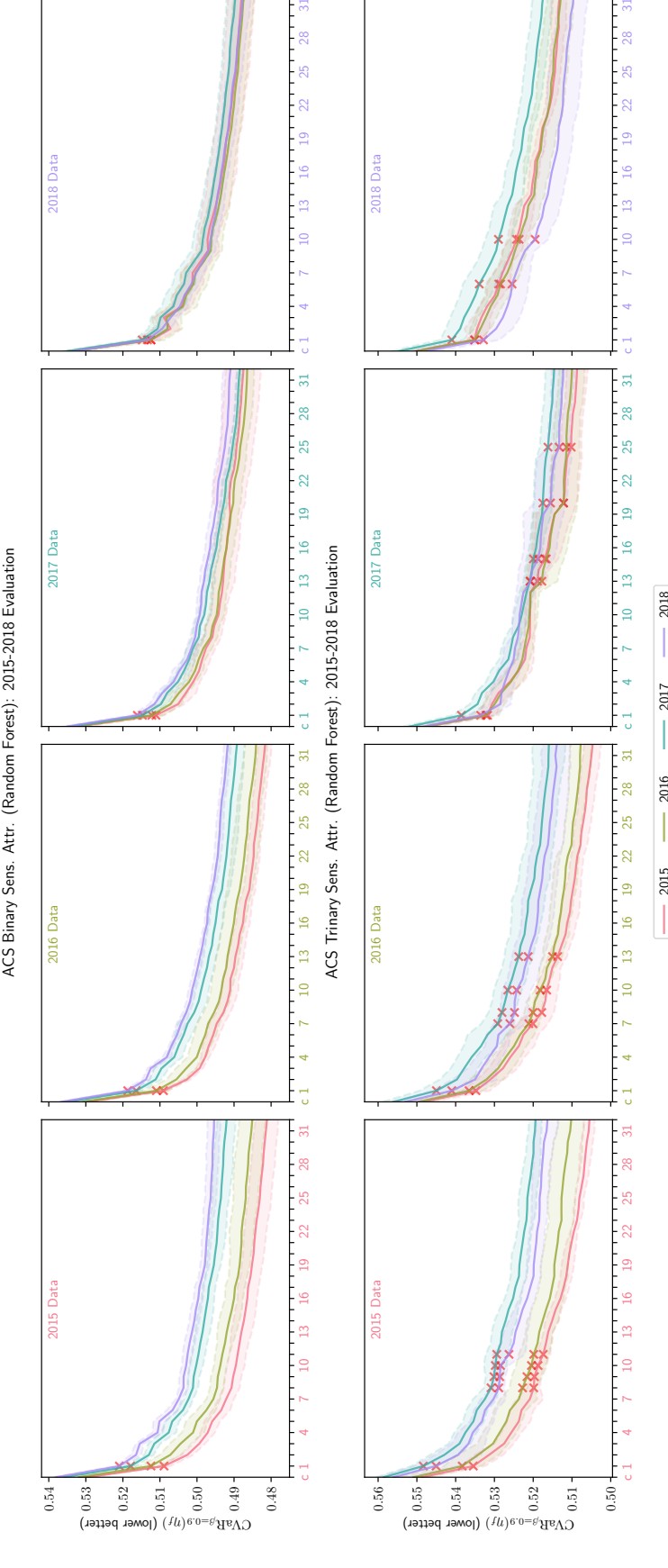

Figure 18: Random forest black-box aggressive CVAR wrapper trained for ACS 2015 to 2018 datasets Each plot is trained on a different dataset year. Each curve colour, indicates the data being used to evaluate the wrapper.

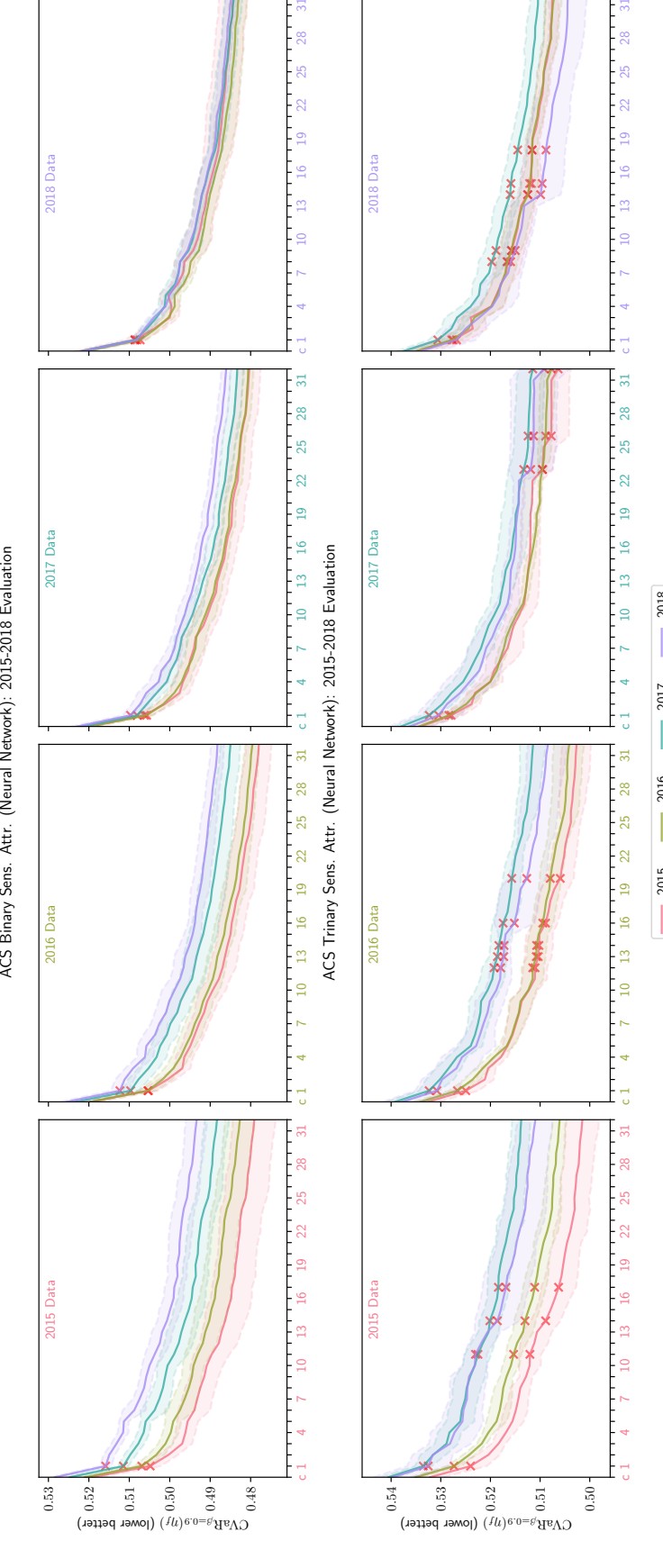

Figure 19: Neural Network black-box conservative CVaR wrapper trained for ACS 2015 to 2018 datasets Each plot is trained on a different dataset year. Each curve colour, indicates the data being used to evaluate the wrapper.

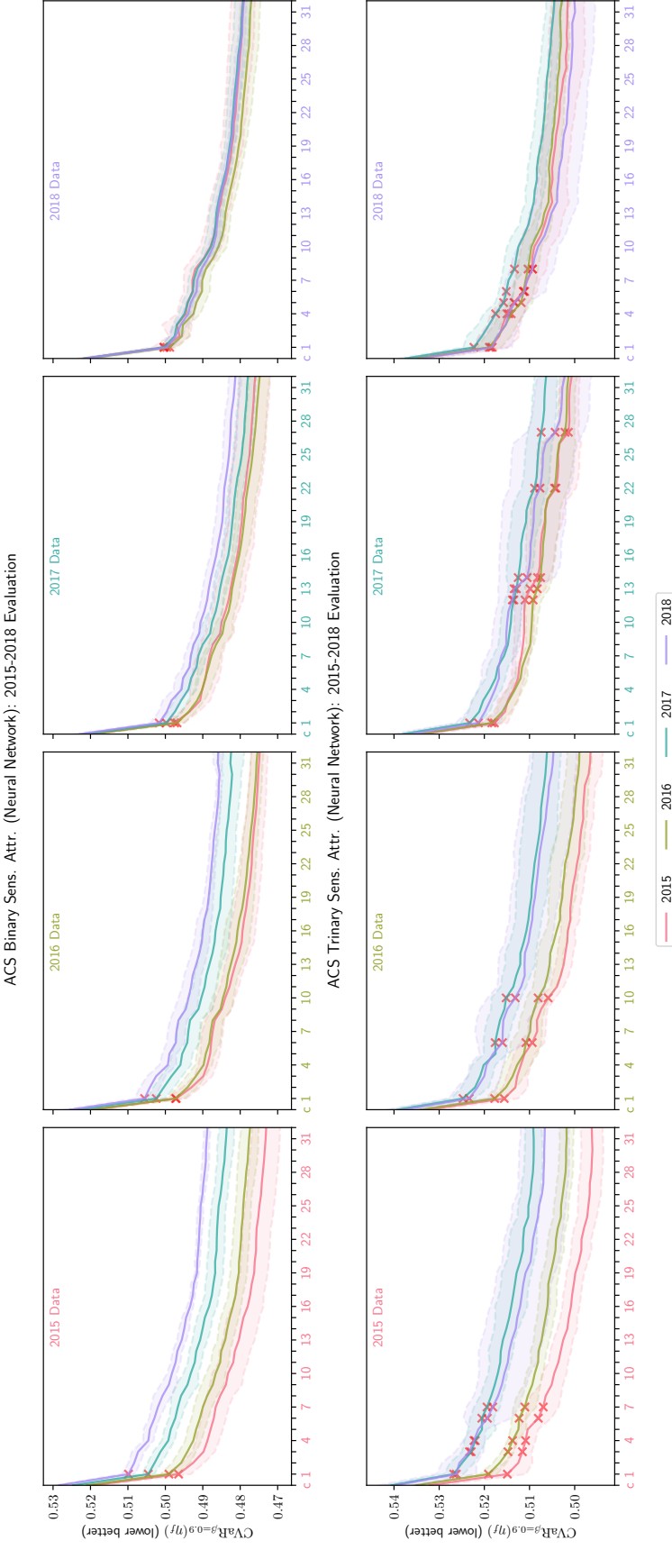

Figure 20: Neural Network black-box aggressive CVAR wrapper trained for ACS 2015 to 2018 datasets Each plot is trained on a different dataset year. Each curve colour, indicates the data being used to evaluate the wrapper.

## XVII    High Clip Value

In this section, we consider a higher clipping value than that used in other experiments. In other sections, we consider a $B = 1$ clipping value which results in posterior restricted between roughly $[0.27, 0.73]$. Although this clipping seems harsh, from the prior experiments one can see that TOPDOWN provides a lot of improvement across all fairness criterion (and we will see $B = 1$ allows TOPDOWN to improve beyond optimization for a large clip value).

We will now consider TOPDOWN experiments which correspond to evaluation over CVAR, EOO, and SP criterion with clipping $B = 3$ (as discussed in theory sections of the main text). This restricts the posterior to be between roughly $[0.05, 0.95]$. Figs. 21 to 23 presents RF plots over German, Bank, and ACS datasets; and Figs. 24 to 26 presents equivalent MLP plots.

In particular:

- Fig. 21 presents the evaluation using a RF black-box with $B = 3$ clipping on the German Credit dataset.
- Fig. 22 presents the evaluation using a RF black-box with $B = 3$ clipping on the Bank dataset.
- Fig. 23 presents the evaluation using a RF black-box with $B = 3$ clipping on the ACS dataset.
- Fig. 24 presents the evaluation using a NN black-box with $B = 3$ clipping on the German Credit dataset.
- Fig. 25 presents the evaluation using a NN black-box with $B = 3$ clipping on the Bank dataset.
- Fig. 26 presents the evaluation using a NN black-box with $B = 3$ clipping on the ACS dataset.

In general, there is only a slight difference between the RF and MLP plots in this clipping setting.

We focus on the RF ACS plot of the higher clipping value, Fig. 23. The most striking issue is that the minimization of CVAR is a lot worse than when using clipping $B = 1$. In particular, BBOX (which in Fig. 23 has $B = 3$) is not beaten by the final wrapped classifier produced by either update of TOPDOWN. However, for EOO and SP there is still a reduction in criterion, although a lower reduction for some cases, *i.e.*, conservative EOO. It is unsurprising that CVAR is more difficult to optimize in this case as the black-box would be closer to an optimal accuracy / cross-entropy value without larger clipping. As a result, CVAR would be more difficult to improve on as it depends on subgroup / partition cross-entropy. In particular, the large spike in the first iteration of boosting is striking. This comes from the fact that we are no directly minimizing a partition's cross-entropy directly, but an upper-bound, where the theory specifies that the upper-bound requires that the original black-box is already an $\alpha$-tree with correct corrections. However, as the the original black-box is not an $\alpha$-tree with correction specified by the update, the initial update can cause an increase in the CVAR (which appears to be more common with higher clipping values).

Despite the initial "jump" and in-ability to recover, let us compare the $B = 3$ plot to the original $B = 1$ RF TOPDOWN plot given in Fig. 10. From comparing the results, one can see that the final boosting iteration for the $B = 1$ aggressive updates beats the $B = 3$ black-box classifiers. Thus, even when comparing against CVAR which is highly influenced by accuracy (thus a higher clipping value is desired), a smaller clipping value resulting in a more clipped black-box posterior is potentially more useful in CVAR TOPDOWN. If one looks at the conservative curves in Fig. 10, these do not beat the $B = 3$ black-box. This further strengthens the argument that the aggressive update is preferred in CVAR TOPDOWN; and is further emphasized by the increase cap between curves with $B = 3$ black-boxes, as shown in Fig. 23.

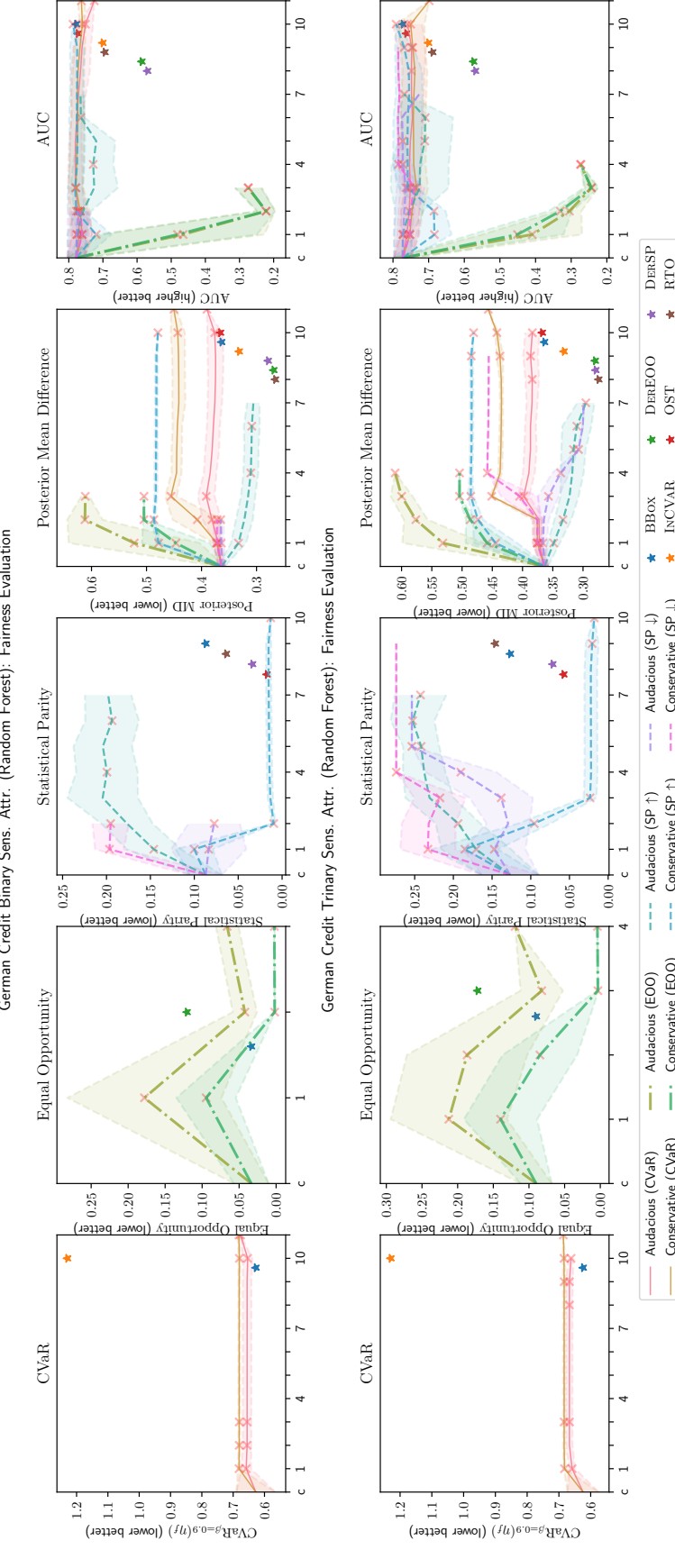

Figure 21: RF with $B = 3$ TopDown optimized for different fairness models evaluated on German Credit with binary (up) and trinary (down) sensitive attributes. Crosses denote when a subgroup's $\alpha$-tree is initiated (over any fold). The shade depicts $\pm$ a standard deviation from the mean. However, this disappears in the case where other folds stop early.

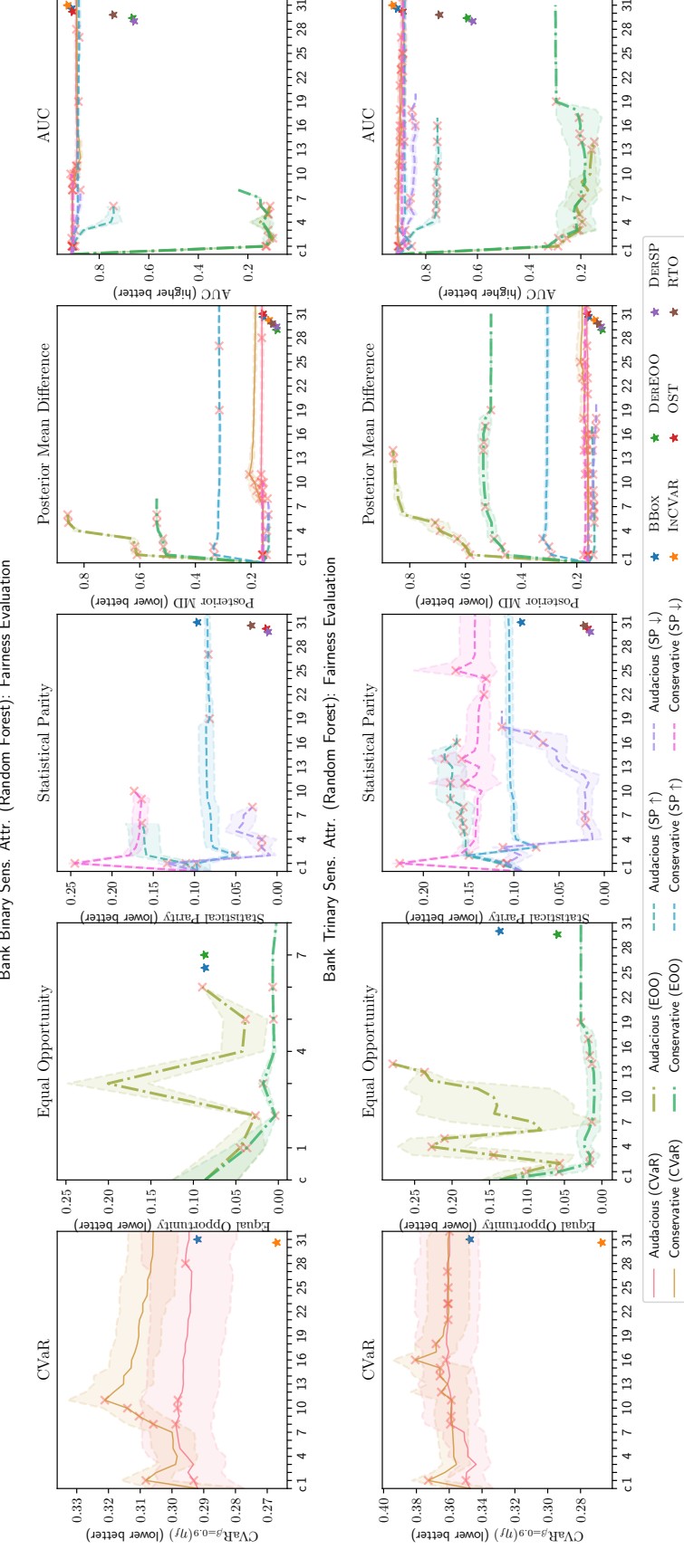

Figure 22: RF with $B = 3$ TopDown optimized for different fairness models evaluated on Bank with binary (up) and trinary (down) sensitive attributes. Crosses denote when a subgroup's $\alpha$-tree is initiated (over any fold). The shade depicts $\pm$ a standard deviation from the mean. However, this disappears in the case where other folds stop early.

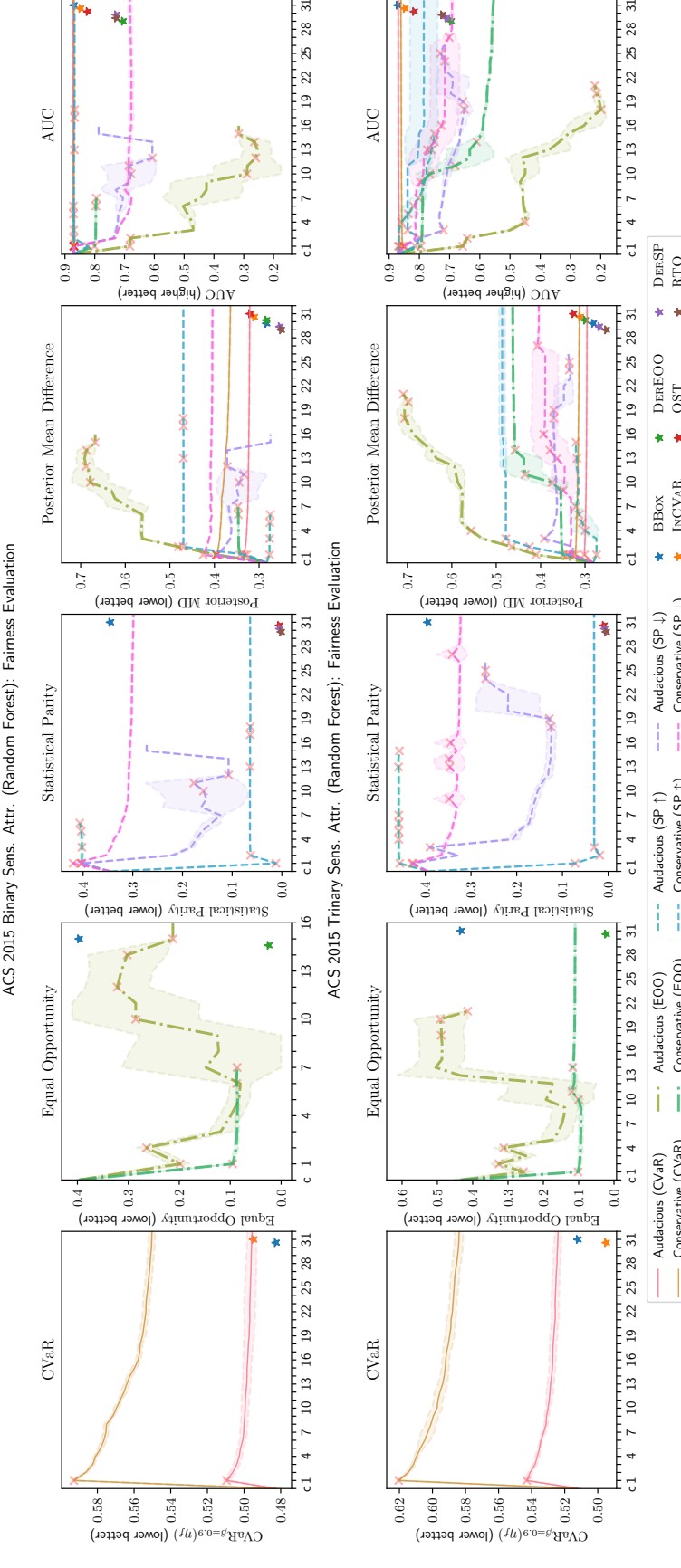

Figure 23: RF with $B = 3$ TOPDOWN optimized for different fairness models evaluated on Bank with binary (up) and trinary (down) sensitive attributes. Crosses denote when a subgroup's $\alpha$-tree is initiated (over any fold). The shade depicts $\pm$ a standard deviation from the mean. However, this disappears in the case where other folds stop early.

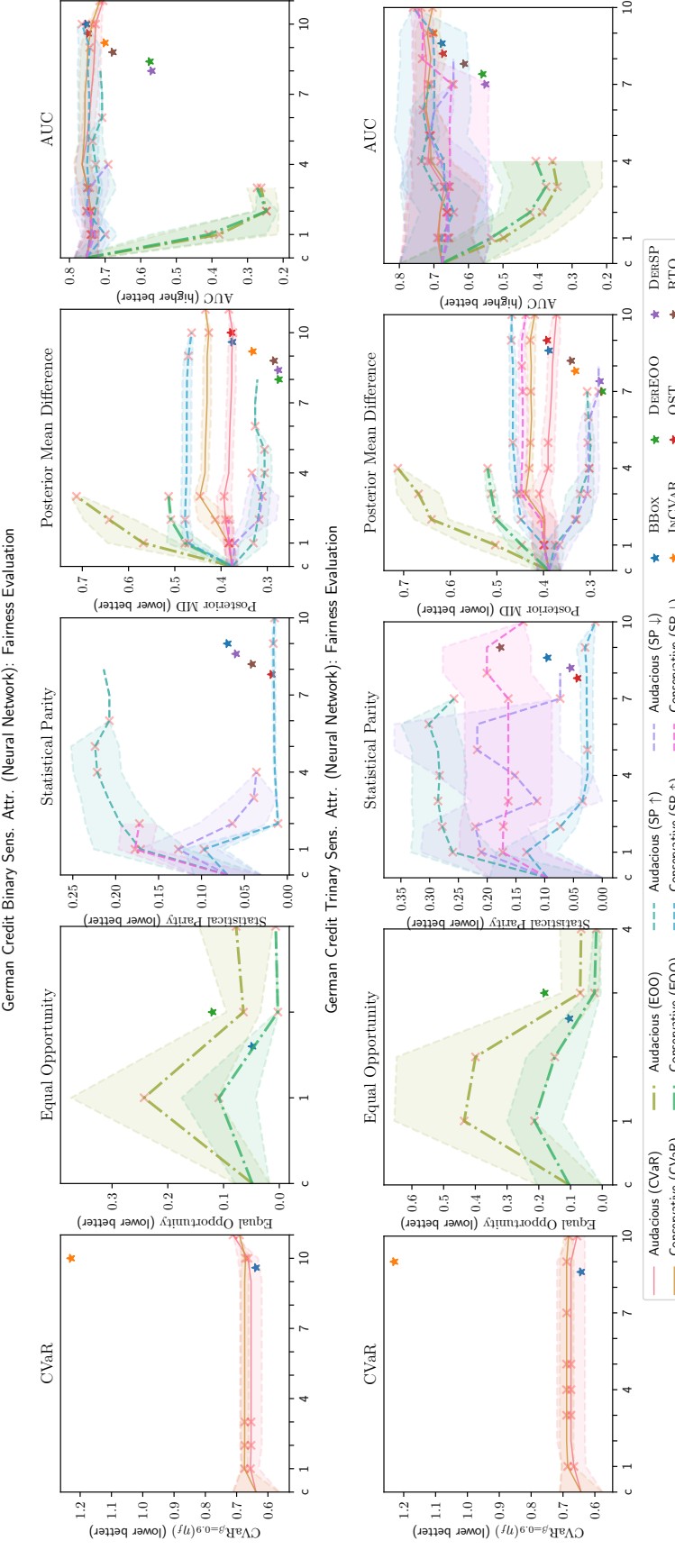

Figure 24: MLP with $B = 3$ TopDown optimized for different fairness models evaluated on German Credit with binary (up) and trinary (down) sensitive attributes. Crosses denote when a subgroup's $\alpha$-tree is initiated (over any fold). The shade depicts $\pm$ a standard deviation from the mean. However, this disappears in the case where other folds stop early.

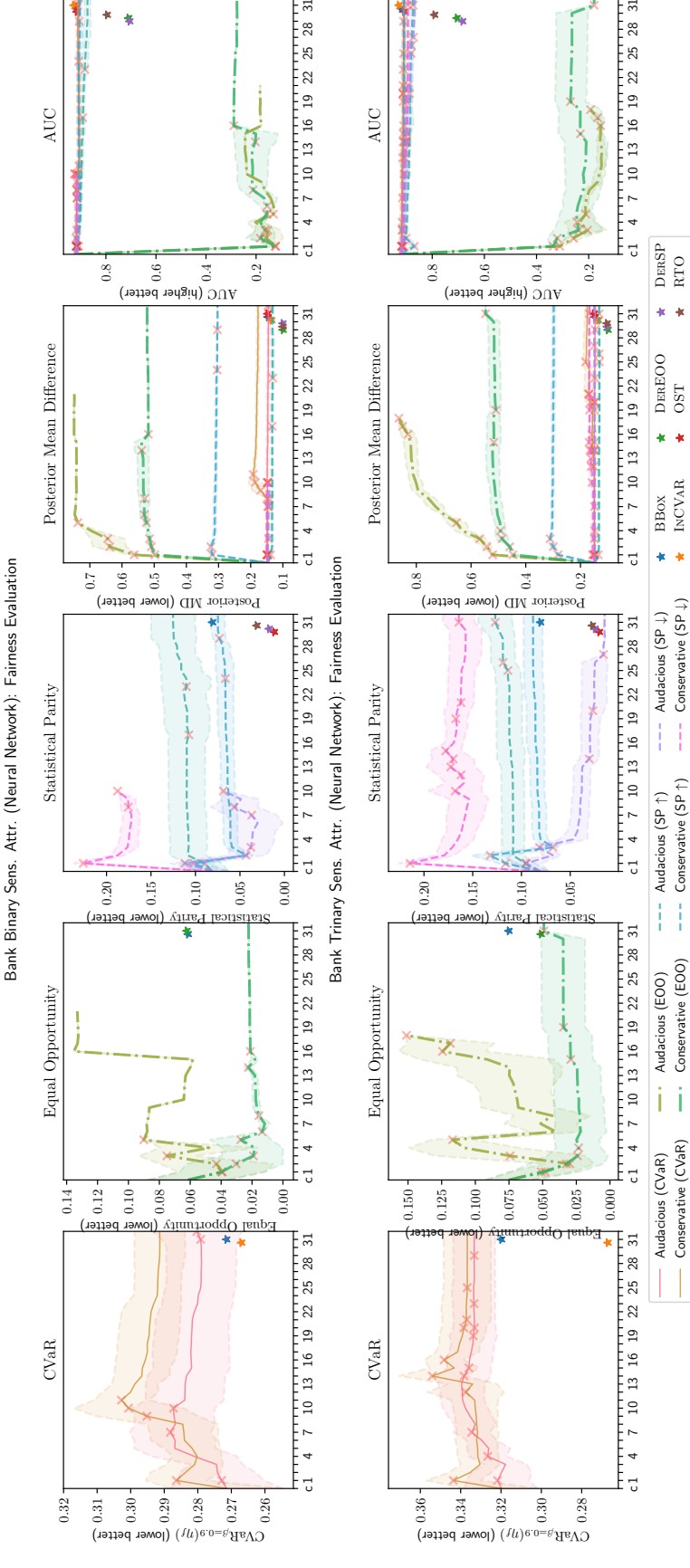

Figure 25: MLP with $B = 3$ TopDown optimized for different fairness models evaluated on Bank with binary (up) and trinary (down) sensitive attributes. Crosses denote when a subgroup's $\alpha$-tree is initiated (over any fold). The shade depicts $\pm$ a standard deviation from the mean. However, this disappears in the case where other folds stop early.

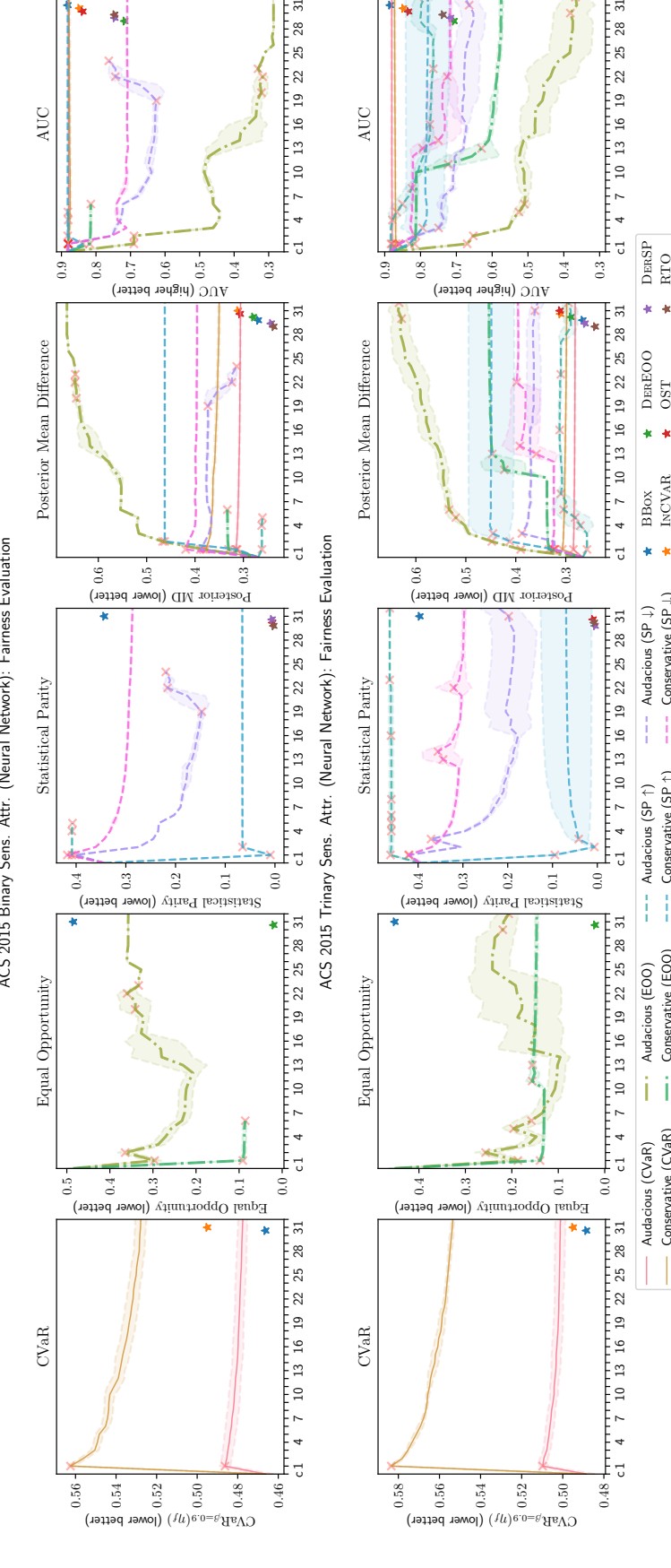

Figure 26: MLP with $B = 3$ TOPDOWN optimized for different fairness models evaluated on Bank with binary (up) and trinary (down) sensitive attributes. Crosses denote when a subgroup's $\alpha$-tree is initiated (over any fold). The shade depicts $\pm$ a standard deviation from the mean. However, this disappears in the case where other folds stop early.

## XVIII Example Alpha-Tree

In this section, we provide an example of an $\alpha$-tree generated using TOPDOWN. In particular, we look at one example from training CVAR TOPDOWN on the Bank dataset with binary sensitive attributes. Fig. 27 presents the example $\alpha$-tree. The tree contains information about the attributes in which splits are made and the $\alpha$-correction made at leaf nodes (and their induced partition). In the example, could note that the $\alpha$ trees for modalities of the age sensitive attribute are imbalanced. The right tree is significantly smaller than the left. One could also note the high reliance on "education" based attributes for determining partitions. These factors could be used to scrutinise the original blackbox; and eventually, even provide constraints on the growth of an $\alpha$-tree which would aim to avoid certain combinations of attribute. We leave these factors for future work.

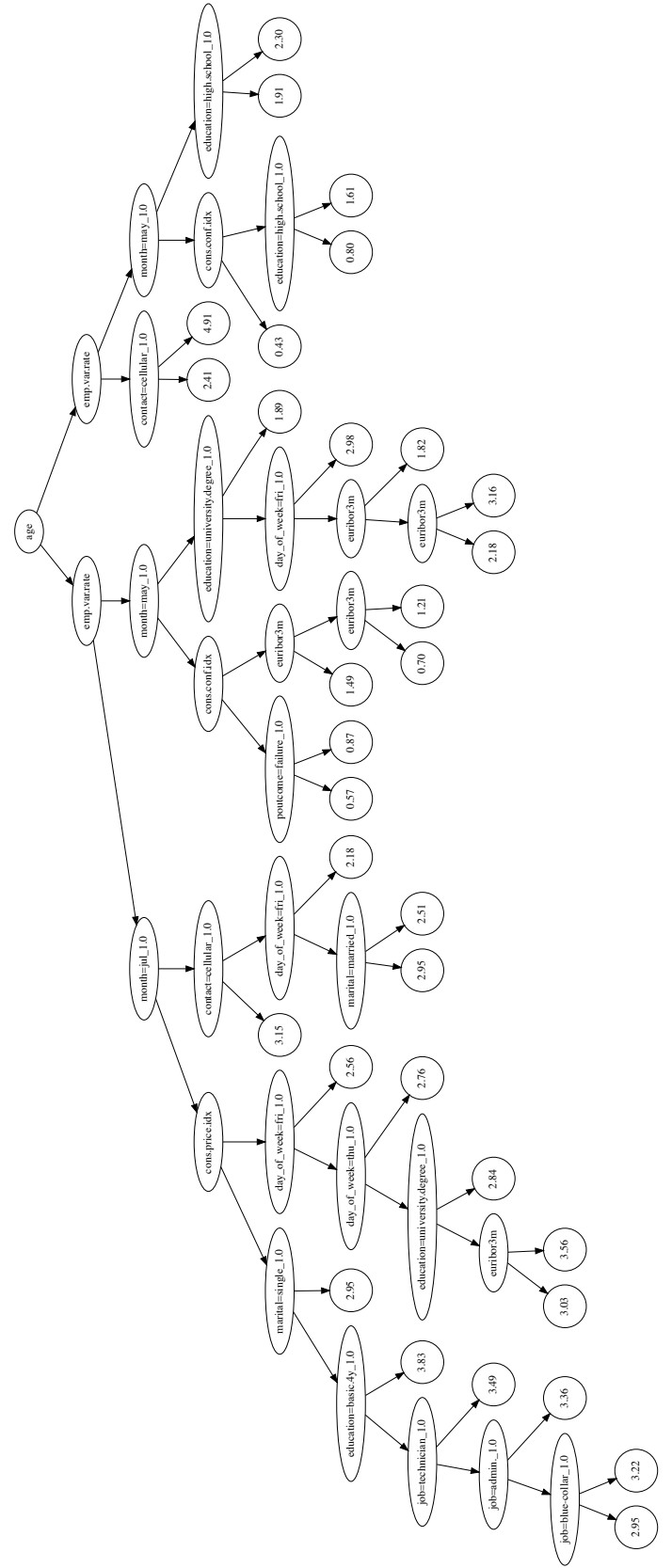

Figure 27: Example tree generated in the optimization of TopDown for CVaR in the Bank dataset with binary sensitive attributes.