# OpenReview forum: "Fair Wrapping for Black-box Predictions"
_NeurIPS.cc/2022/Conference — NeurIPS 2022 Accept_

### Official Review · Reviewer_4eKb · 2022-07-10

**Rating:** 4
**Confidence:** 3
**Soundness:** 4 excellent
**Presentation:** 2 fair
**Contribution:** 2 fair

**Summary:**

The paper proposes a way to correct black-box models to fulfill certain notions of group fairness via post-processing.
Concretely, the authors suggest to reweigh the posterior distribution of a classifier via a learned decision tree.
To this end, the potential bias of a black-box model is cast as a so-called twist. By considering a well-constructed loss, an untwisting function can be learned by minimizing this loss. The authors instantiate this for decision tree learning and three different notions of group fairness (conditional value at risk, equal opportunity, and statistical parity).
Two variants of their decision tree learning algorithm,  a conservative one, which sacrifices less accuracy to achieve fairness, and an audacious one, which allows larger changes in favor of fairness but potentially costing accuracy, are considered.

This is experimentally evaluated on tabular datasets. The proposed approach outperforms different baselines in terms of fairness notion.
Interestingly, empirically it appears that the audacious and conservative versions don't exhibit the trade-off discussed above, but depending on the setting either one is strictly better (both in terms of fairness metric and accuracy).



**Questions:**

- Can you clarify which parts are your key contributions, especially compared to [24] and [14]?
- What does MD (in Figure 4 and Section 8) measure? Is it the accuracy? If so, it seems bad for a binary task.



**Limitations:**

The authors provide a Limitation section, which seems to be adequate.


**Strengths And Weaknesses:**


## Strengths
- Mathematically rigorous derivation.
- Important/interesting use-case: e.g. finetuning a blackbox model to ensure fairness.

## Weaknesses
- Hard to read. (See "Presentation" below.)
- Relation to some related work is unclear. (See "Related" below.)

## Presentation & Evaluation
While I appreciate the mathematical rigor I found the overall work hard to read.
In particular, I think the mathematical presentation almost hides the conceptual approach: i) get blackbox model, ii) train decision tree to debias the posterior. While the choice of loss to train the decision tree is theoretically well-motivated, in the end, it appears quite standard.
Similarly, it remains unclear what the exact contribution and what an instantiation of a prior approach is, e.g. TopDown from [14]. (Note: I don't think instantiating prior work to reach a particular goal is problematic.)

Further, on the first read, it was unclear to me that section 3 recounts background from [24] rather than a new contribution. At the same time, it appears not to be self-contained as binary task is taken from [24] without much explanation.

I also found the evaluation section hard to read as all results are pressed in Figure 4. The selection of results is also quite small (although I appreciated the full additional results in the supplementary material). Lastly, it seems that the accuracy of the classifier is quite bad (see Questions below).

Minor Presentation issues:
- L240 "of mapping" seems to miss an article.
- "Limitations and conclusion" (L335) does not have a section number.
- While normally I don't mind if papers use modified vspace settings, it seems a little overdone here to the point where pages aver visually very dense.

## Related Work
The paper does not discuss to which fairness notions it is limited to, i.e. group fairness, and how this relates to other notions of fairness, i.e. individual fairness, and how they are applied in the research community.

---

> ### Author Response · Authors · 2022-08-02
> **Response to 4eKb**
>
> We thank the reviewer for the thoughtful comments. We apologize for the compactness of the presentation. The points on clarity addressed in the general remarks will also help elucidate the underlying concepts of our work. For the other minor points made, we will make sure to address them in the next version of the paper. For the other points please see the below.
>
> ---
>
> **Prior work and contributions**
>
>
> We clarify that [24->25] explores a “setting” for proper loss functions for binary classification / class probability estimation. We utilize their notation and language as a “jumping point” for rigorously analyzing fair classification.
>
> [14->15] proves boosting convergence bound for decision tree classification. We present a novel generalization of [14->15] which considers the post-processing of a blackbox classifier. The primary technical step is moving from predicting labels to fixing a blackbox. One significant difference that needed to be done was moving from reasoning about the purity of nodes in a decision tree (the positive class balance at a node) to instead considering the “alignment” of the blackbox with respect to data (Eq 13).
>
> ---
>
> **MD and Accuracy**
>
> Posterior mean difference is the error rate / 0-1 loss. A note about the accuracy: we made no attempts at fine tuning the hyperparameters of the initial blackbox. The blackbox is clipped to adhere to Assumption 1, which does harm accuracy, but allows for improvements for fairness. For a “higher” accuracy / “lower clipping” regime see SI, Section XVII. Here we find that our approach can still provide fairness (EO / SP, where CVaR does better in both fairness and accuracy if the blackbox is clipped more initially). However, one needs to be more careful about large accuracy decreases and thus “prune” the $ \alpha $-tree.
>
> ---
>
> **Other notions of fairness**
>
> We have added clarification and noted that the paper only considers group notions of fairness.

---

> > ### Comment · Reviewer_4eKb · 2022-08-07
> > **Thank you**
> >
> > I appreciate the presentation updates and MD/Accuracy clarification.
> > Based on this I have updated my presentation score and overall assessment.
> >
> > Best,
> > Reviewer 4eKb

---

### Official Review · Reviewer_2a51 · 2022-07-12

**Rating:** 6
**Confidence:** 1
**Soundness:** 3 good
**Presentation:** 3 good
**Contribution:** 3 good

**Summary:**

This paper proposes a wrapper function in the post-processing to correct the classifier for fairness results. It adopts a previous paper that modifies the improper loss function to make it fair. The proposed boosting algorithm, called $\alpha$-tree, corrects for fairness bias for three different notions: Conditional value at risk, Equality of opportunity, and Statistical Parity.

**Questions:**

I'm wondering why this algorithm is claimed to be interpretable? The only thing related is this method uses a tree-based model to correct for the bias. But an algorithm consisting of multiple trees like boosting or random forests can be equally complex as a DNN and thus should not be claimed interpretable.

**Strengths And Weaknesses:**

Disclaimer: I'm not an expert in fairness and the improper loss function and my expertise are in interpretability. I can not evaluate the theoretical section due to my incompetence, but this paper is mostly theoretical so my review will be quite shallow.

# Strengths
- Although the empirical performance is not as good as other methods, this wrapper method seems to be the first one that can adapt to different fairness criteria in a post-hoc way. And the conclusion also admits that.

# Weaknesses
- Questionable claim about interpretability.

---

> ### Author Response · Authors · 2022-08-02
> **Response to 2a51**
>
> We thank the reviewer for their comments and noting the flexibility of our approach. Please see the general remark regarding interpretability. Just as a short summary: we will make sure to make it more clear that the interpretability mentioned in the paper corresponds to the post-processing correction. This is only possible because the “architecture” of learning $\alpha(\cdot)$ is a simple $\alpha$-tree.

---

### Official Review · Reviewer_tmha · 2022-07-21

**Rating:** 4
**Confidence:** 3
**Soundness:** 3 good
**Presentation:** 2 fair
**Contribution:** 2 fair

**Summary:**

The paper proposes a post-processing mechanism to wrap black-box predictors such that their predictions become compatible with an ancillary criterion such as fairness. To this end, the authors propose to use so-called "$\alpha$-trees", an adaptation of decision trees, to learn correction coefficients that are used to change the output probabilities of a black-box model in the desired way. They further introduce adaptations of their algorithm that enable wrappers that satisfy fairness notions such as conditional value at risk, equality of opportunity and statistical parity.
To demonstrate the efficacy of the proposed approach, the paper derives a number of bounds, for instance on the divergence of the wrapped predictor from it's black-box source as well as on the amount of unfairness in the adjusted outputs. In addition, empirical results are provided on commonly used datasets.

**Questions:**

- The analysis, in particular also the methodology to achieve fair outputs, relies on $\eta^\ast$, which in section 3, line 85 is defined as the true posterior, approximated from data. What does that mean? Using which approximation scheme? How is it related to some potential other, ground truth posterior? And most importantly, how is the assumption of having access to a true posterior realistic in practice? Without this quantity the fairness constraints in section 6 do not work. There are experiments in section 8, so there must be some way to approximate $\eta^\ast$, but I don't know how it's done (though there seem to be limitations with small datasets).
- The proposed method for achieving fairness is based on splitting the $\alpha$-trees at their root based on the different sensitive feature values. This means that data cannot be shared between the different sub-trees corresponding to different groups. Could this lead to issues in cases where some sub-groups are minorities and thus, insufficient training data for them is available? The minority-setting can be an issue in many scenarios where fairness is of concern.
- Some concepts are not clear:
    - The paper uses the term "twisted" posterior, but it's not fully clear to me what the characteristic property of a twist is and whether it just means "bias" in the conventional way used in the fairness literature, or whether there is some additional meaning to it. Could you clarify that?
    - What is the mixture measure M introduced in section 3 and why do you need it?
    - Regarding requirements (c) and (d), what do you mean by strong algorithmic guarantees (on what?) and what is meant by explainability, e.g. instance-based or for the whole model, about the difference of $\eta_f$ from $\eta_u$ or $\eta_f$'s behavior in general, etc.?
- Minor clarifications:
    - Does theorem 2 imply that the loss is 0 when the conditions hold?
    - It would be good to define the domain of $\alpha$ in section 3. Not all values are permissible and I struggled to understand whether one would generally choose it from e.g. (0, 1), (-1, 1) (with exception of 0), $(-\infty, +\infty)$, etc.
- Potential extension (I do not expect this to be included in this paper): the proposed method can be used to correct twists/biases in black-box predictions. It might be interesting to explore whether the technique can be used to tackle issues other than fairness as well, for example in the context of robustness, to correct for distribution shifts between training and deployment.

**Limitations:**

The main practical limitations I can see are the assumption of the availability of a true posterior and the problem having sufficient data for learning $\alpha$-trees for minority groups. These issues are insufficiently addressed in the paper.

**Strengths And Weaknesses:**

### Strengths
- The paper proposes a method to adjust the outputs of any black-box model to be compatible with a desired fairness goal. Having this ability is quite valuable given that the usage of pre-trained, off-the-shelf models is quite prevalent today which, however, may not be trained with any fairness considerations in mind.
- The paper provides a solid theoretical analysis of the guarantees obtainable from the proposed algorithm along several dimensions, such as the divergence of the wrapped model from its source and the amount of residual unfairness in the wrapped predictions.

### Weaknesses
- The paper has a number of clarity issues:
    - The intuition behind many definitions and results are not clear:
        - Some results (e.g. theorem 1 and 2, lemma 3) propose bounds, but I have no intuition about what they mean. How strict are they? How strict are they relative to the strength of the assumptions made? This makes it hard to understand whether they are meaningful.
        - Some definitions are unclear, see the questions part for more details.
        - A lot of the analysis starts off by introducing some definitions or statements whose meaning only becomes clear later on. It would be very helpful to first provide an intuition, e.g. say what the goal of introducing a certain concept is and what idea it captures, for better understanding the analysis.
    - In the intro, a number of desiderata for the post-processing process are introduced (a-f, and later g). While they mostly make sense, some (esp. c and d) are somewhat vague (see questions) and I am not sure what is meant by e.g. addressing them from a representation point of view and what the difference between the analytical and algorithmic perspective is. It feels like the points are also not addressed in a systematic manner, e.g. in section 4 (b) and (e,f) are mentioned and another point (g) is introduced, in section 5 (c) and (d, f) are mentioned. This scattering, in combination with the unexpressive names of the requirements makes it hard for me to keep track of them and make them overall not very useful. It would be helpful to formulate the requirements more clearly, address them in a more systematic way, and give them more meaningful names.
    - The paper introduces a lot of concepts, esp. in section 3, which subsequently seem to be under-used. For instance, section 3 sets up core concepts using measure spaces, but then this level of abstraction seems to be unnecessary for the remaining paper. It appears like the exposition could be simplified by using more standard and simpler probabilistic notation instead. I did not check all the proofs in the appendix though, maybe this level of abstraction is needed there, but it comes at a high cost in terms of notational complexity which seems not immediately justified.
    - Some concepts are introduced early (e.g. in section 3), but only instantiated a lot later in the paper, which makes it hard to understand them both when they first appear as well as when they are properly defined/used, because you need to go back and re-read their definitions. For example, $\eta_t$ is introduced in section 4, but not really instantiated until section 6 and similarly for mixture measure M in section 3. It would be great to provide at least some intuition behind those concepts when first introducing them and ideally moving their introductions and instantiations closer together.
- The implications of some of the theoretical results are unclear, i.e. what do the bounds mean? See clarity above for more details. This makes it hard to judge how useful the derived bounds are.
- Some assumptions might be unrealistic for practical deployment, most importantly the assumption of having access to a true posterior $\eta^\ast$. See the first question for more details.

---

> ### Author Response · Authors · 2022-08-02
> **Response to tmha (part 1)**
>
> Thank you for your detailed comments. Please see below for our response to your concerns.
>
> ---
>
> **Additional points on clarity**
>
> Please see our general response on clarity and presentation for our response regarding the mathematical abstraction and instantiation / ordering of concepts.
> With respect to the general remarks, have added intuitive comments in the introduction of the measure space notation. Note: the utilization of such a notation has become quite standard for proper loss functions [24 -> 25]. As such we aimed to utilize this notation to be consistent in that literature.
> We address the other concerns outlined by the reviewer.
>
> We thank the reviewer for their suggestions regarding the description of the desiderata we have listed. The following grouping, ordering, renaming, and clarification aims to alleviate the review’s concerns:
>
> __”Analytical Choice”__
> These are desiderata which are covered by utilizing the $\alpha$-loss for correct fairness, but not the specific “architecture” choice of $ \alpha(\cdot)$ nor the learning / algorithmic process.
> - **(a)** $\rightarrow$ **(Flexibility)**: the ability of covering multiple (group) fairness criteria.
> - **(b)** $\rightarrow$ **(Proximity)**: that the final post-processed classifier is still close to the original.
> - **(e)** $\rightarrow$ **(Composability)**: we can compose multiple corrections together.
> - **(g)** $\rightarrow$ **(Invertibility)**: we can easily reverse the correction.
>
> __”Representation Choice”__
> This corresponds to picking what $ \alpha(\cdot) $ is realized as, in our case an $\alpha$-tree.
> - **(d)** $\rightarrow$ **(Explainability)**: the corrections are explainable, ie, the interpretability of the $\alpha$-tree as per the general remarks.
> - **(f)** $\rightarrow$ **(Complexity)**: implications on complexity from the correction (ie, Radamacher complexity for generalization).
>
> __”Algorithmic Choice”
> This corresponds to how we learn the correction.
> - **(c)** $ \rightarrow $ **(Convergence)**: For us, this is specifically boosting compliant convergence.
>
> The grouping, ordering, renaming, and clarification changes will be present in the desiderata in the next version. Like the current paper, we will de-emphasize the (invertibility) property.
>
> ---
>
> **Intuition of theoretical results**
>
> As mentioned in the general response to clarity, we will ensure that the additional explanations are added to help build the reader’s intuition for the theoretical results.
>
> For Theorem 1 the important takeaway is that given the assumption **(S1)**, regardless of the data and original base classifier, the distortion is bounded. We point the reviewer to SI, Section II for a more general result (with weaker assumptions), additional comments, and figure (which was deferred for space) (please note that the region in Figure 5 is mislabeled and should refer to (S2)). The more restrictive result of Theorem 1 is presented for a more intuitive presentation. For the more general Theorem A, a main proof idea is to consider the Taylor expansion of the $\alpha$-corrected posterior about $\alpha=1$. Thus its strictness relates to how far away $ \alpha $ deviates from 1. Also, the RHS in Theorem 1 is a strictly decreasing function of $B$. Disregarding the constraint on the choice of $B$, the RHS vanishes as $B \rightarrow +\infty$, which is tight as it implies $\alpha \rightarrow 1$ almost everywhere (no correction).
>
> Theorem 2 is similar in style to classical decision tree boosting / convergence results. It states that as long as the WLA holds and we have an appropriate amount of leaves, then we can get an arbitrary low loss. Of course the WLA will not hold all the time, but practically we do not aim to boost forever.
>
> Lemma 3 essentially says that if we consider decision trees for our blackbox classifiers, the Rademacher complexity of the post-processed classifier is upper bounded by considering decision trees of their combined (maximum) depth of the original classifier and the $ \alpha $-tree. The interest here is that post-processing through an $ \alpha $-tree can lead to bounded Rademacher complexity through simple structural results, see [3].
>
> We would also want to clarify that in the definition of the $ \alpha $-loss, $ \alpha $ can be any value in $[-\infty, \infty]$, where its corresponding correction is well defined at limit points.
>
> ---
>
> (continue below)

---

> > ### Author Response · Authors · 2022-08-02
> > **Response to tmha (part 2)**
> >
> > **Assumption of true posterior**
> >
> > One of the reviewer’s concerns is regarding the assumption of accessing the “ground truth” posterior $\eta^{\star}$. There are two different versions of approximation needed.
> >
> > For the first, and in most cases, $\eta^{\star}$ is used to calculate expectation quantities which does not require an explicit approximator for $\eta^{\star}$.
> > We utilize $\eta^{\star}$ to make our analysis general with respect to the input distribution, but practically expectation calculated here can be reduced into empirical expectations.
> > For example, as per Section 6, the CVaR criteria takes as “input” the subgroup risk of the (currently) corrected blackbox and $\eta^{\star}$. In this case, the subgroup risk calculation requires $\eta^{\star}$ to be calculated. However, calculating the subgroup risk can just be approximated through an empirical expectation, thus reducing the assumption of $\eta^{\star}$ into requiring a set of samples (for CVaR it would be a finite sum over samples which are a part of the specific subgroups).
> > This empirical estimation can of course be used in any probability calculation with respect to the data distribution (via expectation of indicator functions).
> >
> > A second approximation is required in (only) Definition 5 for the EOO criteria. Here we do require an explicit posterior estimation. Practically, we utilize an out-of-the-box Gaussian Naive Bayes from sklearn (as per submitted code). We apologize for forgetting to include this implementation detail and have clarified this requirement into the updated manuscript.
> >
> > We would also like to note that the requirement of an explicit estimator of $\eta^{\star}$ is only required for our refined approach to EOO. If we did not have access (or did not want to calculate) such an estimator, we can use a more naive approach, similar to that done for SP (see Appendix p28). Thus, if we were in an environment where we only had a small amount of data, we could default to the SP style approach and not compute an explicit approximator of the posterior.
> >
> > ---
> >
> > **Small datasets and alpha-tree initial split**
> >
> > Besides the issue of approximating the posterior in low data environments, the reviewer also commented about the effects of small / imbalanced datasets in our $ \alpha $-tree strategy: splitting / partitioning over sensitive groups resulting in subtrees trained on small amounts of data.
> >
> > Let us first make a general remark regarding small datasets: small data cases can be problematic for our approach, as mentioned in our limitations section (Line 338). We further provide an extensive suite of experiments in the SI, Section XIII to XV. As mentioned in the limitation section, we believe it is important to be as transparent as possible regarding the weakness of approaches in fairness.
> >
> > As such, having a small amount of data at a subtree (from the initial split) could possibly be problematic (ie, when we have small data + large class imbalance). However, suppose we are using the CVaR procedure. In this case, even if there is a small amount of data in one of these subtrees, we will still correct for this small partition if it performs on average worse than other subgroups. The only downside is that it may have poor generalization or harm accuracy.
> >
> > Despite this, one should consider the use case: if we treat post-processing fairness as a fine tuning task, we should make sure that the data quality is good for what we care about. For instance, a company / organization / government may wish to utilize a known publicly available classifier but wish to make sure it is fair for certain subpopulations. In this case, non-public data should be curated / utilized to account for these subpopulations, which should be reflected in a somewhat balanced (or at the least “decently” sized) dataset to post-process / fine-tune the blackbox.
> >
> > ---
> >
> > **”Twisted posterior”**
> >
> >
> > The notion of “twist” comes from [29->30], where they describe the twist as any discrepancy due to noise a (learnt) posterior has when compared to a true posterior. In this work, we describe a “twisted posterior” as a learned posterior which deviates from that of a fair posterior, the twist is the unfairness (with respect to a criteria). However, as the $ \alpha $-loss is twist-proper it can undo any form of twist, including that of unfairness criteria.
> >
> > ---
> >
> > **Extension**
> >
> > With regards to the reviewer's comments regarding applying the proposed method to other domains … Indeed, given the twist-properness / universality of the $ \alpha $-loss very much might be applicable to the suggested domain, but of course this is left for future work. (For robustness see: https://arxiv.org/pdf/1906.02314.pdf)

---

> > > ### Comment · Reviewer_tmha · 2022-08-09
> > > **Thank you for the detailed response!**
> > >
> > > Thanks a lot for the detailed response and the clarifications and changes! They strengthen the paper and address some of my concerns, esp. regarding clarity. I have updated my rating accordingly.
> > >
> > > However, I still have some remaining issues with the work, primarily wrt. its practical applicability to fairness problems. The requirements regarding access to a true posterior for EOO, which imo is an important fairness notion, and the limitations around small sample sizes (I appreciate being upfront here), which might be especially problematic for underrepresented groups at which fairness interventions are often primarily targeted, limit the utility of the work in many scenarios.
> > > While the paper is clearer now, the notation still feels hard to grasp in a number of places.

---

### Author Response · Authors · 2022-08-02
**General Remarks**

We would like to thank the reviewers for their time reviewing our submission and for the valuable comments made. We would like to first address some general comments first, with per-reviewer discussion under each individual review. In the general remarks, we discuss aspects of clarity and mathematical presentation; and also clarify the interpretability / explainability property discussed in the paper.

Note: we have indicated whenever a change in citation number has changed from the old version to the revised version, i.e., [14->15] for [14] in the old version and [15] in the new.

---

**Clarity and Mathematical Presentation**


We thank the reviewer for providing feedback regarding clarity and mathematical clarity. Just as the reviewers have identified, the mathematical rigor of our approach is one of our strengths. As such, we hope to keep much of the rigor whilst improving clarity.

We believe that although some parts of the narrative are mathematically dense, specifying the precise notation early in the paper is important for readers to understand the technical arguments whilst also compressing the notation required to understand parts of the paper. For example, the notation of “mixture measure” $M$ provides a convenient notation for instantiating TopDown. We found that utilizing densities / pdf notation (e.g., $p(x,y)$) becomes rather unwieldy as we wish to instantiate TopDown with many combinations of measures. Nevertheless, we have provided comments in the introduction of this notation to clarify this is just a notational convenience for breaking down the space of features and labels.

More generally, as many of the reviewers have suggested we have added “nuggets” of intuition around other technical parts of the paper to help with readability. This includes clarifying our contributions and prior work; and restructuring some components of the paper (the desiderata). Please see the revised manuscript.

---

**Interpretability / explainability**


We would also like to clarify our claim on interpretability / explainability as a few of the reviewers had comments regarding this. Interpretability here refers to the post-processing step, ie, the transformation from $ \eta_{\rm u} $ to $ \eta_{\rm f} $. For our case, this is the interpretability / explainability of the $ \alpha $-correction. If the “architecture” of $\alpha(\cdot)$ was complex, ie, random forests as mentioned by reviewer 2a51, then interpretability would be impossible. However, our $ \alpha $-tree is limited to a “decision tree” structure with axis aligned splits (either determined by a feature value if discrete or just a single threshold / inequality if continuous).

Another argument one could make against our interpretability is the difference of $ \alpha $-tree leaf node values compared to decision trees with labels. However despite $ \alpha $ taking values from $[-\infty, \infty]$, we can make some discretizations to build our intuitive picture:
1. If $ \alpha > 1 $, then the post-processing strengthens the prediction of the blackbox at the leaf;
2. If $\alpha \in (0, 1) $, then the post-processing weakens the prediction of the blackbox at the leaf, but maintains its prediction;
3. If $ \alpha < 0 $, then post-processing flips the prediction at the leaf.
Here, “strengthens prediction” means making the posterior value more “extreme” (closer to either 0 or 1, which makes the prediction more “certain”).

Take Figure 1 for example: to improve CVaR, the $ \alpha $-tree has a subtree which splits the (sub)domain for those who have high school education and those who do not. It then for one partition strengthens the prediction of the blackbox (bright red $\alpha = 1.61$) whilst weakening the prediction but keeping the prediction labels for the other partition (darker red $\alpha = 0.80$). This could possibly be used to audit the post-processing or analyzing the blackbox in hindsight.

---

### Meta-Review · Area_Chair_k8ah · 2022-08-27

**Recommendation:** Accept
**Confidence:** Less certain

**Metareview:**

This paper tackles the important and interesting problem of how to transform black-box models so that their outputs having improved fairness.  The proposal of using an "alpha-tree", an axis-aligned decision tree that re-weights the existing model, seems elegant and does indeed have some useful form of interpretability.  The primary issue---which was essentially raised by all reviewers---was that of the paper's clarity: the presentation often prioritizes rigor over readability.  I think the authors have done a good job in the comments explaining their work, and as no technical issues were raised, I recommend the paper for acceptance.  However, the authors should take steps to improve readability (e.g. my incorporating many of the discussion comments within the text, as space allows).

**Award:**

No

---

### Decision · Program_Chairs · 2022-09-14

Accept